# MonoQPM: Splitting Features into Concepts for Inherent Interpretability and Predictive Performance

## Abstract

Alongside superposition, polysemanticity is one of the primary obstacles to achieving interpretable deep neural networks. While these phenomena are typically intrinsically linked, the Quadratic Programming Enhanced Model (QPM) naturally decouples them. By representing classes with a sparse binary assignment of very few features, QPM prevents superposition by design on its final features but still exhibits polysemanticity. However, measuring polysemanticity is an open problem. This work proposes a utility-focused approach to measuring polysemanticity by quantifying the decrease in interference-induced activation errors, which yields the practical utility of tighter prediction sets using Conformal Prediction (CP). We explicitly disentangle the polysemantic features of QPM into monosemantic concepts to create the Monosemantic QPM (MonoQPM). Because its features are disentangled, Mono-QPM acts as a significantly more efficient Conformal Predictor. Additionally, we introduce CUBCars, a dataset providing ground truth information about shared concepts. Using this and other datasets, we demonstrate that polysemanticity emerges in QPM across all tested architectures, but is effectively alleviated by MonoQPM. For instance, MonoQPM guarantees 88% coverage using Adaptive Prediction Sets on ImageNet with just 66% of the frozen QPM's set size.

**Polysemantic QPM Feature**

Feature 1        Feature 1

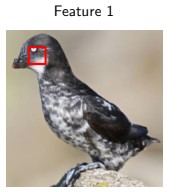 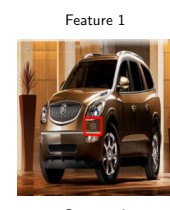

Concept 1       Concept 2

**Monosemantic MonoQPM Concepts**

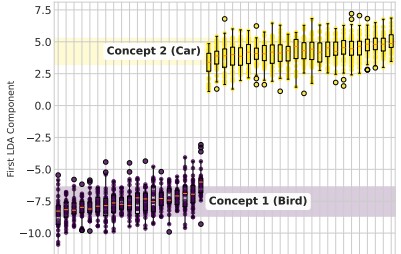

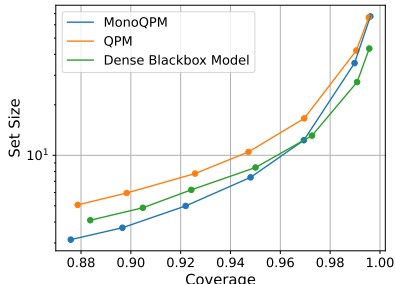

(a) Exemplary visualization for one polysemantic feature and two of its assigned classes. The red square denotes the location with maximum activation.

(b) 1D LDA projections of the feature activations of the feature in (a), grouped by class. The resulting concept clusters are highlighted.

(c) Average prediction set size (log-scale, lower better) vs. coverage for APS Conformal Prediction on CUB-2011. MonoQPM yields more efficient sets.

Figure 1: Overview of MonoQPM with exemplary visualization (a) for CUBCars. MonoQPM is derived post-hoc from an existing QPM by splitting each of its polysemantic features into monosemantic concepts. This feature splitting is achieved by clustering the ReLU activation signs. (b). Yellow and purple dots denote samples from CUB-2011 (Wah et al., 2011) and Stanford Cars (Krause et al., 2013) classes, respectively. Shaded regions indicate standard deviations of the fitted GMM concept clusters. Notably, MonoQPM perfectly disentangles these dataset concepts without any explicit supervision. Due to reduced interference, the resulting MonoQPM is a significantly more efficient Conformal Predictor, even on datasets without forced polysemanticity (c).

Figure 2: Exemplary visualization of one class representation for *Western Gull* from a QPM trained on the mixed CUBCars dataset. Below each heatmap we show all concepts that MonoQPM identifies within the five frozen polysemantic features, each represented by a prototypical sample chosen via its 1D LDA projection (Fig. 1b). The red squares mark the location of maximum activation. Above the line, for each feature, is the concept that MonoQPM assigns to *Western Gull*. Below the line are the remaining concepts of the same features, which QPM also activates on but which MonoQPM does not assign to this class. Thus, while QPM represents classes only through abstract polysemantic features, MonoQPM identifies the individual monosemantic concepts within each one, assigns them to the class, and thereby improves the explanations of both individual classes and single samples.

# 1 Introduction

As deep neural networks become increasingly capable, aligning them with human values is essential to mitigate potential risks (Wang et al., 2026). Towards that goal, there is an increasing need for *mechanistic interpretability*, understanding individual neurons to uncover what the model has actually learned. This contrasts with relying solely on the outputs of specific test samples, as models can learn to conceal misalignment (Hubinger et al., 2024). However, two intertwined phenomena, superposition and polysemanticity, pose a significant hurdle to understanding individual neurons (Elhage et al., 2022). Polysemanticity refers to individual neurons activating on entirely unrelated concepts in the data. Superposition refers to models that perform computation not in an axis(neuron)-aligned fashion, but rather using non-orthogonal vectors. Polysemanticity can be shown to robustly emerge when there are more relevant concepts in the data than neurons to encode them and the decoding happens in a dense fashion, i.e. the model can be doing computation in superposition (Marshall & Kirchner, 2024).

Interestingly, the Quadratic Programming Enhanced Model (QPM) (Norrenbrock et al., 2025b) prevents superposition on its final features as a consequence of its inherently interpretable structure. Nevertheless,

it has been noted to learn seemingly polysemantic features. Hence, QPM is uniquely suited to analyse polysemanticity without the confounding effects of superposition. However, accurately measuring this phenomenon remains a fairly open problem. Prior work has predominantly focused on quantifying alignment with discrete human concepts (Kopf et al., 2025), despite robust evidence that deep neural networks utilise features fundamentally different from those humans use (Ilyas et al., 2019).

Therefore, this work measures the phenomenon using a utility-focused approach that requires minimal assumptions about the learned features, as extensively described in Section 4. The core idea is measuring whether explicitly splitting up features into their concepts (Fig. 1a) removes false residual activations on irrelevant classes, a phenomenon known as interference (Kong et al., 2025). The practical utility of this effect is then captured by the disentangled QPM's efficiency as a Conformal Predictor, which strongly correlates with fewer false activations on unlikely classes that would otherwise inflate the prediction sets. As shown in Fig. 1c, our proposed Monosemantic QPM (MonoQPM), a disentangled version of the frozen QPM, significantly reduces the set sizes required to guarantee a certain accuracy when using APS (Romano et al., 2020) Conformal Prediction (CP). This proposed MonoQPM is generated by clustering the signs of the internal activations, inspired by Spline Theory (Balestriero et al., 2024; 2018). Specifically, the ReLU codes obtained on activating samples are reduced to 1 dimension using Principal Component Analysis, followed by Linear Discriminant Analysis. In this space, the samples are clustered using Gaussian Mixture Models with a varying number of components. This allows us to automatically identify the optimal number of concepts, determined by the minimum Bayesian Information Criterion (Schwarz, 1978). Classes are then assigned to each specific concept that they exhibit rather than to the broader polysemantic feature, leading to improved logits with fewer interference-induced false activations. The resulting MonoQPM provides a much clearer lens into the model's decision-making. As illustrated by the Western Gull class representation in Fig. 2, MonoQPM decomposes the abstract features of the QPM into distinct monosemantic concepts. This enables MonoQPM to succinctly explain individual classes and decisions using coherent concepts and allows for a more thorough auditing of it.

To additionally quantify the quality of disentanglement from an alternative perspective, we introduce the CUBCars dataset, a mixture of CUB-2011 and Stanford Cars. This provides a clear ground truth for classes that are highly unlikely to share a concept. Across a range of architectures, MonoQPM disentangles all cross-dataset features on CUBCars into their respective datasets with 100% accuracy, quantified by the novel Concept Classes Similarity. Fig. 1b shows one exemplary one-dimensional representation of the ReLU codes obtained for one feature and visualizes the evident polysemanticity in the internal activations.

Beyond CUBCars, our utility-focused evaluation demonstrates that QPM learns similarly polysemantic features across all probed real-world datasets, including ImageNet-1K (Russakovsky et al., 2015). We show that this polysemanticity is robustly reduced in MonoQPM. We supplement this evaluation with a DINOv3 (Siméoni et al., 2025) based approach, confirming that disentangled concepts respond to visually similar image patches far more consistently than the original polysemantic features. Since this work introduces a novel problem formulation, we also establish a strong baseline based on hierarchically clustering class representations. This allows us to sweep the number of concepts learned by every feature in relation to CP efficiency. As shown in Fig. 3, this efficiency exhibits a distinct local minimum, providing clear visual evidence that polysemanticity exists.

Our main **contributions** are as follows:

- **Studying Polysemanticity in Isolation (Section 4):** We propose utilizing the QPM (Norrenbrock et al., 2025b), which decodes contrastive features in a white-box manner without superposition, as a real-world testbed to study, disentangle, and quantify polysemanticity in isolation. This architectural transparency facilitates the actual disentanglement of features into individual, interpretable concept neurons with a directly traceable impact on the logits. Consequently, the quality of this disentanglement can be robustly evaluated from a utility-focused perspective via Conformal Prediction.

- **Baseline Visualizes Evident Polysemanticity (Fig. 3 and Section 4.4):** We propose a simple yet surprisingly effective baseline based on clustering class representations with a configurable degree of disentanglement. This allows us to sweep the level of disentanglement against Conformal Prediction

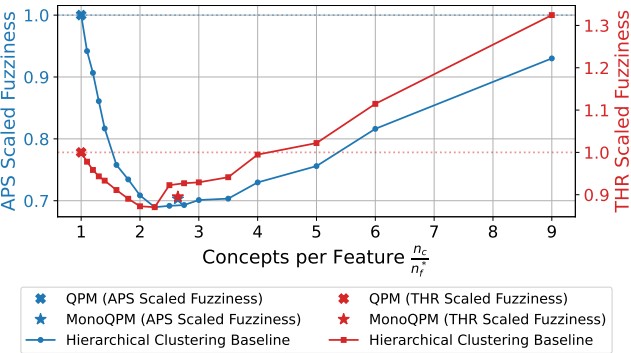

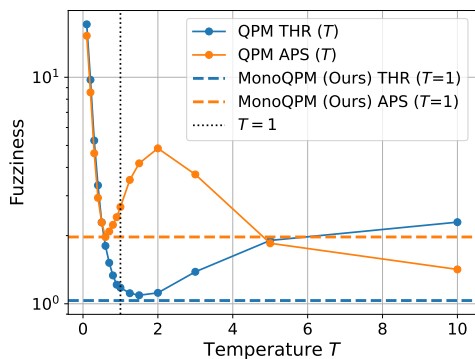

Figure 3: THR and APS Fuzziness as a fraction of the baseline QPM over the number of concepts for a CUBCars-trained ResNet-50 QPM. Both metrics show a distinct local minimum when splitting each feature into on average two concepts, thereby removing dataset interference.

Figure 4: Comparison of the fuzziness metrics achieved by our proposed MonoQPM on CUB-2011 compared to simply dividing the QPM logits by Temperature $T$: The results demonstrate that MonoQPM non-trivially enhances the logits, yielding simultaneous improvements in both THR and APS fuzziness.

efficiency, revealing a distinct local minimum below the baseline QPM. This result clearly indicates the presence of polysemanticity even in the absence of superposition.

- **Explicit Disentanglement via MonoQPM (Section 5):** We introduce the Monosemantic QPM (MonoQPM), a post-hoc transformation for frozen QPMs that explicitly disentangles individual features into their constituent concepts. Inspired by spline theory, MonoQPM is constructed by clustering the 1D representations of ReLU activations that are generated when a feature activates. This yields a more transparent architecture, as the global class representations are now composed of monosemantic concepts.

- **Dataset with Obvious Polysemanticity (Fig. 6, Section 6.2, and Table 4):** We introduce CUBCars, a union of CUB-2011 and Stanford Cars designed with a strong prior regarding class-concept independence. Alongside this, we introduce Concept Classes Similarity as a complementary metric to quantify the similarity of classes that are assigned to the same concept. MonoQPM decomposes features into concepts that not only reflect ground-truth class similarities but also show heightened visual consistency with DINOv3 features, ultimately yielding a more efficient conformal predictor.

- **Consistent Trends across Diverse Datasets (Table 2 and Section 6.3):** We demonstrate that polysemanticity occurs similarly across multiple architectures and real-world datasets, confirming it is not unique to our proposed benchmark. By successfully reducing polysemantic interference, MonoQPM achieves superior efficiency in Conformal Prediction, resulting in significantly smaller set sizes at identical coverage levels.

## 2 Related Work

Research on interpretability in deep neural networks can be broadly divided into applying post-hoc techniques (Kim et al., 2018; Bau et al., 2017; Fel et al., 2023; Kalibhat et al., 2023; Oikarinen & Weng, 2023; Kaiser et al., 2025) to black-box models, and training inherently interpretable models. While inherently interpretable architectures offer greater structural transparency, they frequently rely on further post-hoc analysis to achieve thorough interpretability. Because this work introduces a post-hoc transformation that explicitly disentangles polysemantic features within an inherently interpretable model, this section first provides an overview of classification models with built-in interpretability before discussing prior work on polysemanticity.

Inherently interpretable models can be classified as either concept-based models or prototype-based models. Both approaches employ a white-box final layer that utilises their respective features, either concepts or prototypes, as input.

Prototype-based models operate on the principle of *This looks like that* (Chen et al., 2019), where the final classification layer relies on the similarity between specific aspects of a given sample and a set of learned prototypes. While an intriguing idea, this similarity measure is typically unpredictable to humans (Kim et al., 2022; Hoffmann et al., 2021), resulting in deceptive interpretability. Notably, Baniecki & Biecek (2025) uses the CUB-2011 and Stanford Cars datasets as a strong prior of disjoint concepts to show that prototypical models can learn any kind of arbitrary, uninterpretable similarity measure, even representing birds exclusively with car prototypes. We similarly utilise this contrast between the datasets as a ground truth for minimal concept overlap.

Conversely, concept-based models aim to provide interpretable inputs to the final classification layer. The foundational Concept Bottleneck Model (CBM) (Koh et al., 2020) relies on explicit supervision for all such input features. To alleviate this annotation burden, more recent approaches utilise foundation models to generate pseudo-labels for concepts (Oikarinen et al., 2023; Rao et al., 2024). While these models typically employ standard linear layers for classification, a distinct subclass imposes strict architectural constraints to learn interpretable concepts entirely without supervision (Norrenbrock et al., 2022; 2024; 2025b;a). These architectures rely on extremely sparse, low-dimensional class representations composed of their features. Because a feature is similarly important across a specific subset of classes, it is forced to detect a general, shared concept between them. Furthermore, their class representations enable global interpretability, allowing for direct class comparisons independent of individual test samples. Within this subclass, we specifically employ the QPM (Norrenbrock et al., 2025b), as it exhibits polysemantic features but fundamentally cannot encode them in superposition.

## 2.1 Polysemanticity

The phenomena of polysemanticity and superposition pose the most significant challenges to mechanistic interpretability (Elhage et al., 2022). Since dense architectures compute in superposition by default, their internal features remain opaque and incomprehensible to human analysis. Polysemantic behaviour generally arises when a model must encode more concepts than it has dimensions, or when the theoretical advantages of redundant encoding supersede the penalties of interference (Elhage et al., 2022; Marshall & Kirchner, 2024). To address this, prior research primarily focuses on disentangling polysemantic representations in black-box models into monosemantic, human-understandable concepts for inspection or potentially steering (Templeton, 2024). Interpretability-focused methods can broadly be grouped into Sparse Autoencoders (Bricken et al., 2023; Gao et al., 2025; Rajamanoharan et al., 2024), attribution clustering (Dreyer et al., 2024), and layer activation clustering (Kopf et al., 2025; Dreyer et al., 2025), typically operating within a lower-dimensional or foundation-model latent space. These techniques successfully align individual representations within the probed black-box models with human-understandable concepts. However, they cannot explain the entire representation, let alone the entire prediction, as these more disentangled representations are either not utilised at all or are integrated in a black-box manner that incurs a noticeable drop in performance (Chen et al., 2025; Karvonen et al., 2025).
On the other side, reducing polysemanticity and interference has been evaluated aiming for downstream performance (Sawmya et al., 2024; Kong et al., 2025). However, in these works, individual neurons are still mostly uninterpretable and decoded densely, although an explicit splitting led to improved predictive performance. To our knowledge, no prior work has attempted to explicitly split polysemantic neurons into individual concepts that are utilised in a white-box manner for prediction. This transparency prevents superposition, enables evaluation via predictive performance gains, and leads to further improved global interpretability of the model.

## 3 Preliminaries

### 3.1 QPM

This section describes the Quadratic Programming Enhanced Model (QPM) (Norrenbrock et al., 2025b), which we use as the basis for our analysis. It transforms every image $\boldsymbol{X} \in \mathbb{R}^{w_X \times h_X \times 3}$ into an interpretable feature vector $\boldsymbol{f}^* \in \mathbb{R}^{n_f^*}$, which is then mapped to the output logit vector $\boldsymbol{z} \in \mathbb{R}^{n_y}$ via a sparse binary linear layer with weights $\boldsymbol{W}^* \in \{0,1\}^{n_y \times n_f^*}$. Functionally, this layer acts as an explicit assignment between the classes $n_y$ and the strictly smaller set of selected features $n_f^* < n_y$. Typically, a maximum of five features is assigned to each class ($n_{wc} \leq 5$). This threshold is grounded in cognitive science, as humans can comfortably process up to five concepts simultaneously without difficulty (Miller, 1956). Furthermore, empirical results demonstrate that relaxing this sparsity constraint yields no significant performance gains on the real-world datasets used to evaluate the QPM.

QPM is obtained following the pipeline shown in Figure 5: First, a dense model is trained with an auxiliary Feature Diversity Loss (Norrenbrock et al., 2022) that encourages the emergence of disentangled features. Then, a quadratic programming (QP) problem is solved to select a small subset of features and jointly assign them to classes they correlate with, while encouraging diversity between selected features. Finally, the selected features are normalized and undergo fine-tuning while their class assignments remain strictly fixed. During this phase, the features adapt specifically to their assigned classes, culminating in the final QPM. Consequently, these features develop highly desirable properties for interpretability, as they are forced to detect general concepts shared among their target classes.

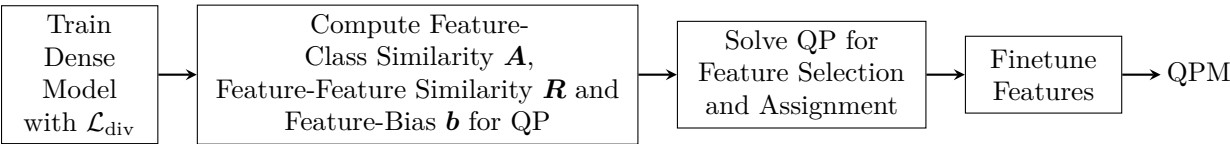

Figure 5: Overview of the pipeline to construct a QPM

### 3.2 Conformal Prediction

We evaluate our model using Conformal Prediction (CP) (Vovk et al., 2005), a robust framework with minimal assumptions that guarantees a maximum error rate $\alpha$ (see (Correia et al., 2024) for a detailed introduction). Specifically, we employ split CP (Papadopoulos et al., 2002), which uses a separate calibration dataset $\mathcal{D}_{cal}$ to calibrate the model. For any test sample $(\boldsymbol{X}_{test}, y_{test})$ exchangeable with $\mathcal{D}_{cal}$ (a slightly weaker assumption than i.i.d.), it guarantees:

$$P(y_{test} \in \mathcal{B}(\boldsymbol{X}_{test})) \geq 1 - \alpha \tag{1}$$

The prediction set $\mathcal{B}(\boldsymbol{X}_{test})$ is built using a nonconformity score $s(\boldsymbol{X}, y)$, which measures how poorly a candidate label $y$ fits the input $\boldsymbol{X}$. Computing these scores over $\mathcal{D}_{cal}$ yields a score set $\mathcal{S} = \{s(\boldsymbol{X}_{cal}, y_{cal}) \mid (\boldsymbol{X}_{cal}, y_{cal}) \in \mathcal{D}_{cal}\}$, which is then used to construct the final prediction sets:

$$\mathcal{B}(\boldsymbol{X}_{test}) = \{y \in \mathcal{Y} : s(\boldsymbol{X}_{test}, y) \leq \text{Quantile}(1 - \alpha, \mathcal{S})\} \tag{2}$$

Thus, a class $y$ is only included if its score falls below the $(1 - \alpha)$ quantile of $\mathcal{S}$, where $(1 - \alpha)$ is the target coverage. The scoring function $s(\boldsymbol{X}_{test}, y)$ is highly flexible. It only needs to capture a valid notion of conformity and produce sufficiently distinct values for proper quantile resolution. Different CP methods aim for different coverages: *Threshold* (THR) (Sadinle et al., 2019) optimally achieves unconditional coverage (ensuring average coverage across the test set) while *Adaptive Prediction Sets* (APS) (Romano et al., 2020) aims for conditional coverage (maintaining coverage for individual, often difficult, samples). These two methods differ in their non-conformity score, with THR using the negative probability of each class and APS using the sum of all probabilities that are higher:

$$s_{THR}(\boldsymbol{X}, y) = -\hat{\boldsymbol{z}}_y \quad \text{and} \quad s_{APS}(\boldsymbol{X}, y) = \sum_{i:\hat{\boldsymbol{z}}_i \geq \hat{\boldsymbol{z}}_y} \hat{\boldsymbol{z}}_i \quad \text{with} \quad \hat{\boldsymbol{z}} = \text{softmax}(\boldsymbol{z}) \tag{3}$$

Consequently, the scores differ in how they react to changes on the logits: THR is fairly unaffected by small increases on logits for unlikely classes, as it only causes a small drop in the probability of the more likely ones. APS in contrast is very sensitive. Since the reduced probabilities of likely classes are cumulatively summed, it takes more classes to reach the threshold, causing a significantly larger prediction set $\mathcal{B}(\boldsymbol{X}_{\text{test}})$ (Angelopoulos et al., 2021). The difference in the CP methods causes opposite temperature scaling behaviour, as shown in Fig. 4.

All CP methods are designed to guarantee the same error rate $\alpha$. The quality of a conformal predictor is thus evaluated based on the size of the resulting prediction sets needed to reach that rate. Additionally, Observed Fuzziness (Vovk et al., 2017) measures the sum of p-values $p(\boldsymbol{X}_{\text{test}}, y)$ of all false labels and can be used as an $\alpha$-agnostic measure of the quality of the predictor:

$$\text{Fuzziness}(\boldsymbol{X}_{\text{test}}, y_{\text{test}}) = \sum_{y \in \mathcal{Y} \backslash \{y_{\text{test}}\}} p(\boldsymbol{X}_{\text{test}}, y) \quad \text{with} \quad p(\boldsymbol{X}_{\text{test}}, y) = \frac{|\{s' \in \mathcal{S} : s' \geq s(\boldsymbol{X}_{\text{test}}, y)\}| + 1}{|\mathcal{S}| + 1} \quad (4)$$

Fuzziness is averaged across the test dataset and a lower fuzziness quantifies a more efficient conformal predictor.

## 4 A New Perspective on Polysemanticity

We investigate polysemanticity by explicitly disentangling features into individual concepts. To evaluate the success of this approach, we analyse the resulting predictive performance utilising the interpretable image classification model QPM (Norrenbrock et al., 2025b).

### 4.1 Why QPM

We choose QPM for our analysis as it shows several properties that make it particularly well suited for our analysis of its polysemantic neurons:

- **No Superposition:** The sparse binary assignment precludes superposition in the features, allowing us to isolate and investigate polysemanticity independently.

- **White-Box Connection between Polysemantic Neurons and Output:** The sparse binary final layer directly connects features to the output logits, allowing us to explicitly disentangle polysemantic neurons into their constituent concepts and analyze their direct impact on the network's predictions. This architectural transparency enables a novel, utility-focused perspective on measuring polysemanticity—evaluated via downstream model performance rather than mere alignment with human concepts—which we discuss in Section 4.3.

- **Contrastive Features:** QPM's features emerge as highly contrastive (Norrenbrock et al., 2025b), meaning they typically exhibit distinct, binary-like activation patterns for a given input. This simplifies subsequent analysis, as each feature can be interpreted as a discrete detector for at least one concept, rather than a continuous measure. Consequently, our methodology is designed to model these polysemantic, contrastive neurons as representing a logical OR, rather than a logical AND, relationship among their constituent concepts.

#### 4.1.1 Why Polysemanticity Emerges in QPM

Polysemanticity emerges within the QPM as every feature $d$ is assigned to a set of classes $\mathcal{Y}^d$, where each class $y$ is correlated with the presence of a set of general concepts in the data $\mathcal{C}^y = \{c_1^y, \ldots, c_{|\mathcal{C}^y|}^y\}$. To successfully minimise the training loss, the model must ensure that the logit corresponding to the ground-truth class strictly exceeds those of all other classes. Consequently, if the target label belongs to the assigned set $\mathcal{Y}^d$, the feature $d$ is heavily incentivised to activate strongly. It could essentially learn to detect the complex classes already and predict high activations if any of $|\mathcal{Y}^d|$ classes is deemed likely:

$$f_d^* \propto \max_{y \in \mathcal{Y}^d} P(y|\boldsymbol{X}) \quad (5)$$

However, a more efficient solution is the learning of few simpler shared general concepts between the classes,

$$f_d^* \propto \max_{c \in \mathcal{C}^d} P(c|\boldsymbol{X}), \quad \text{with} \quad |\mathcal{C}^d \cap \mathcal{C}^y| \geq 1 \quad \forall \, y \in \mathcal{Y}^d \tag{6}$$

where $\mathcal{C}^d$ denotes the set of concepts that the feature $d$ activates upon. Intuitively, learning detectors of shared concepts is easier, as they are simpler, making $P(c|\boldsymbol{X})$ easier to learn than $P(y|\boldsymbol{X})$. Additionally, the cardinality of $\mathcal{C}^d$ should be significantly lower, $|\mathcal{C}^d| \ll |\mathcal{Y}^d|$, when features are assigned to highly similar classes, as is typically observed (Norrenbrock et al., 2025b). While the fundamental premise of the QPM is to learn a single, shared concept per feature across all assigned classes ($|\mathcal{C}^d| = 1$), the emergence of polysemanticity remains a notable limitation, particularly when scaling to massive datasets like ImageNet-1K with 1000 classes. However, it should be noted that it is also impossible for a human to identify a single concept that is equally important to 100 classes on average. Hence, a potential label for the feature would likely not exist and it would therefore seem polysemantic either way. The proposed method aims to distinguish between seemingly uninterpretable features that are *monosemantic for the neural network*, but potentially detect concepts not yet named by humans (Hewitt et al., 2025), and polysemantic features that consist of distinct concepts present in the data.

## 4.2 Problem Definition

The primary goal of this work is to introduce a prediction-based evaluation for the disentanglement of polysemantic neurons. This approach serves as a quantifiable measure of both polysemanticity and the quality of the resulting disentanglement, functioning entirely independently of any assumptions regarding the specific concepts learned by the model. Specifically, we aim to find different concepts in the data that the probed QPM has recognized as being distinct. This is in contrast to prior work, which relies on alignment with human concepts, e.g. using text (Kopf et al., 2025) to measure polysemanticity and the quality of disentanglement. Only measuring alignment with human concepts is problematic as the model's concepts may not align with those and deep neural networks have been shown several times to use different concepts than humans (Ilyas et al., 2019; Kim et al., 2022; Burgert et al., 2025; Hewitt et al., 2025). From the lens of the model, concepts that are close in language space, e.g. *black* and *brown*, might be perceived entirely differently but just happen to be encoded as different data concepts into the same polysemantic neuron. While a language-based approach happily summarizes these examples as *dark* with high language similarity, the proposed evaluation measures what matters: Is the model a better predictor if the feature is split up into *black* and *brown*?

Towards that goal, we explicitly disentangle the features of a trained QPM to create a *Monosemantic QPM (MonoQPM)*. This MonoQPM effectively operates with the same features as the original QPM, but with its polysemantic features split up into their different meanings and the assignment between concepts and classes adapted accordingly. Instead of returning a polysemantic feature vector $\boldsymbol{f}^* \in \mathbb{R}^{n_f^*}$, the MonoQPM returns a split up feature vector $\boldsymbol{f}^{\mathrm{mono}} \in \mathbb{R}^{n_c}$ of $n_c$ concepts, where every polysemantic feature is split up into its different meanings. The monosemantic assignment matrix $\boldsymbol{W}^{\mathrm{mono}} \in [0,1]^{n_y \times n_c}$ then reflects which class is associated with which of the concepts of its assigned features.

## 4.3 Measuring Polysemanticity with a Utility-Focused Approach

We propose measuring how well a method splits up the individual, more interpretable features of a trained QPM into its individual concepts with minimal assumptions. Specifically, if a feature captures multiple concepts $\mathcal{C}^d$, it will activate on any of them which leads to an increased logit of all assigned classes $\mathcal{Y}^d$ instead of just the classes that share the concept $\mathcal{Y}^c$. If only one of the concepts, $c \in \mathcal{C}^d$, is present in the image, there are wrong activations on those classes that do not share the concept $c$:

$$\Delta \boldsymbol{z}^d = f_d^* \cdot (\mathbb{1}^{\mathcal{Y}^d} - \mathbb{1}^{\mathcal{Y}^c}) \tag{7}$$

Here, $\mathbb{1}^{\mathcal{Y}^s} \in \{0,1\}^{|\mathcal{Y}|}$ denotes the indicator vector for any subset $\mathcal{Y}^s \subseteq \mathcal{Y}$, where its $y$-th element is 1 if $y \in \mathcal{Y}^s$, and 0 otherwise. This $\Delta \boldsymbol{z}^d$ will primarily be focused on the classes with relatively low activation overall. Otherwise, it would have likely been prevented as the training loss is significantly affected else.

Hence, a metric is needed that is sensitive to false residual activations on classes that the QPM predicts as unlikely. One such metric is the Fuzziness of APS Conformal Prediction (Section 3.2) Therefore, we propose measuring the quantity of false residual activations as a consequence of polysemanticity of all features using the framework of Conformal Prediction.

In order to keep different methods comparable and focus on clearly showing *that* polysemanticity exists even in absence of superposition in QPM, we apply minimal changes to the QPM. Specifically, given a frozen QPM with $\boldsymbol{W}^*$, we evaluate the predictive performance using CP before and after disentanglement. The only change is using an adapted assignment matrix $\boldsymbol{W}^{\text{mono}}$, which reflects the disentanglement of the features. This then causes the resulting extended feature vector $\boldsymbol{f}^{\text{mono}}$ and altered logits $\boldsymbol{z}^{\text{mono}} \in \mathbb{R}^{n_y}$.
In order not to change the top-1 prediction, we determine which concept of $\boldsymbol{f}^* \in \mathbb{R}^{n_f^*}$ is present in a given image based on the maximum logit of every assigned class:

$$f_c^{\text{mono}} = \begin{cases} f_d^* & \text{if } \max_{y \in \mathcal{Y}^c} z_y = \max_{y \in \mathcal{Y}^d} z_y, \\ \mu_d & \text{otherwise.} \end{cases} \tag{8}$$

Note that when another concept has a logit with higher activation, we replace the activation of the inactive concept with $\mu_d$, the value of $f_d^*$ when no evidence for it is found.
In order to remove $\Delta \boldsymbol{z}^d$ (eq. (7)) using $\boldsymbol{W}^{\text{mono}}$, we adapt the logits so that only the maximally activating assigned concept of each class contributes to the class logit in the resulting logits $\boldsymbol{z}^{\text{mono}}$:

$$z_y^{\text{mono}} = \sum_{d \in \mathcal{F}^d} \max_{c \in \mathcal{C}^d} f_c^{\text{mono}} \cdot w_{y,c}^{\text{mono}}, \tag{9}$$

where $\mathcal{F}^d$ denotes the non-zero indices in $\boldsymbol{W}_{:,d}^*$ and $\mathcal{C}^d$ denotes the indices of the concepts of feature $d$ in $\boldsymbol{W}^{\text{mono}}$. The logit of class $y$ is therefore the sum across all its assigned features, where for every feature the maximum across its assigned concepts is used.

APS CP is most affected by the wrong residual activations (eq. (7)) we are looking to remove, as discussed in Section 3.2. Additionally, the QPM is trained to minimize negative log-likelihood (NLL). Naturally, the model's polysemanticity will not be significantly negatively affecting this metric. Therefore, we positively value any explicit disentanglement of polysemantic features into concepts that improves the APS Fuzziness, directly reflecting less residual activations on unlikely classes, without harming NLL. An excellent disentanglement additionally improves THR Fuzziness, even though it is very reliant on not reducing the logit of the ground truth.
On top of the utility-focused evaluation, we introduce an experiment to measure whether different methods are able to split features correctly, when the classes $\mathcal{Y}^d$ can be assumed not to share any concept. To quantify it with a notion of ground truth, we include a set of experiments on a newly mixed CUBCars dataset, which is simply the union of CUB-2011 and Stanford Cars. Consistent with prior work (Baniecki & Biecek, 2025), we assume that these two datasets, comprising natural images of birds and man-made cars, share no underlying concepts across their classes, with the sole exception of color. Consequently, since car classes exhibit significant color variance, any feature assigned to both bird and car classes almost certainly encodes at least two distinct underlying concepts. Notably, this is precisely the behavior observed when training the QPM on this composite dataset, as illustrated in Fig. 6.

## 4.4 Hierarchical Clustering Baseline

This section introduces a baseline with a configurable degree of disentanglement based on hierarchically clustering class representations. Specifically, we cluster the final layer representations $\boldsymbol{W} \in \mathbb{R}^{n_y \times n_f}$ of the dense model before applying the QP. We use these dense class representations, as they have been shown to encode a significant amount of information about the class and its relationship to other classes (Sammani & Deligiannis, 2025).
This approach aims to find clusters of maximally distinct classes, hence it by design does not return shared concepts between classes. Consequently, each class is assigned to one concept per initially assigned feature $d$:

$$\sum_{c \in \mathcal{C}^d} w_{y,c}^{\text{mono}} = 1 \qquad \forall\, y \in \mathcal{Y}^d \tag{10}$$

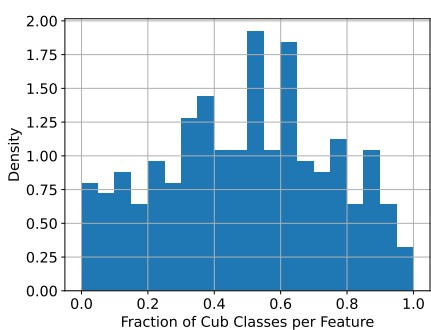
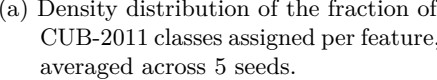
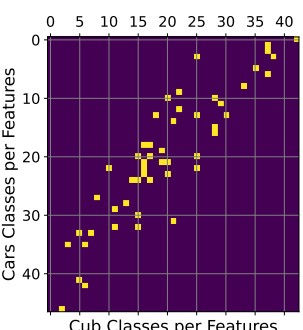

(a) Density distribution of the fraction of CUB-2011 classes assigned per feature, averaged across 5 seeds.

(b) Joint distribution of CUB and Cars class assignments per feature for a representative seed.

Figure 6: Dataset preference of QPM with 50 features and 5 assignments per class trained on the mixed CUBCars dataset: The figures show that features are not generally assigned to just one dataset.

For clustering, we apply agglomerative hierarchical clustering using Ward's minimum variance method (Ward Jr, 1963).

Clustering each feature $d$ independently, the pairwise distances $M^{(d)} \in \mathbb{R}^{|\mathcal{Y}^d| \times |\mathcal{Y}^d|}$ between all classes $u, v \in \mathcal{Y}^d$ assigned to feature $d$ are computed:

$$M_{u,v}^{(d)} = \|\boldsymbol{W}_u - \boldsymbol{W}_v\|_2 \tag{11}$$

Using this matrix, we obtain a dendrogram for every feature, where classes are merged based on the minimum increase in total within-cluster variance. This increase is used to evaluate the model at varying degrees of disentanglement. Specifically, to reach a target number of concepts $n_c$, we iteratively select and split the feature whose subdivision yields the greatest reduction in total within-cluster variance until the target number $n_c$ is reached. This formulation allows us to sweep the number of splits we perform and explicitly adapt the model's decision-making pipeline, as shown in Fig. 3. Evidently, even this simple baseline showcases that polysemanticity exists without superposition in QPM, as splitting features leads to improved CP performance. However, this baseline has several drawbacks. It conceptually precludes shared concepts even though it is likely occurring, as discussed in Section 4.1.1. Additionally, it is biased towards predictive performance, as similar classes are grouped together by design. Practically, it requires dense representations and has no direct way of determining how many concepts are present in a feature, relying solely on the number of clusters we choose to split into.

## 5 MonoQPM

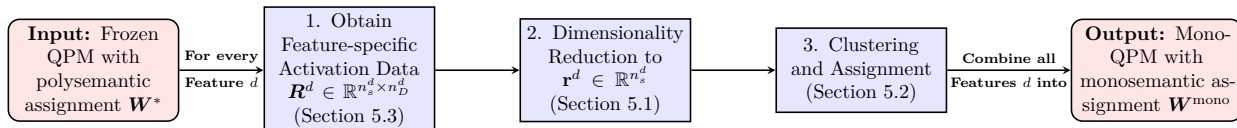

Figure 7: Overview of the MonoQPM feature decomposition pipeline.

After introducing the problem formulation, novel evaluation strategy and a baseline, this section presents a more sophisticated approach, which has none of the drawbacks of the baseline. It allows shared concepts, has minimal assumptions, requires nothing but the trained QPM with its training data and automatically determines the number of concepts $n_c$. This proposed method turns each feature $d$ of a frozen QPM individually into more monosemantic concepts. The resulting Monosemantic QPM (MonoQPM) offers elevated global interpretability, as classes that share the same concept are more likely to actually share a similar concept from a data perspective too. MonoQPM is based on the idea of clustering the internal

activations of the probed QPM, inspired by Spline Theory (Balestriero et al., 2018; 2024).

As shown in Table 5, the proposed pipeline can utilize various feature specific inputs $\boldsymbol{R}^d \in \mathbb{R}^{n_s^d \times n_D^d}$ where each of the $n_s^d$ samples with this feature is described by a vector of length $n_D^d$. For splitting any feature $d$, we only consider the samples, for which the label is in $\mathcal{Y}^d$ and that are correctly classified on the training dataset. This filtering focuses the analysis on distinguishing exactly what the feature responds to, ensuring robustness against mislabeled examples that the QPM failed to learn. As our proposed input is tailored to the following pipeline, we first describe the following pipeline and conclude with the exact definition of how the internal activations are processed in Section 5.3. After obtaining $\boldsymbol{R}^d$, we apply a technique similar to Fisherfaces (Belhumeur et al., 2002) to project the activations into a one-dimensional space, in which distinct concepts can be separated. In this one-dimensional space, we apply GMM-based clustering to find the different concepts and use the Bayesian Information Criterion (BIC) (Schwarz, 1978) to determine how many concepts are present in the feature, as described in Section 5.2.

Finally, after all individual features are split independently into their individual concepts, the entire model and its prediction is altered based on these splits. This results in the monosemantic assignment $\boldsymbol{W}^{\mathrm{mono}}$, which causes altered $\boldsymbol{f}^{\mathrm{mono}}$ and $\boldsymbol{z}^{\mathrm{mono}}$ as described in Section 4.3. An overview of the proposed pipeline is shown in Fig. 7.

### 5.1 Dimensionality Reduction

This section describes the dimensionality reduction used to compress the high-dimensional ReLU code into a one-dimensional space so that one can apply simple clustering algorithms. For dimensionality reduction, we start by using Principal Component Analysis (PCA) (Pearson, 1901), and preserve all the principal components above the noise level, as determined by the participation ratio (Gao et al., 2017)

$$\mathrm{PR}(\mathbf{R}) = \frac{(\sum_i \lambda_i(\boldsymbol{\Sigma}))^2}{\sum_i \lambda_i(\boldsymbol{\Sigma})^2} \quad \text{where} \quad \boldsymbol{\Sigma} = \mathrm{Cov}(\mathbf{R}), \tag{12}$$

where $\lambda_i$ describes the $i$-th Eigenvalue of the covariance matrix $\boldsymbol{\Sigma}$. We project the input data $\mathbf{R}^d$ to this PCA space $\boldsymbol{P} \in \mathbb{R}^{n_s^d \times n_{\mathrm{PCA}}}$, where $n_{\mathrm{PCA}} = \mathrm{round}(PR(\boldsymbol{R}^d))$ is the result of rounding to the next integer.

Afterwards, we use Linear Discriminant Analysis (LDA) (Rao, 1948) to project the data to one component $\mathbf{r}^d \in \mathbb{R}^{n_s^d}$. We choose to maintain only the first component, as it is sufficient to robustly disentangle polysemantic features, one-dimensional data can be directly visualized and it enables a significantly faster and optimal following clustering. The proposed dimensionality reduction is related to Fisherfaces (Belhumeur et al., 2002), which applies PCA as means of stabilizing the following LDA maintaining all $|\mathcal{Y}_d|-1$ components. This difference is due to different goals: MonoQPM aims to distinguish between concepts that are shared between the classes in an easily analyzable space, while Fisherfaces does classification using these components.

### 5.2 Clustering and Assignment

Using the compressed representation of internal activations for the individual feature $d$, $\mathbf{r}^d$, we aim to determine its concepts and their assignment to classes. Towards that goal, we evaluate a range of potential cluster counts $m$ in the data. This is done by fitting a Gaussian Mixture Model (GMM) $\mathcal{G}_m = \{(w_j, \mu_j, \sigma_j^2)\}_{j=1}^m$ and evaluating its BIC (Schwarz, 1978):

$$\mathrm{BIC}(m) = \nu(m) \ln(n_s^d) - 2 \ln(\hat{L}(m)) = (3m - 1) - 2 \ln(\hat{L}(m)) \tag{13}$$

where $\nu(m)$ is the total number of estimated parameters in the GMM and $\hat{L}(m)$ is the maximized value of the likelihood function for the model with $m$ components.

In order to get optimal solutions deterministically, we initialize the clusters using the optimal $k$-Means solution, which is computable in one dimension via Fisher-Jenks Natural Breaks (Fisher, 1958). The sweeping of number of clusters is shown in Fig. 8a. We choose the number of components which minimizes the BIC as the number of concepts $n_c^d = \mathrm{argmin}(\mathrm{BIC}(m))$ of $d$, as it is designed for it (Schwarz, 1978). Notably, a difference of just 10 is typically already considered very strong evidence that a specific model is a more appropriate fit to the data (Kass & Raftery, 1995). In contrast, when MonoQPM splits a feature, the

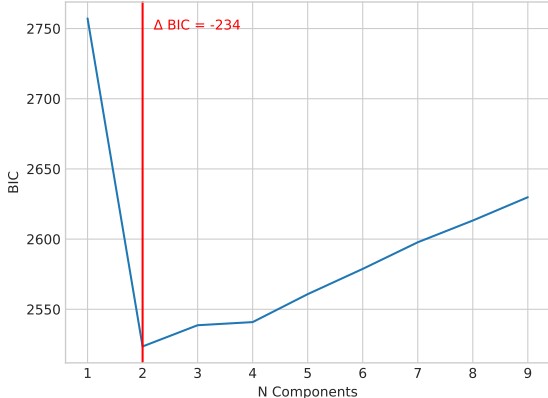
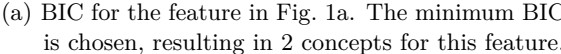
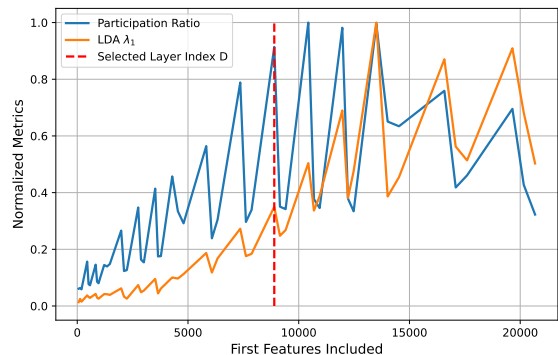

(a) BIC for the feature in Fig. 1a. The minimum BIC is chosen, resulting in 2 concepts for this feature.

(b) Participation Ratio (eq. (12)) and Eigenvalue of first LDA component over Depth for feature in Fig. 1a. Note that the applied ResNet-50 uses bottleneck layers with reduced dimensionality.

Figure 8: Visualization of the pipeline steps employed to obtain a MonoQPM.

difference is typically an order of magnitude higher. This constitutes further evidence for polysemanticity. After obtaining the GMMs $\mathcal{G}_{n_c^d} = \{(w_j, \mu_j, \sigma_j^2)\}_{j=1}^{n_c^d}$ that represent the data, we assign the initial classes $\mathcal{Y}^d$ to these new concepts $\mathcal{C}^d$. Towards that goal, we compute the probabilities for all samples in $\boldsymbol{r}^d$ to belong to any of these components $\boldsymbol{P} \in [0, 1]^{n_s^d \times n_c^d}$. For every class $y$, we then compute the average probability that its samples belong to any of the modes $\boldsymbol{p}^y \in [0, 1]^{n_c^d}$. We then assign this class to all concepts for which the average probability is above a threshold $\tau$:

$$\boldsymbol{W}_{y,i(c)}^{\mathrm{mono}} = \begin{cases} 1 & \text{if} \quad \boldsymbol{p}_c^y \geq \tau \\ 0 & else \end{cases} \tag{14}$$

Here, $i(c)$ returns the index in $\boldsymbol{W}^{\mathrm{mono}} \in [0, 1]^{n_y \times n_c}$, that corresponds to the specific concept index $c$. A class that is evenly distributed across all concepts, would have an average likelihood of $\boldsymbol{p}_{\mathrm{uniform}} = \frac{1}{n_c^d}$. Hence, $\tau$ cannot logically exceed this value, as some classes might otherwise remain unassigned. Therefore, $\tau$ should scale with it, $\tau = \beta \cdot \boldsymbol{p}_{\mathrm{uniform}}$, where $\beta$ is chosen to ensure that all classes assigned to a concept $c$ have significant overlap with it. A resulting clustering is visualized in Fig. 1b and in the appendix.

### 5.3 Spline Input

Spline Theory suggests that the model partitions the input plane into regions, and ultimately tries to have stable regions for every class (Humayun et al., 2024). We hypothesize that a similar phenomenon occurs in the feature space of QPM, where distinct concepts, comparable to distinct classes one layer below, of the same feature might lie in different regions. To show that this aspect can generally be used to disentangle QPM features, we use the simplest feature based on that theory, similar to *Feature 1* (Balestriero et al., 2024):

$$\mathbf{r}_j^{(l)} = \mathrm{mean}_{m,n}\left[\mathbb{1}\left(\mathbf{A}_{j,m,n}^{(l)} > 0\right)\right] \tag{15}$$

where $\boldsymbol{A}^{(l)}$ is the feature map of the $l$-th layer of the QPM and $\mathbf{r}^{(l)}$ just encodes which fraction of each channel is active. The goal is to concatenate all the layer-wise codes $\mathbf{r}^{(l)} \in [0, 1]^{n_l}$ into one sample representation $\boldsymbol{r}_{\mathrm{sample}} \in [0, 1]^{n_D^d}$

$$\mathbf{r}_{\mathrm{sample}} = [\mathbf{r}^{(1)}; \mathbf{r}^{(2)}; \dots; \mathbf{r}^{(D)}], \tag{16}$$

where $D$ is the number of layers to concatenate and $n_D = \sum_{l \leq D} n_l$. However, it is advisable that $D < L$, the number of layers, as the information encoded in the activations is declining with increasing network depth, as

Table 1: Statistical overview of datasets.

| Dataset | CUB-2011 | Stanford Cars | CUBCars | FGVCAircraft | ImageNet-1K |
|---|---|---|---|---|---|
| # Classes $n_y$ | 200 | 196 | 396 | 100 | 1000 |
| # Training | 5 994 | 8 144 | 14 138 | 6 667 | 1 281 167 |
| # Testing | 5 774 | 8 041 | 13 815 | 3 333 | 50 000 |

the representation starts to collapse to the classifier decision (Feng et al., 2022). As we are looking to partition features into general concepts, rather than specific classes, we automatically determine the optimal $D$ for every feature by evaluating how many dimensions our dimensionality reduction would preserve. Specifically, we use the participation ratio (eq. (12)) as measure of effective dimensionality, for which an example is shown in Fig. 8b alongside the first Eigenvalue of the following LDA on the projected data. Evidently, the participation ratio declines while the discriminability increases. For our pipeline, this would mean trading general information for class-specific ones. Therefore, we heuristically choose the last maximum before the global maximum of the participation ratio (eq. (12)):

$$D = \underset{l < l_{\max}}{\arg\max}\, \mathrm{PR}(\mathbf{R}^{(l)}) \quad \text{where} \quad l_{\max} = \underset{l \in \{1,\dots,L\}}{\arg\max}\, \mathrm{PR}(\mathbf{R}^{(l)}) \tag{17}$$

After obtaining $D$, the input to our proposed method is the concatenation of all layer representations of all samples in which our model likely recognized this feature $d$, denoted as $\mathbf{R}^d \in [0,1]^{n_s^d \times n_D^d}$. It determines for each feature individually how many concepts the QPM has learnt and explicitly splits them up, leading to the more interpretable MonoQPM.

## 6  Experiments

This section describes the experiments following Section 4.3 designed to show that polysemanticity exists in QPM and that the MonoQPM reduces it. We evaluate our methods on the datasets, CUB-2011, Stanford Cars, and ImageNet-1K, and architectures, ResNet (He et al., 2016)-50 and -34, Inception-v3 (Szegedy et al., 2016), and Swin-Transformer (Liu et al., 2021), used in the original QPM work (Norrenbrock et al., 2025b). Additionally, we use the mixed CUBCars dataset, as direct union of CUB-2011 and Stanford Cars and FGVC-Aircraft (Maji et al., 2013) for our ablations as a dataset with fewer classes, shown in Table 1. After first introducing the implementation details and metrics, we present the results of our experiments, showcasing that polysemanticity exists across the range of probed QPMs. Finally, we show the impact of the design choices of MonoQPM in Section 6.4.

### 6.1  Implementation Details

This section describes the implementation details of our experiments. We generally train multiple QPM's with different random seeds and report the mean results for each dataset and architecture combination. The compactness ablations in Fig. 10 are averaged across 3 seeds, all other results across 5 seeds. The hyperparameters exactly mirror the original QPM paper (Norrenbrock et al., 2025b), with CUBCars being trained with the CUB-2011 setting. We always compare different methods of disentangling the features each applied to the same frozen QPM. We set $\tau = \frac{1}{2 \cdot n_c^d}$, which was empirically optimized on ResNet-50 QPMs with different random seeds. We follow prior work (Norrenbrock et al., 2025a) to obtain our CP results. Specifically, we split 10 samples per class of the test dataset to use as our exchangeable calibration dataset $\mathcal{S}$. We implement CP using the *torchcp* (Huang et al., 2024) framework and use *scikit-learn* (Pedregosa et al., 2011) for GMMs and *jenkspy* (Viry, 2024) for the optimal breaks. Note that we limit the sweeping of the depth, as described in Section 5.3, to a maximum of 5000 samples, which is only coming to affect on experiments with ImageNet-1K. Additionally, for ImageNet-1K, we omit initializing the GMM with optimal breaks due to time constraints and instead use 10 $k$-Means++ (Arthur & Vassilvitskii, 2007) initializations per swept $k$ with default parameters. We compare this approach with our proposed deterministic version in Section 6.4. In order to include the number of components with minimum BIC in our sweep, we choose the

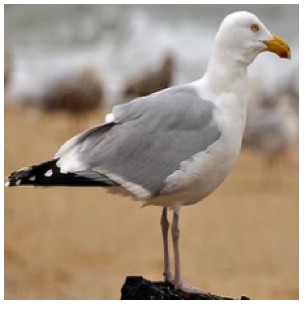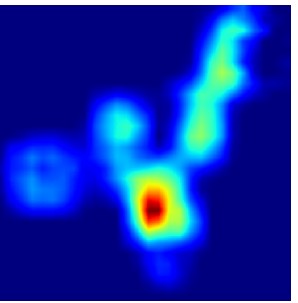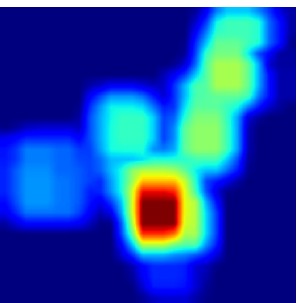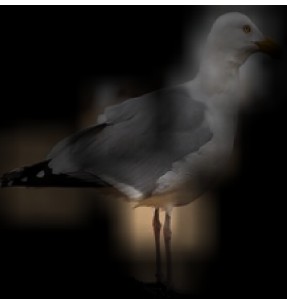

Figure 9: Visualization of Steps in Masking Pipeline to obtain DINOv3 feature representation for feature $d$ from left to right: Input image $\boldsymbol{X}$, Activation of Feature $d$, max pooled activation, and resulting masked image $(\boldsymbol{X} \odot \tilde{\boldsymbol{M}}_d)$ of which the CLS token is used.

Table 2: Comparison across datasets with Resnet50 for the prediction metrics in response to splitting up individual features: MonoQPM predicts smaller sets (Lower Fuzziness) that reach the same target coverage using APS and THR, while not worsening Negative-Log-Likelihood (NLL) or Average Calibration Error (ACE). The best value among interpretable models is highlighted bold, underlined are the ones that beat the baseline QPM.

| Dataset | Method | $\frac{n_c}{n_f^*}$ | THR Fuzz. ↓ | APS Fuzz. ↓ | THR ACE ↓ | APS ACE ↓ | NLL ↓ |
|---|---|---|---|---|---|---|---|
| **CUB-2011** | Dense Resnet50 Pre-FT | 1 | 0.65 | 2.05 | 0.65 | 0.72 | 0.53 |
| | QPM | 1 | 1.18 | 2.75 | **0.65** | 0.57 | **0.69** |
| | 1-Split Baseline | 2 | 1.13 | 2.00 | 0.66 | 0.63 | 0.72 |
| | 2-Split Baseline | 3 | 1.25 | 1.98 | **0.65** | **0.56** | 0.78 |
| | MonoQPM (Ours) | 2.58 | **1.03** | **1.97** | 0.69 | 0.68 | **0.69** |
| **StanfordCars** | Dense Resnet50 Pre-FT | 1 | 0.24 | 1.48 | 0.54 | 0.62 | 0.30 |
| | QPM | 1 | 0.75 | 1.92 | 0.71 | 0.97 | **0.40** |
| | 1-Split Baseline | 2 | 0.71 | 1.39 | 0.76 | **0.48** | 0.41 |
| | 2-Split Baseline | 3 | **0.69** | **1.32** | 0.77 | 1.01 | 0.42 |
| | MonoQPM (Ours) | 2.38 | **0.69** | 1.47 | **0.67** | 0.83 | **0.40** |
| **CUBCars** | Dense Resnet50 Pre-FT | 1 | 0.50 | 3.05 | 0.68 | 0.72 | 0.43 |
| | QPM | 1 | 1.31 | 3.82 | 0.69 | 0.67 | 0.55 |
| | 1-Split Baseline | 2 | **1.15** | 2.71 | 0.69 | **0.51** | **0.54** |
| | 2-Split Baseline | 3 | 1.21 | **2.68** | **0.68** | 0.55 | 0.56 |
| | MonoQPM (Ours) | 2.64 | 1.18 | 2.69 | 0.73 | 0.61 | **0.54** |
| **ImageNet** | Dense Resnet50 Pre-FT | 1 | 2.15 | 5.49 | 0.52 | 0.26 | 0.96 |
| | QPM | 1 | 9.73 | 20.46 | 0.34 | 0.34 | **1.35** |
| | 1-Split Baseline | 2 | 10.2 | 17.03 | 0.33 | **0.23** | 1.39 |
| | 2-Split Baseline | 3 | 11.28 | 17.0 | **0.31** | 0.36 | 1.46 |
| | MonoQPM (Ours) | 4.93 | **9.33** | **16.45** | 0.32 | 0.29 | 1.36 |

minimum between 30 and $\frac{|\mathcal{Y}^d|}{2}$ as upper limit for the number of components. This limit is never chosen by MonoQPM in our experiments.

## 6.2 Metrics

This section describes the metrics used in our experiments. As elaborated in Section 4.3, our main aspect to compare models and their disentanglement is the utility as conformal predictor captured by Observed Fuzziness (eq. (4)) for both APS and THR CP (eq. (3)). We additionally measure the Average Calibration Error (ACE) across the test dataset $\mathcal{D}_{\text{test}}$ to quantify whether the disentanglement maintains the coverage

guarantees:

$$\text{ACE} = \frac{1}{|\mathcal{D}_{\text{test}}|} \sum_{i=1}^{|\mathcal{D}_{\text{test}}|} \left| p_{(i)} - \frac{i-1}{|\mathcal{D}_{\text{test}}| - 1} \right| \quad \text{with} \quad p_{(1)} \leq p_{(2)} \leq \cdots \leq p_{(|\mathcal{D}_{\text{test}}|)} \tag{18}$$

Here, $p_{(i)}$ refers to the sorted p-values (eq. (4)) of ground truth classes and a uniform distribution of these ensures marginal validity (Vovk et al., 2005), i.e. $p(i) = \frac{i-1}{|\mathcal{D}_{\text{test}}| - 1}$.

Additionally, we quantify the coherence of the resulting concepts of our MonoQPM with Concept Classes Similarity:

$$\text{Concept Classes Similarity} = \frac{1}{n_c} \sum_c \left( \frac{1}{|\mathcal{P}^c|} \sum_{(u,v) \in \mathcal{P}^c} \psi_{u,v}^{d(c)} \right) \quad \text{with} \quad \mathcal{P}^c = \{(u,v) \mid u,v \in \mathcal{Y}^c, u < v\} \tag{19}$$

The similarity matrix $\Psi^{d(c)} \in \mathbb{R}^{n_y \times n_y}$ denotes how similar the feature of concept $c$ is for the samples of the respective classes. Concept Classes Similarity quantifies the average similarity of classes assigned to the same concept. One intuitive measure for $\Psi^d$ is the use of the representation space of a foundation model, similar to prior work (Dreyer et al., 2024). Specifically, we use DINOv3-ViT-L (Siméoni et al., 2025) to compute $\Psi^{d,\text{DINO}}$:

$$\psi_{u,v}^{d,\text{DINO}} = \cos\left(\text{mean}_{\boldsymbol{X} \in \mathcal{X}^v} \boldsymbol{h}(\boldsymbol{X}, d), \text{mean}_{\boldsymbol{X} \in \mathcal{X}^u} \boldsymbol{h}(\boldsymbol{X}, d)\right) \quad \text{with} \quad \boldsymbol{h}(\boldsymbol{X}, d) = \text{CLS}(\boldsymbol{X} \odot \tilde{\boldsymbol{M}}_d) \tag{20}$$

Here, $\boldsymbol{h}(\boldsymbol{X}, d)$ refers to the CLS-token of the foundation model applied to the input image $\boldsymbol{X}$ that is masked according to the activation of feature $d$. Specifically, we apply max pooling with a 3x3 kernel to the feature maps before interpolation and multiplication to ensure that a sufficiently large image section is used. Hence, $\psi_{u,v}^{d,\text{DINO}}$ describes the cosine similarity of the average embedding obtained for the feature $d$ across all test samples $\mathcal{X}^u$, or $\mathcal{X}^v$, that belong to the respective classes $u$ and $v$. This pipeline is visualized in Fig. 9. We take the CLS token of this masked image, as the ablation in Table 5 shows that only this masking captures the feature sufficiently precise.

Utilizing the annotations in CUB-2011 and the prior information obtained by creating CUBCars, we also compute the Ground Truth Concept Classes Similarity. CUB-2011 contains annotations $\boldsymbol{a}_y \in [0,1]^{312}$ for every class $y$, indicating the fraction of images with label $y$, in which a human perceives the attribute $j$ to be present. Following Norrenbrock et al. (2025b), we compute the ground truth structural class similarity $\boldsymbol{\Psi}^{\text{gt}} \in [0,1]^{n_y \times n_y}$ with $\psi_{c,c'}^{\text{gt}}$ being the cosine similarity between $\boldsymbol{a}_c$ and $\boldsymbol{a}_{c'}$. For CUBCars, we directly use the dataset of origin for class similarity:

$$\Psi^{\text{gt}}(u,v) = \begin{cases} 1 & \text{if } (u,v) \in \text{CUB-2011} \vee (u,v) \in \text{Stanford Cars} \\ 0 & \text{otherwise} \end{cases}$$

Note that $\boldsymbol{\Psi}^{\text{gt}}$ does not depend on a feature. This simplifies eq. (19), as we can substitute $\Psi^{d(c),\text{gt}} = \Psi^{\text{gt}}$ for all $c$. Combining $\boldsymbol{\Psi}^{\text{gt}}$, $\boldsymbol{\Psi}^{Dino}$ and eq. (19), we thus report the Ground Truth Concept Classes Similarity quantifying if similar classes are grouped to the same concept, and Dino Concept Classes Similarity. Dino Concept Classes Similarity quantifies how coherent concepts are to DinoV3-large (Siméoni et al., 2025).

## 6.3 Main Results

This section presents the main results of this paper. Table 2 shows the average number of concepts $n_c$ per feature, the fuzzinesses, calibration errors and NLL for ResNet-50 QPMs across datasets. Regarding the main evaluation of CP, the proposed MonoQPM reaches a lower Fuzziness in both THR and APS, without lowering the calibration accuracy. Additionally, already the hierarchical clustering baseline improves the frozen QPM in both metrics, showcasing the evident polysemanticity. However, the baseline generally increases NLL, while MonoQPM does not. The unaffected NLL is also reflected by even slightly improved TopN-performance, shown in Table 11. Notably, MonoQPM even outperforms the dense model before turning it into a QPM on APS Fuzziness on all fine-grained image classification datasets. Thus, the resulting interpretable MonoQPM is a more efficient conformal predictor with the guarantees of APS CP, than the black-box model. Fig. 1c visualizes how reduced fuzziness translates to smaller set sizes for relevant coverages. Fig. 10 shows the

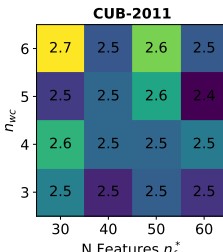 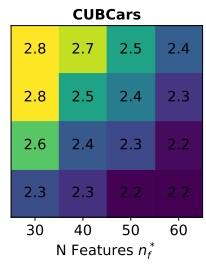 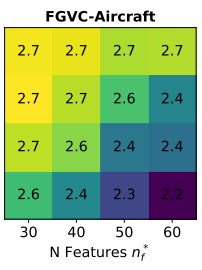 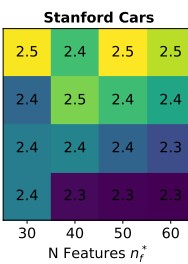

Figure 10: Average number of concepts per feature $\frac{n_c}{n_f^*}$ of MonoQPM applied to Resnet50 QPMs for different datasets with varying number of features $(n_f^*)$ and features per class $(n_{wc})$. The color code is based on unrounded values.

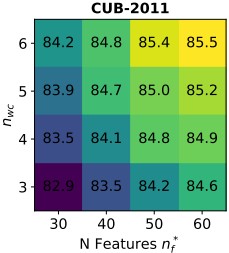 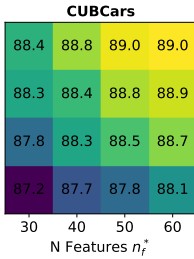 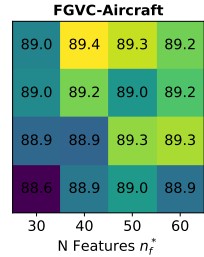 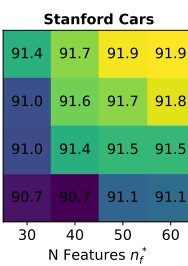

Figure 11: Classification Accuracy of the Resnet50 QPMs for different datasets with varying number of features $(n_f^*)$ and features per class $(n_{wc})$. The color code is based on unrounded values.

number of concepts our MonoQPM identifies when varying the number of classes assigned to each feature $\mathcal{Y}_d$ via the sparsity $n_{wc}$ and dimensionality $n_f^*$ of the class representations. FGVC-Aircraft is included as a dataset with fewer classes. All configurations lead to a QPM with good accuracy, as shown in Fig. 11. Our proposed method identifies on average more than 2 meanings per feature for all configurations, and the disentanglement of them consistently leads to improved predictive performance, shown in Table 3. Hence, it seems like some degree of polysemanticity is always an efficient solution to the training problem. Additionally, Fig. 10 showcases that MonoQPM generally identifies a higher degree of polysemanticity when every feature is assigned to more classes, in the top left of the heatmap, as one would intuitively expect. However, the exact number of concepts $n_c$ and gradient with respect to $|\mathcal{Y}_d|$ are clearly dataset dependent. This adaptiveness may also explain why only MonoQPM and not the simple baseline can effectively improve both CP metrics in most compactness configurations. The full prediction metrics for all configurations are shown in Figs. 20 to 22.

Table 3: Comparison of mean THR and APS Fuzziness as percent of baseline QPM across all the compactness configurations shown in Fig. 10. Values in parentheses denote the percentage of grid configurations achieving an improvement (Fuzziness < 100% of QPM). Most consistent (highest) improvement percentages are highlighted in bold.

| Metric | Method | CUB-2011 | CUBCars | FGVC-Aircraft | Stanford Cars |
|---|---|---|---|---|---|
| THR Fuzziness ↓ | 1-Split Baseline | 104.3 (31%) | 89.8 (94%) | 105.1 (6%) | 92.9 (88%) |
| | 2-Split Baseline | 112.1 (12%) | 95.6 (62%) | 106.7 (19%) | 95.4 (62%) |
| | MonoQPM (Ours) | 92.4 (**88%**) | 89.1 (**100%**) | 96.0 (**94%**) | 93.3 (**94%**) |
| APS Fuzziness ↓ | 1-Split Baseline | 80.3 (**100%**) | 75.9 (**100%**) | 79.8 (**100%**) | 75.8 (**100%**) |
| | 2-Split Baseline | 80.5 (88%) | 75.8 (**100%**) | 75.7 (**100%**) | 72.5 (**100%**) |
| | MonoQPM (Ours) | 77.7 (**100%**) | 74.3 (**100%**) | 76.8 (**100%**) | 80.7 (**100%**) |

Table 4: Comparison of predictive performance and Concept Classes Similarity on the CUB-2011 (CUB) and CUBCars (CUCA) datasets across different architectures. The table compares the baseline QPM against the hierarchical clustering baseline with 1 or 2 Splits, and the proposed MonoQPM. Best values are highlighted bold, underlined are the ones that beat the baseline QPM.

| Architecture | Method | Predictive Performance | | | | | | Concept Classes Similarity | | | |
|---|---|---|---|---|---|---|---|---|---|---|---|
| | | THR Fuzz. ↓ | | APS Fuzz. ↓ | | NLL ↓ | | GT ↑ | | DINO ↑ | |
| | | CUB | CUCA | CUB | CUCA | CUB | CUCA | CUB | CUCA | CUB | CUCA |
| **ResNet-50** | QPM (No Split) | 1.18 | 1.31 | 2.75 | 3.82 | **0.69** | 0.55 | 0.64 | 0.62 | 0.28 | 0.18 |
| | 1-Split Baseline | 1.13 | **1.15** | 2.00 | 2.71 | 0.72 | 0.54 | 0.70 | 0.93 | 0.39 | 0.27 |
| | 2-Split Baseline | 1.25 | 1.21 | 1.98 | **2.68** | 0.78 | 0.56 | **0.73** | 0.96 | **0.45** | **0.32** |
| | MonoQPM (Ours) | **1.03** | 1.18 | **1.97** | 2.69 | **0.69** | 0.54 | 0.70 | **1.00** | 0.39 | 0.26 |
| **ResNet-34** | QPM (No Split) | 1.50 | 2.04 | 3.06 | 5.16 | **0.79** | **0.67** | 0.63 | 0.57 | 0.26 | 0.16 |
| | 1-Split Baseline | 1.49 | 1.96 | 2.28 | 4.05 | 0.84 | **0.67** | 0.70 | 0.80 | 0.39 | 0.24 |
| | 2-Split Baseline | 1.55 | 1.93 | **2.16** | 3.83 | 0.90 | 0.68 | **0.73** | 0.84 | **0.45** | **0.28** |
| | MonoQPM (Ours) | **1.34** | **1.87** | 2.25 | **3.53** | 0.80 | **0.67** | 0.69 | **1.00** | 0.35 | 0.24 |
| **Inception-v3** | QPM (No Split) | 1.97 | 3.00 | 3.81 | 4.79 | **0.85** | **0.72** | 0.62 | 0.56 | 0.19 | 0.12 |
| | 1-Split Baseline | **1.87** | 2.94 | 3.30 | 4.20 | 0.88 | 0.73 | 0.70 | 0.85 | 0.34 | 0.22 |
| | 2-Split Baseline | **1.87** | 2.99 | **3.27** | 4.22 | 0.91 | 0.75 | 0.72 | 0.91 | **0.40** | **0.28** |
| | MonoQPM (Ours) | 1.88 | **2.87** | 3.28 | **3.94** | 0.89 | **0.72** | 0.69 | **1.00** | 0.33 | 0.20 |

Table 5: Ablation study evaluating different input spaces and clustering techniques for splitting polysemantic features.

| Method | *Dataset: CUB-2011* | | | *Dataset: StanfordCars* | | | *Dataset: CUBCars* | | |
|---|---|---|---|---|---|---|---|---|---|
| | $\frac{n_c}{n_f^*}$ | THR Fuzz. ↓ | APS Fuzz. ↓ | $\frac{n_c}{n_f^*}$ | THR Fuzz. ↓ | APS Fuzz. ↓ | $\frac{n_c}{n_f^*}$ | THR Fuzz. ↓ | APS Fuzz. ↓ |
| *Clustering Variations:* | | | | | | | | | |
| $k$-Means | 2.36 | 1.07 | 1.99 | 2.13 | **0.69** | 1.48 | 2.10 | 1.17 | 2.72 |
| GMM | 2.54 | 1.06 | 2.02 | 2.35 | **0.69** | 1.51 | 2.69 | 1.17 | 2.71 |
| MonoQPM (Ours) | 2.58 | **1.03** | 1.97 | 2.38 | **0.69** | 1.47 | 2.64 | 1.18 | 2.69 |
| *Input Space Variations:* | | | | | | | | | |
| DINO$_\text{w-Tok}$ | 2.73 | 2.56 | 3.52 | 1.37 | 0.75 | 1.86 | 3.08 | 1.31 | 2.92 |
| DINO$_\text{CLS}$ | 6.93 | 2.26 | 3.39 | 8.07 | 0.75 | 1.90 | 14.56 | 1.64 | 3.70 |
| DINO$_\text{M-CLS}$ | 4.18 | 1.10 | 2.02 | 5.08 | 0.72 | 1.35 | 4.33 | **1.15** | **2.62** |
| DINO$_\text{P-M-CLS}$ | 4.26 | 1.09 | 1.98 | 5.50 | 0.71 | 1.35 | 4.82 | **1.15** | 2.64 |
| Activation | 2.76 | 1.04 | **1.94** | 2.62 | 0.72 | 1.43 | 2.62 | 1.17 | 2.72 |

After establishing that polysemanticity exists in ResNet-50, Table 4 shows the comparison for the CNNs with ReLU that QPM has been applied to. Evidently, MonoQPM effectively recognizes and disentangles polysemanticity across architectures, leading to improved efficiency as conformal predictor. Additionally, the Concept Classes Similarity quantifies that the resulting concepts are more coherent according to DINO and reflect the ground truth more accurately, with a 100% accuracy at disentangling the datasets in CUBCars without this supervision. More results with other architectures and visualizations of split features are included in the appendix.

## 6.4 Ablation Studies

This section contains the ablation studies, validating the design choices. We first establish that our proposed processing pipeline from the ReLU signs (eq. (15)) to the monosemantic assignment matrix $\boldsymbol{W}^{\text{mono}}$ outperforms using a fixed number of layers in the network (Fig. 12a) or variance in the PCA (Fig. 12b). Fig. 12a visualizes how our proposed automatic choice of $D$ generally outperforms any fixed choice for THR, while being only worse in APS in one configuration. Interestingly, this sweep also visualizes that including the final 30% of the model leads to an increased number of concepts and reduced performance, as theory suggests (Feng et al., 2022), validating the choice of a limited and automatically chosen $D$. Fig. 12b shows the metrics when varying $n_{\text{PCA}}$ on inputs where $l$ is chosen so that 55% of the activations are included, which simplifies the

Table 6: Performance evaluation of different splitting methods for Swin-tiny on the CUB-2011 dataset. Best values are highlighted bold, and values outperforming the QPM are underlined.

| Method | THR Fuzz. ↓ | APS Fuzz. ↓ | NLL ↓ | GT ↑ | DINO ↑ |
|---|---|---|---|---|---|
| QPM | **1.33** | 1.56 | **0.81** | 0.64 | 0.23 |
| 1-Split Baseline | 1.47 | 1.44 | 0.89 | 0.71 | 0.36 |
| 2-Split Baseline | 1.63 | 1.48 | 0.96 | **0.74** | 0.42 |
| DINO$_{\text{P-M-CLS}}$ | 1.58 | 1.50 | 0.92 | 0.72 | **0.43** |
| Activation | 1.42 | 1.42 | 0.87 | 0.71 | 0.36 |
| MonoQPM (Ours) | 1.36 | **1.38** | 0.87 | 0.71 | 0.36 |

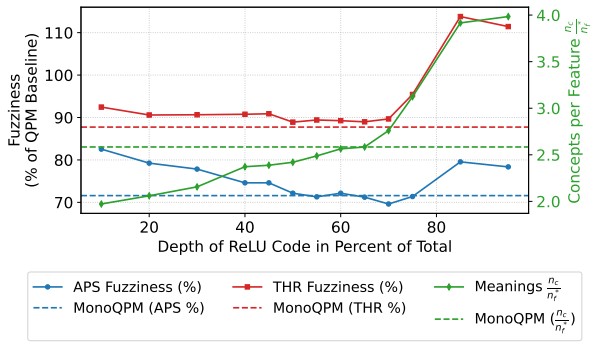

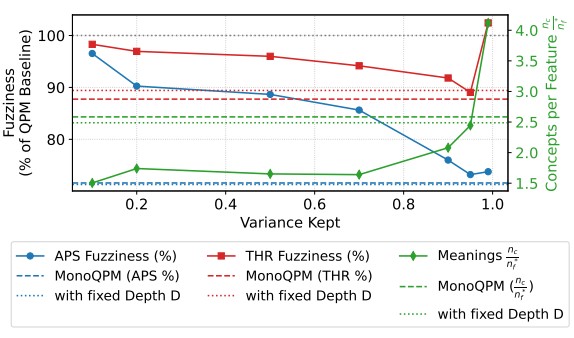

(a) Impact of Varying a Fixed Depth $D$

(b) Impact of Variance Kept with Fixed Depth of 55%

Figure 12: Impact of how ReLU code is compressed before LDA for Resnet50 MonoQPM on CUB-2011. The left y-axis denotes Fuzziness as percent of baseline QPM.

ablation, and leads to a similar conclusion: Maintaining more information about the feature is generally preferential until the number of discovered concepts $n_c$ starts sharply decreasing. At this point, the utility decreases. Additionally, using the participation ratio (eq. (12)) is clearly superior to any fixed variance, with the dynamically chosen $D$ further boosting performance. Fig. 13 visualizes the impact of the dimensionality of the final representations to cluster with similar results for CUB-2011 and Stanford Cars. Just considering the CP efficiency, it shows a local minimum when using more than one component. However, doing so causes a significant increase in both the number of discovered concepts and, crucially, the NLL. Hence, only using one component improves CP efficiency while staying neutral to the training goals, effectively only alleviating the interference induced by polysemanticity. However, future work might investigate if certain features require more components. Finally, when comparing the results at 1 component, using the optimally initialized GMM is superior. This is also shown in Table 5, which contains the results for different categorical alternatives to our proposed MonoQPM. Its optimally initialized GMM clustering slightly outperforms $k$-Means and classically initialized GMMs. Additionally, we compare different input options to our MonoQPM pipeline. Specifically, we use *Activation*, with $\mathbf{r}_d^{(l)} = \text{mean}_{m,n} \mathbf{A}_{d,m,n}^{(l)}$ as heavily related alternative and different options using the features of DINOv3-ViT-L. *Activation* shows fairly similar results to our proposed sign of the activations, but validates that only using the Spline-Theory inspired approach is not discarding necessary information. DINO$_{\text{w-Tok}}$ uses the patch embeddings obtained on the entire image scaled with the interpolated feature map of the respective feature, while DINO$_{\text{CLS}}$ simply uses the CLS token of the entire image. Both of these approaches show detrimental performance and are unable to consistently disentangle polysemantic features effectively. Therefore, we use the masked image approach for our metric in eq. (20). These are encoded by DINO$_{\text{M-CLS}}$ with the $P$ denoting the max pooling operation applied to the feature map and show generally strong performance, similar to MonoQPM. All strong performing approaches show very similar utility, which may indicate that they all split the same impactful and evident polysemantic features, but disagree on those with more negligible impact, resulting in the different absolute number of concepts. We additionally show

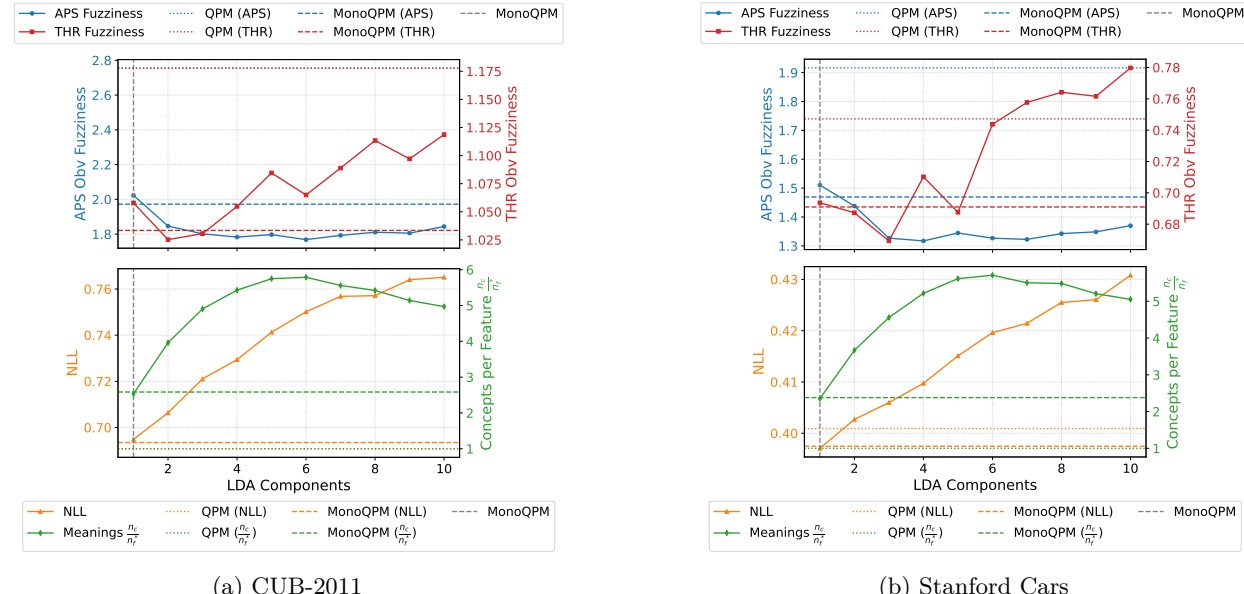

(a) CUB-2011

(b) Stanford Cars

Figure 13: Impact of Number LDA Components for Resnet50 MonoQPM without using optimally initialized GMM.

results on Swin-Transformer-tiny in Table 6. There, the DINO-based approach fails, as the heatmaps it relies on are not localizing well consistently, as shown in Fig. 49. In contrast, the internal activations used by MonoQPM are unaffected by the saliency maps. However, MonoQPM's performance is relatively weak, likely as Swin-Transformer does not use ReLU. In conclusion, our proposed pipeline is generally very robust and flexible. It can effectively make use of strong foundation models or other internal activations, but its straightforward principled approach generally comes very close to the optimal disentanglement found. Finally, appendices B and C justify the choice of BIC and show the impact of the assignment threshold $\tau$, while appendix D discusses the ressource requirements for applying MonoQPM.

## 7 Conclusion

In this paper, we introduced utility-focused measuring of polysemanticity in the inherently interpretable QPM which has no superposition on its final features. By evaluating whether a disentangled QPM shows higher utility as a conformal predictor, we are able to compare different approaches quantitatively without any assumptions. This allows the thorough evaluation of various methods, including a simple hierarchical clustering baseline that demonstrates that polysemanticity exists in QPM. Outperforming this baseline, our proposed Monosemantic QPM (MonoQPM) consistently improves the efficiency of a frozen QPM by reducing interference and represents classes using concepts rather than features, resulting in improved global interpretability. Furthermore, we show that MonoQPM effectively splits polysemantic features into monosemantic concepts using the proposed Concept Classes Similarity on the introduced CUBCars dataset with artificially created ground truth. Finally, we believe that this avenue can be used to evaluate further techniques for disentangling polysemantic representations and hence can contribute to building safe intelligent systems.

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

# A    Visualization of Feature Split

This section describes the way the explicit disentanglement of features is visualized, as e.g. shown in Figs. 14 and 17. Specifically, the goal is to visualize the entire split of one individual feature $d$, including all the classes it is assigned to $\mathcal{Y}^d$. For all concepts $c \in \mathcal{C}^d$, one exemplary training image is shown for each class $y$ that is assigned to the concept $c$. The images are chosen so that they ideally visualize how the concept is expressed for this class as determined by our method. Hence, we choose the sample that maximizes the probability of belonging to that concept $\boldsymbol{P} \in [0,1]^{n_s^d \times n_c^d}$ (Section 5.2). The images are ordered by the index of the primary concept their class is assigned to, and in each concept $c$ classes are sorted by the average probability of the specific class $y$ belonging to that concept $p_c^y$, obtained via indexing $\boldsymbol{p}^y \in [0,1]^{n_c^d}$ (Section 5.2).

The assignment to individual concepts is highlighted primarily via color: The border of every image encodes the concepts this class is assigned to. For example in Fig. 14, the purple border of *Glaucous winged Gull* indicates that this class is exclusively assigned to the purple concept, like all the other terns assigned to this feature. In contrast, the sparrows show a yellow border, highlighting that they are exclusively assigned to the second concept. Finally, *Parakeet Auklet* appears twice with a two-colored border, indicating that it belongs to both discovered concepts. In order to improve readability without color, we additionally include this information in the respective image caption.

The first row of each caption encodes the activation of the feature for this image, followed by the zero-based index of the concept this image is chosen for separated by ␣. The chosen concept is also indicated by boldness in the row below. Finally, the row concludes with the class name.
The second row contains the probabilities $\boldsymbol{P}$. The first $|\mathcal{C}^d|$ entries denote the average probabilities of the classes belonging to each concept $\boldsymbol{p}^y$, with the final figure denoting the probability of the chosen sample.

We indicate the feature activation via a red square that highlights the patch of maximum activation in the middle of it. This technique is chosen as it allows for a simple overview while maintaing color and the feature maps of QPM are generally very localized, as for example shown in (Fig. 15).

Additionally, we include the one-dimensional representations $\mathbf{r}^d \in \mathbb{R}^{n_s^d}$ (Section 5.1) of internal activations below the images. For these, every individual sample is shown as a dot and each class is summarized using a box from the first to third quantile with the median highlighted. The color of each dot is the average of the colors of the assigned concepts weighted by class probabilities $\boldsymbol{p}^y \in [0,1]^{n_c^d}$ (Section 5.2). Therefore in Fig. 14, the *Parakeet Auklet* dots are 66% purple of concept 1 and 34% yellow of concept 2.

# B    Impact of Information Criteria

This section discusses the impact of the chosen Bayesian Information Criterion in determining the number of concepts per feature. Table 7 shows the APS and THR Fuzziness, alongside the average number of concepts per feature across the four criteria BIC, AIC, HQIC and Elbow. In order to implement an Elbow criterion, we use the *kneed* Satopaa et al. (2011) library with default configuration on the likelihood computed by the GMM. Evidently, the proposed method is robust to the choice of criterion and performs similarly in Fuzziness scores across all four tested criteria. However, the methods differ in the returned number of concepts. Due to its penalty, BIC always returns fewer concepts than AIC or HQIC, while the heuristic-based Elbow method has more relative variance. This ablation justifies the choice of BIC as a principled approach to prove that polysemanticity exists and can be used with positive predictive impact.

Table 7: Comparison of THR Fuzziness, APS Fuzziness, and number of Meanings across selection criteria with Resnet50.

| Method | Dataset: CUB-2011 | | | Dataset: StanfordCars | | | Dataset: CUBCars | | |
|---|---|---|---|---|---|---|---|---|---|
| | THR Fuzziness ↓ | APS Fuzziness ↓ | $\frac{n_c}{n_f^z}$ | THR Fuzziness ↓ | APS Fuzziness ↓ | $\frac{n_c}{n_f^z}$ | THR Fuzziness ↓ | APS Fuzziness ↓ | $\frac{n_c}{n_f^z}$ |
| BIC (Ours) | 1.03 | 1.97 | 2.58 | 0.69 | 1.47 | 2.38 | 1.18 | 2.69 | 2.64 |
| AIC | 1.05 | 1.96 | 3.37 | 0.71 | 1.46 | 3.23 | 1.17 | 2.70 | 3.74 |
| HQIC | 1.03 | 1.98 | 2.85 | 0.71 | 1.46 | 2.72 | 1.17 | 2.73 | 3.05 |
| Elbow | 1.05 | 1.95 | 2.47 | 0.69 | 1.48 | 2.48 | 1.17 | 2.72 | 2.03 |

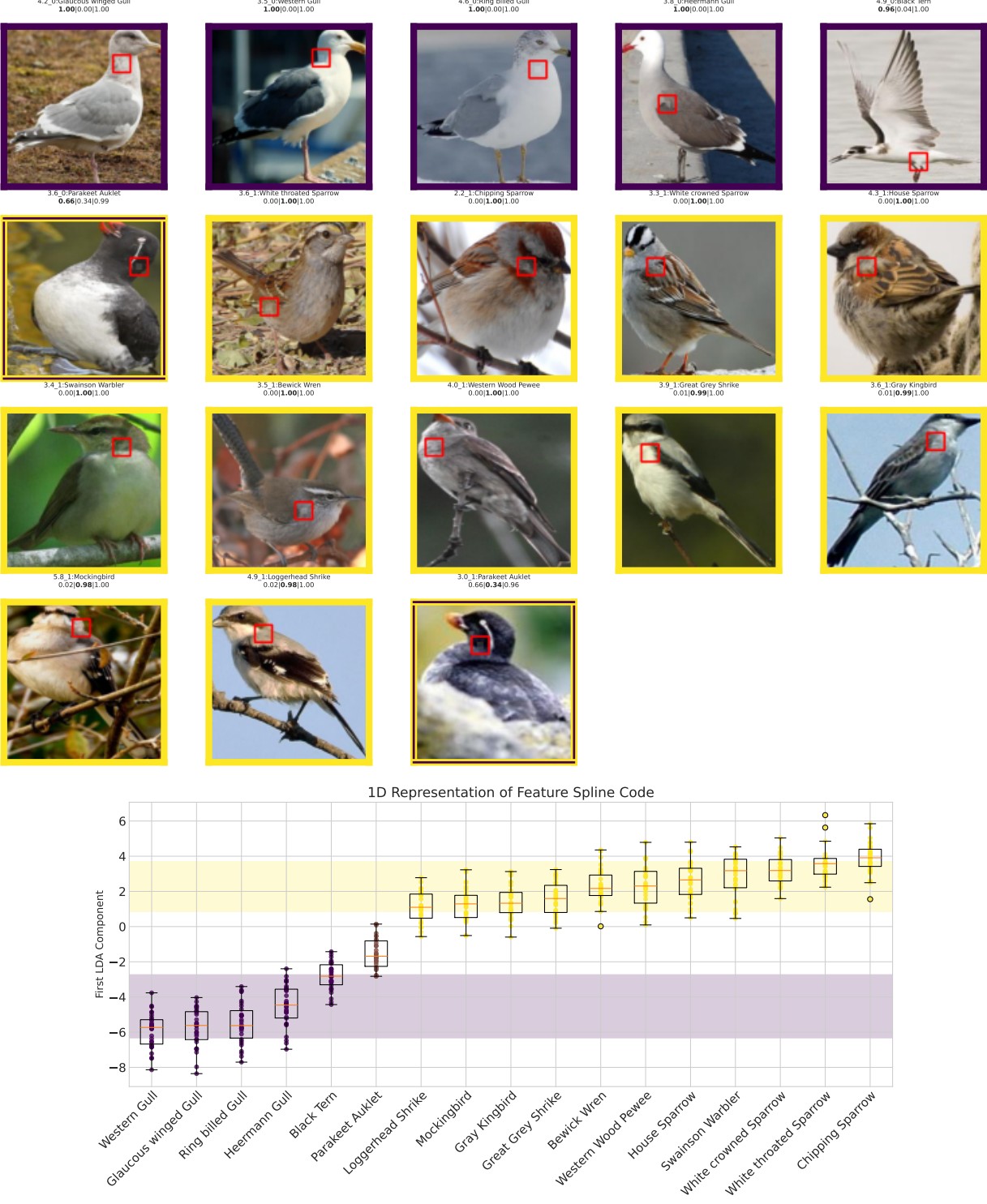

Figure 14: Exemplary Split Feature of QPM trained on CUB-2011. The top figure shows exemplary images and the bottom shows one-dimensional representation of internal activations, as described in appendix A.

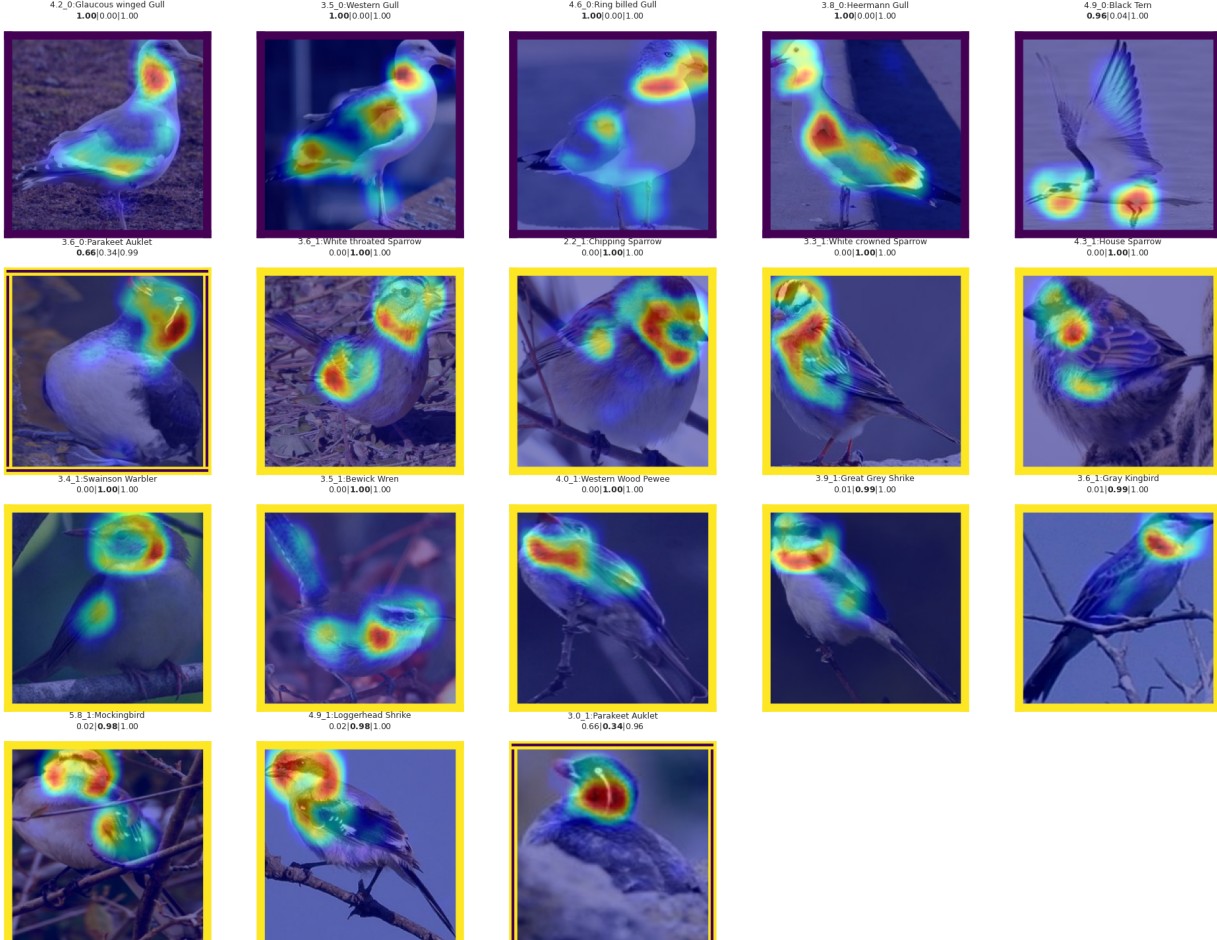

Figure 15: Visualization of the concepts and classes split based on the clustering shown in Fig. 14 using the entire activation map.

## C Impact of Assignment Threshold

This section discusses the influence of the assignment threshold used for assigning classes to individual split up concepts in Section 5.2. Fig. 16 shows the impact of changing the assignment threshold $\tau$ via the $\beta$ parameter in $\tau = \beta \cdot \frac{1}{n_c^d} = \beta \cdot \boldsymbol{p}_{\text{uniform}}$. The proposed MonoQPM clearly outperforms the frozen QPM in both APS and THR Fuzziness across the entire range of values, showcasing the robustness of the method to this hyperparameter. Additionally, the plots justify the used value of $\tau = \frac{1}{2 \cdot n_c^d}$, as it performs well across the three datasets, being near optimal for THR, while APS Fuzziness generally improves with more disentanglement.

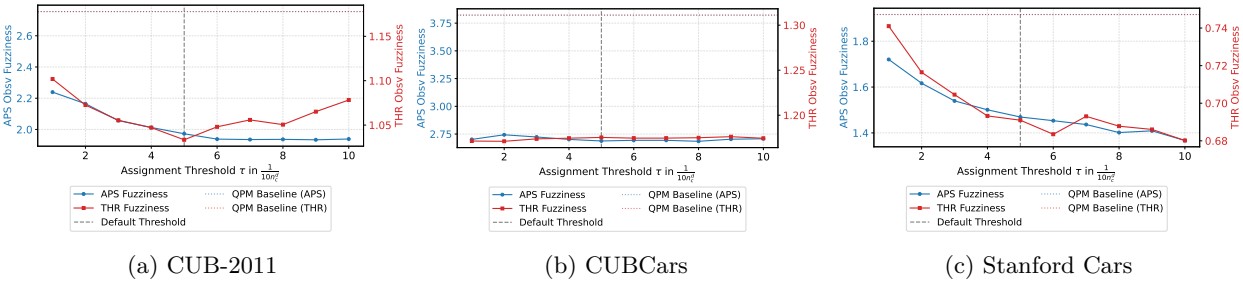

(a) CUB-2011  (b) CUBCars  (c) Stanford Cars

Figure 16: Impact of Assignment Threshold $\tau = \beta \cdot \frac{1}{n_c^d}$ on resulting APS and THR Observed Fuzziness scores with Resnet50.

## D Resource Requirements

Table 8 shows the resources it takes to turn a QPM into a MonoQPM using our implementation. Regarding time, it includes collecting the activations via one forward pass over the training dataset, applying the processing of Section 5.1 and Fig. 8b and the clustering and assignment of Section 5.2. Note that the optimal clustering using Jenks Natural Breaks is skipped on ImageNet-1K, as it takes prohibitively long. Additionally, we limit the sweep of the depth of the representation, as described in Fig. 8b, to 5000 samples due to VRAM constraints of 10GB.

While the three individual steps of the preprocessing do account for most of the time on CUBCars and CUB-2011, there is a significant gap on ImageNet. This is due to unoptimized code, that needlessly scales poorly with increasing number of samples. However, after obtaining the one-dimensional sample representations, the remaining time can be almost completely attributed to fitting the GMMs, with potentially optimally initialized clusters, and computing the Bayesian Information Criterion (BIC). Notably, explicitly constructing the new assignment matrix is negligible with less than 0.1 seconds. Overall, Table 8 shows that MonoQPM is very fast on small datasets like CUB-2011. While it takes almost 12 hours to compute the splitting on ImageNet-1K, it can still be readily applied there and the resulting MonoQPM can then be used without further increase in computation. Additionally, all the steps could be perfectly parallelized, as the proposed method operates on each feature independently.

Finally, the memory column shows that the current implementation requires a lot of memory on ImageNet. However, this could be significantly reduced as we currently hold all sample representations of all features in memory at once, since the method was tuned on the small datasets and only ran on ImageNet for 3 seeds.

## E Exemplary Feature Splits

This section introduces the additional visualizations Fig. 23 to Fig. 48 of split features in the appendix. Grouped by dataset and sorted by increasing number of splits they provide a qualitative impression of what is determined as individual concepts in polysemantic features. As discussed, rating these splits visually is very difficult, as humans simply do not know the concepts the model actually relies on. However, our utility-focused evaluation confirms that the disentanglements by MonoQPM are effective and suggests that analyzing these

Table 8: Resource requirements for applying MonoQPM to ResNet50 on different datasets, with increasing number of samples as shown in Table 1. Time is generally denoted in seconds and describes the total time spent on this task for one model.

| Dataset | Activation Collection | Swept Rank | PCA | LDA | Entire Preprocessing | Jenks | GMM | BIC | Total after Preprocessing | Total Time (Hours) | Max Memory (GB) |
|---|---|---|---|---|---|---|---|---|---|---|---|
| CUB-2011 | 31 | 66.66 | 23.80 | 1.61 | 96.10 | 1.00 | 3.99 | 0.29 | 5.41 | 0.04 | 11 |
| CUBCars | 100 | 215.78 | 44.89 | 3.49 | 273.47 | 15.79 | 15.99 | 0.85 | 33.01 | 0.11 | 15 |
| ImageNet | 2828 | 3490.66 | 3585.10 | 344.23 | 12852.46 | - | 25622.48 | 287.74 | 25913.70 | 11.56 | 300 |

concepts rather than the features is more appropriate. However, the examples clearly confirm the results obtained via Concept Classes Similarity. The features are frequently split into concepts that are assigned to highly similar classes, e.g. Gull and Sparrow (Fig. 25), white water bird and flycatcher-like (Figs. 26 and 27) or all the examples where cars and birds are separated in Figs. 17 and 34 to 42. Regarding DINOv3 Concept Classes Similarity, a general increase in visual coherence of concept activations is also evident. However, certain splits like Figs. 26 and 27 showcase how visual inspection is insufficient: The feature could both respond to the eye of all of the birds, or of specific properties of the subgroups, while the internal activations show two clear clusters. Figs. 43 to 48 additionally show examples from the other used datasets that allow similar conclusions. One interesting observation are concepts for FGVC-Aircraft that are only assigned to one class, as in Fig. 45. This might be due to lower semantic similarity between the classes in that dataset. For Swin-Transformers two visualizations with heatmaps are shown in Figs. 49 and 50. They showcase how the heatmaps are of highly varying quality. Finally, one visualization is shown that showcases a limitation of just reporting the number of total concepts $n_c$: Fig. 41 shows a feature for which 3 concepts are determined via BIC, but two concepts are assigned to identical classes. Hence, for inference this feature is effectively split just once, highlighting an avenue for future work.

## F  Future Work

This work introduces a new problem to optimize alongside an effective baseline and a strong initial solution. After having established that our utility-focused approach can be used to quantify polysemanticity and its disentanglement, future work can further improve our proposed method and the evaluation methodology. As discussed above, $n_c$ should likely be discussed jointly with the number of concept assignments per class. Additionally, the inference scheme might be relaxed to allow for more sophisticated ways of determining which concept is active, like classifying the internal activations. Table 5 shows that our proposed pipeline can make use of several inputs effectively. Hence, one can try combining foundation model representations and internal activations. Finally, more elaborate inputs based on Spline Theory should be explored.

For broader significance, it is worth considering how the proposed method and evaluation can be applied to other models than just end-to-end trained QPMs. While the proposed evaluation and inference is intrinsically tied to the classification setting, many of the currently relevant higher-stakes problems can be considered as such. Examples include next token prediction, segmentation and intent classification. This leads to potentially many applications in various domains and modalities. Firstly, the recent DINO-QPM (Zimmermann et al., 2026) shows that frozen visual foundation models can be adapted to operate as a QPM for inherently interpretable classification. Thus, MonoQPM may directly be applied to improve such then-interpretable adapted models, opening up applications in the various domains foundation models excel on. Notably, both DINO-QPM and MonoQPM can theoretically be applied to all sorts of domains, including language problems such as intent classification, which is a promising avenue for future work to safeguard powerful LLMs. Secondly, it is of interest to apply this work to individual neurons in other interpretable models, such as the latent embeddings of a Sparse Autoencoder, as they are designed to have representations with low superposition and aim for monosemantic neurons. This could be done by following MonoQPM to split up individual neurons and measure downstream predictive performance. It may guide the choice of optimal capacity for SAEs or help researchers find ways to more consistently obtain monosemantic neurons. Finally, even though individual neurons in dense models are generally not interpretable due to superposition, an improved disentanglement of the features should still transfer to improved CP efficiency. Thus, future work

Table 9: Comparison across datasets with Inception-v3 for the main metrics in response to splitting up individual features: Using the proposed method, smaller sets (Lower Fuzziness) can be predicted to reach the same target coverage using APS and THR, while not worsening point-predictive performance (NLL) or calibration error (Average Calibration Error). The best value among interpretable models is highlighted bold, all values underlined that beat the baseline QPM.

| Dataset | Method | $n_c$ | THR Fuzz. ↓ | APS Fuzz. ↓ | THR ACE ↓ | APS ACE ↓ | NLL ↓ |
|---|---|---|---|---|---|---|---|
| **CUB-2011** | Dense Inception-v3 Before FT | 1 | 0.69 | 2.04 | 0.76 | 0.72 | 0.56 |
| | QPM | 1 | 1.97 | 3.81 | 0.79 | **0.62** | **0.85** |
| | 1-Split Baseline | 2 | **1.87** | 3.30 | **0.74** | 0.75 | 0.88 |
| | 2-Split Baseline | 3 | **1.87** | **3.27** | 0.81 | 1.07 | 0.91 |
| | **Proposed Method** | 3.58 | 1.88 | 3.28 | 0.85 | 0.77 | 0.89 |
| **StanfordCars** | Dense Inception-v3 Before FT | 1 | 0.42 | 1.52 | 0.61 | 0.67 | 0.31 |
| | QPM | 1 | **1.50** | 2.37 | 0.66 | 0.84 | **0.50** |
| | 1-Split Baseline | 2 | 1.52 | 2.13 | **0.60** | 0.74 | 0.51 |
| | 2-Split Baseline | 3 | 1.51 | 2.08 | 0.68 | **0.68** | 0.52 |
| | **Proposed Method** | 4.05 | 1.53 | **2.07** | 0.68 | **0.68** | 0.52 |
| **CUBCars** | Dense Inception-v3 Before FT | 1 | 0.64 | 2.96 | 0.71 | 0.62 | 0.43 |
| | QPM | 1 | 3.00 | 4.79 | 0.67 | 0.55 | **0.72** |
| | 1-Split Baseline | 2 | 2.94 | 4.20 | **0.65** | 0.55 | 0.73 |
| | 2-Split Baseline | 3 | 2.99 | 4.22 | 0.67 | **0.39** | 0.75 |
| | **Proposed Method** | 2.48 | **2.87** | **3.94** | 0.67 | 0.49 | **0.72** |

Table 10: Comparison across datasets with Renset34 for the main metrics in response to splitting up individual features: Using the proposed method, smaller sets (Lower Fuzziness) can be predicted to reach the same target coverage using APS and THR, while not worsening point-predictive performance (NLL) or calibration error (Average Calibration Error). The best value among interpretable models is highlighted bold, all values underlined that beat the baseline QPM.

| Dataset | Method | $n_c$ | THR Fuzz. ↓ | APS Fuzz. ↓ | THR ACE ↓ | APS ACE ↓ | NLL ↓ |
|---|---|---|---|---|---|---|---|
| **CUB-2011** | Dense Renset34 Before FT | 1 | 0.66 | 2.17 | 0.71 | 0.91 | 0.59 |
| | QPM | 1 | 1.50 | 3.06 | 0.68 | 0.82 | **0.79** |
| | 1-Split Baseline | 2 | 1.49 | 2.28 | 0.67 | **0.74** | 0.84 |
| | 2-Split Baseline | 3 | 1.55 | **2.16** | 0.76 | 0.97 | 0.90 |
| | **Proposed Method** | 2.41 | **1.34** | 2.25 | **0.66** | 0.81 | 0.80 |
| **StanfordCars** | Dense Renset34 Before FT | 1 | 0.29 | 2.35 | 0.68 | 0.78 | 0.35 |
| | QPM | 1 | 1.15 | 2.57 | 0.56 | **0.53** | **0.48** |
| | 1-Split Baseline | 2 | 1.15 | 2.55 | 0.56 | 0.77 | **0.48** |
| | 2-Split Baseline | 3 | 1.15 | 2.55 | 0.56 | 0.67 | **0.48** |
| | **Proposed Method** | 2.07 | **1.08** | **1.91** | **0.54** | 0.73 | **0.48** |
| **CUBCars** | Dense Renset34 Before FT | 1 | 0.55 | 4.27 | 0.81 | 0.71 | 0.49 |
| | QPM | 1 | 2.04 | 5.16 | 0.62 | **0.59** | **0.67** |
| | 1-Split Baseline | 2 | 1.96 | 4.05 | **0.51** | 0.66 | **0.67** |
| | 2-Split Baseline | 3 | 1.93 | 3.83 | 0.54 | 0.84 | 0.68 |
| | **Proposed Method** | 2.74 | **1.87** | **3.53** | 0.57 | 0.61 | **0.67** |

may use the utility-focused evaluation of the disentanglement of polysemantic neurons to help develop dense models with less polysemanticity.

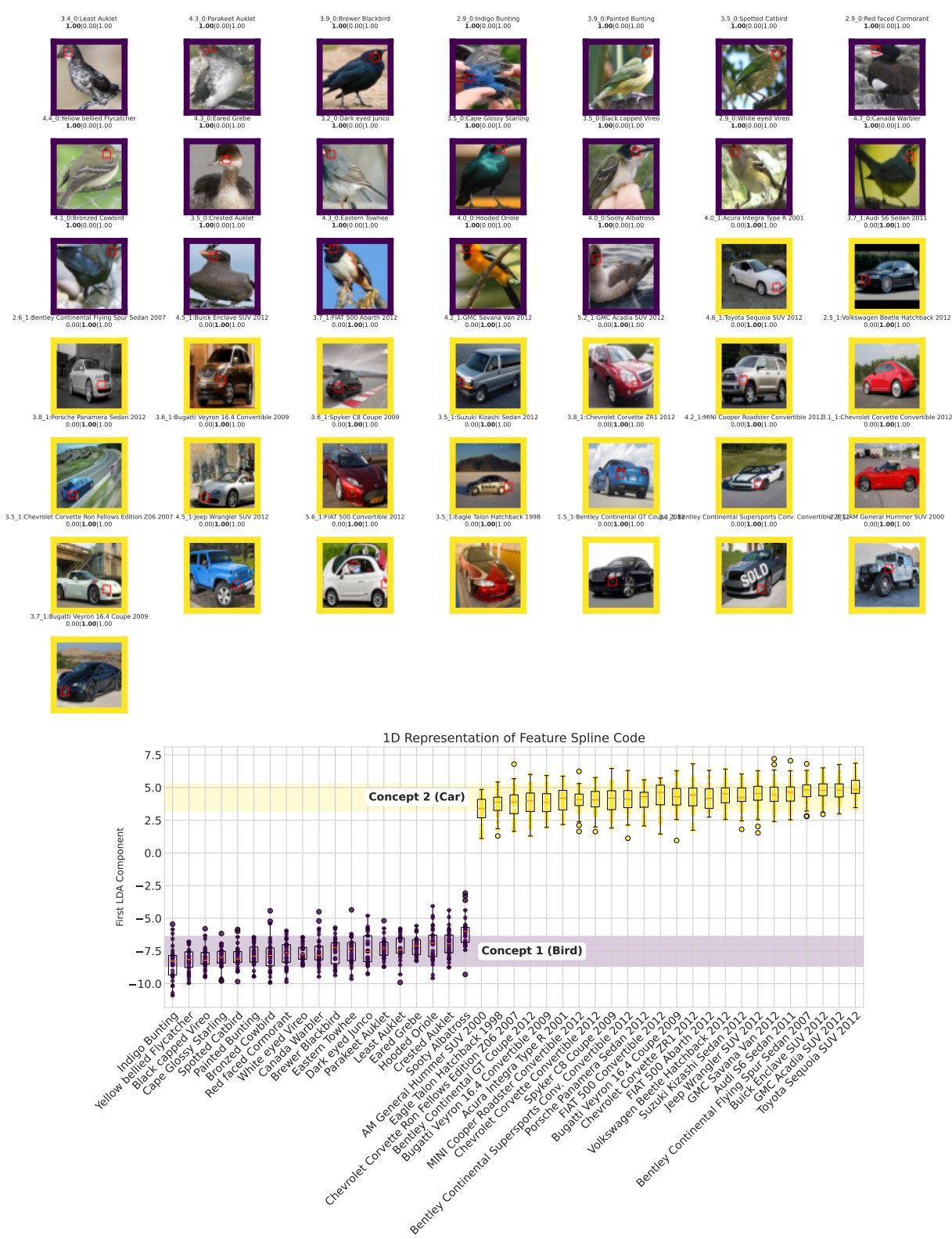

Figure 17: Exemplary visualized disentangled Feature of QPM trained on CUBCars, shown in Fig. 1a. The top figure shows exemplary images and the bottom shows one-dimensional representation of internal activations, as described in appendix A.

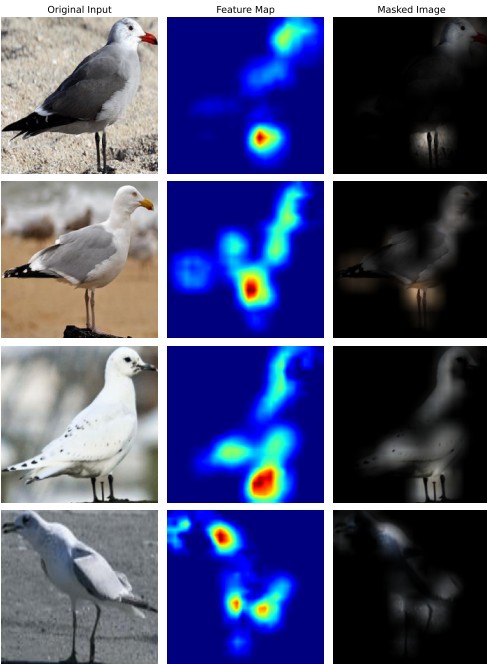

Figure 18: Exemplary Visualization of the Masking process used to evaluate Dino-Sim without dilated feature activation.

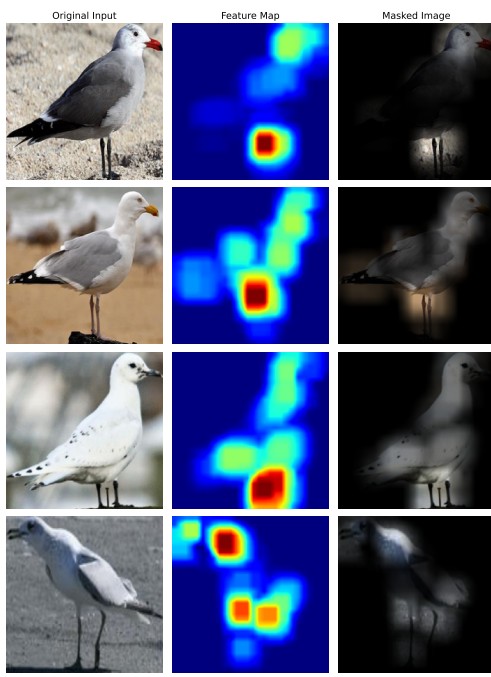

Figure 19: Exemplary Visualization of the Masking process used to evaluate Dino-Sim with dilated feature activation.

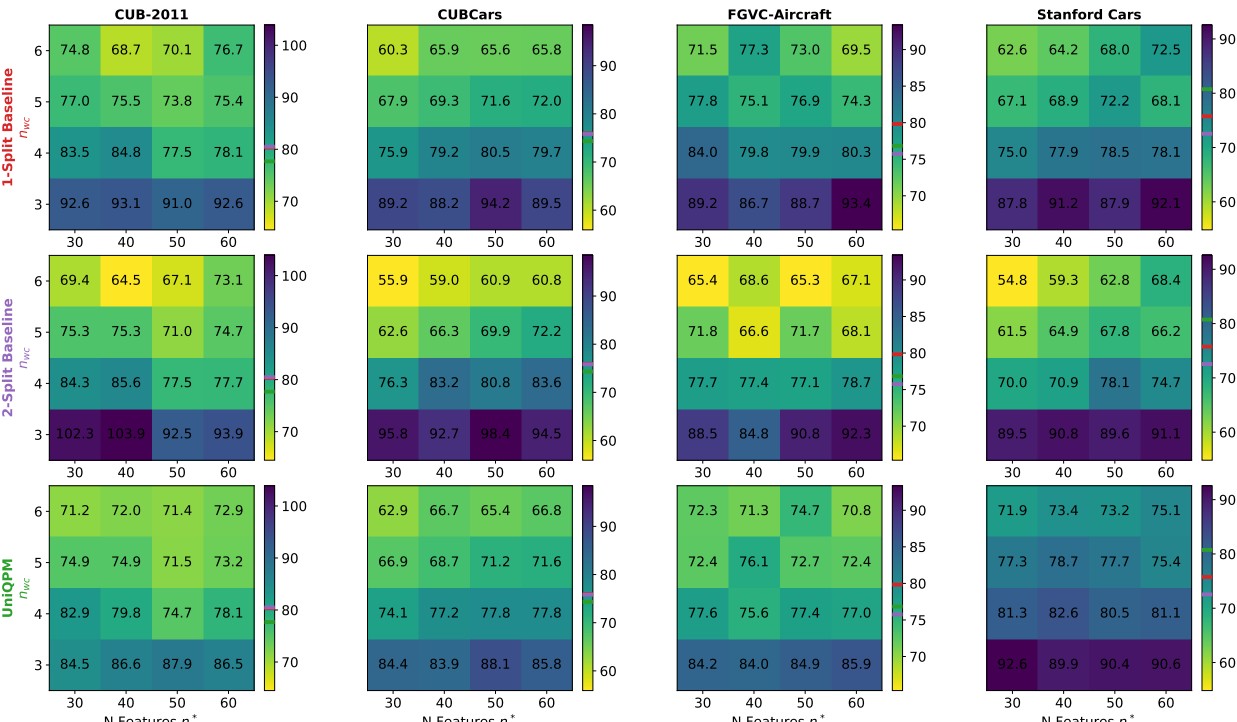

Figure 20: Scaled APS Fuzziness as percent of base QPM of MonoQPM applied to Resnet50 QPMs for different datasets with varying number of features $(n_f^*)$ and features per class $(n_{wc})$. The color code is based on unrounded values. The colorbar is shared across methods for one dataset and the mean of each method is indicated via color.

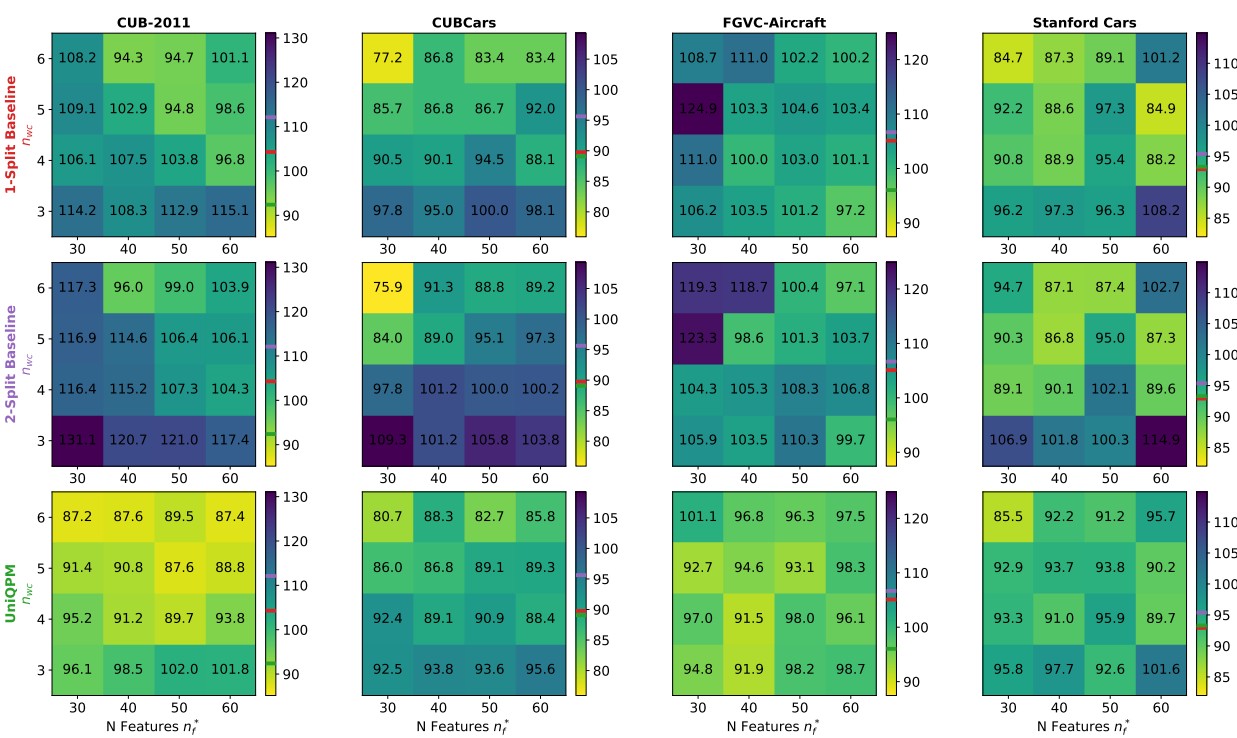

Figure 21: Scaled THR Fuzziness as percent of base QPM of MonoQPM applied to Resnet50 QPMs for different datasets with varying number of features $(n_f^*)$ and features per class $(n_{wc})$. The color code is based on unrounded values.The colorbar is shared across methods for one dataset and the mean of each method is indicated via color.

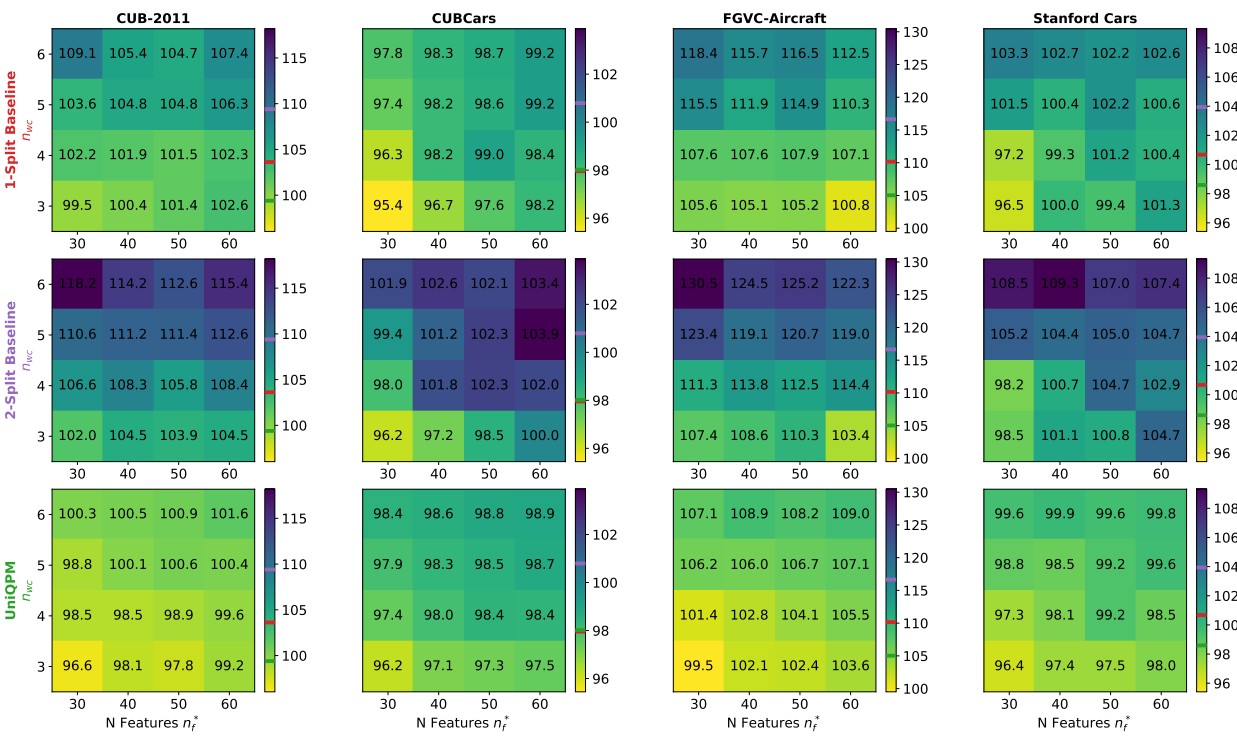

Figure 22: Scaled Negative Log-Likelihood (NLL) as percent of base QPM of MonoQPM applied to Resnet50 QPMs for different datasets with varying number of features ($n_f^*$) and features per class ($n_{wc}$). The color code is based on unrounded values.The colorbar is shared across methods for one dataset and the mean of each method is indicated via color.

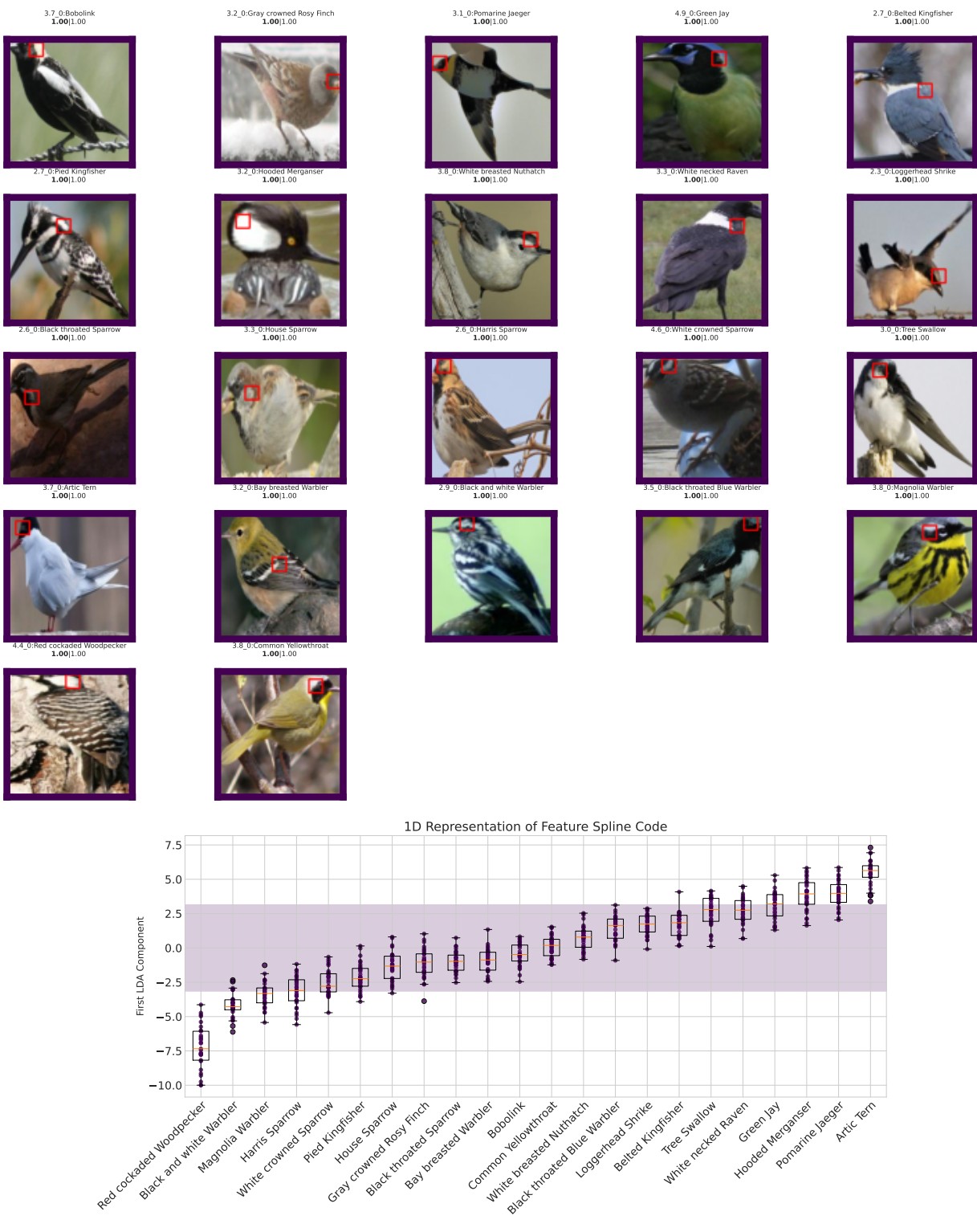

Figure 23: Exemplary Feature of QPM trained on CUB-2011 that MonoQPM maintained as monosemantic concept. The top figure shows exemplary images and the bottom shows one-dimensional representation of internal activations, as described in appendix A.

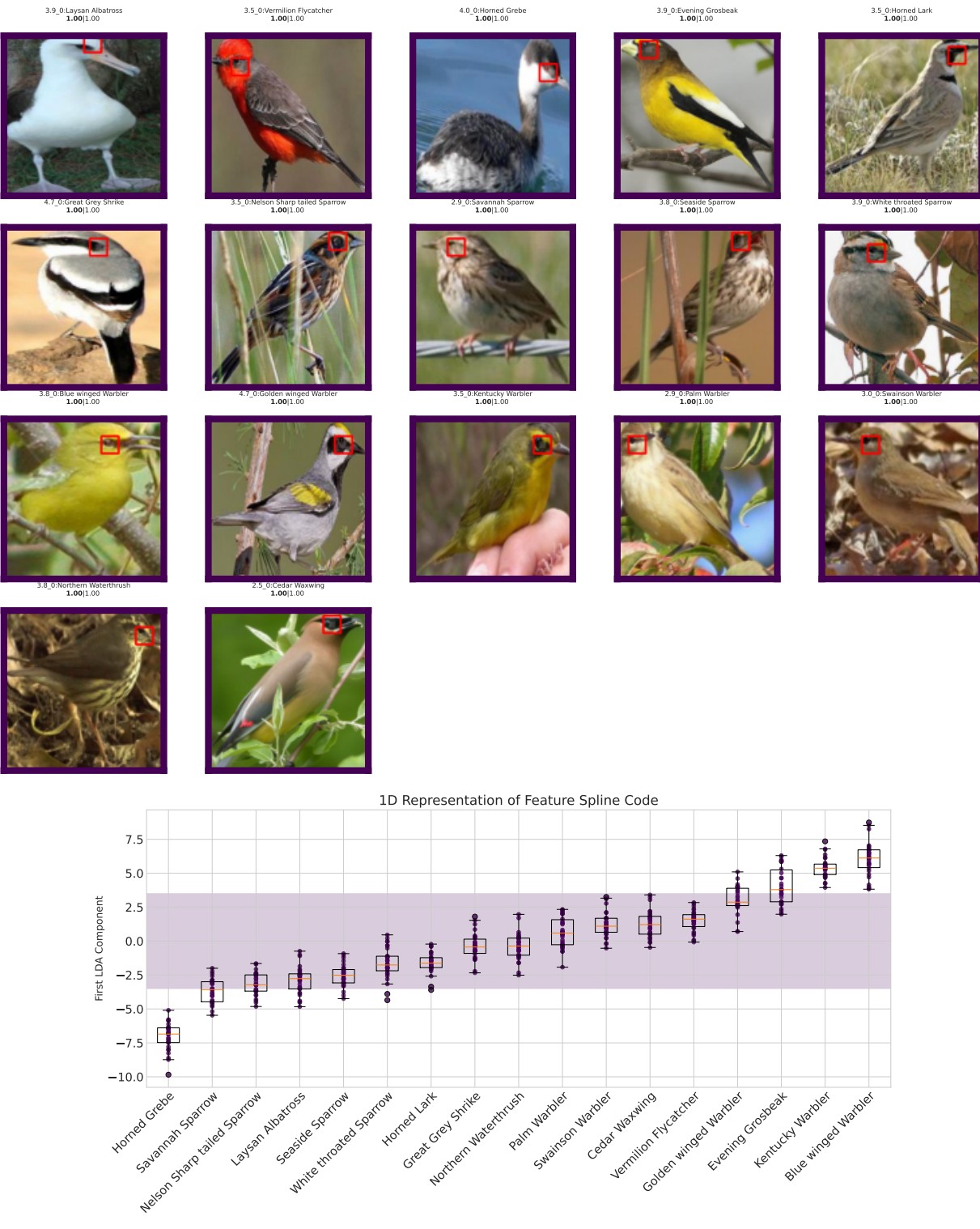

Figure 24: Exemplary Feature of QPM trained on CUB-2011 that MonoQPM maintained as monosemantic concept. The top figure shows exemplary images and the bottom shows one-dimensional representation of internal activations, as described in appendix A.

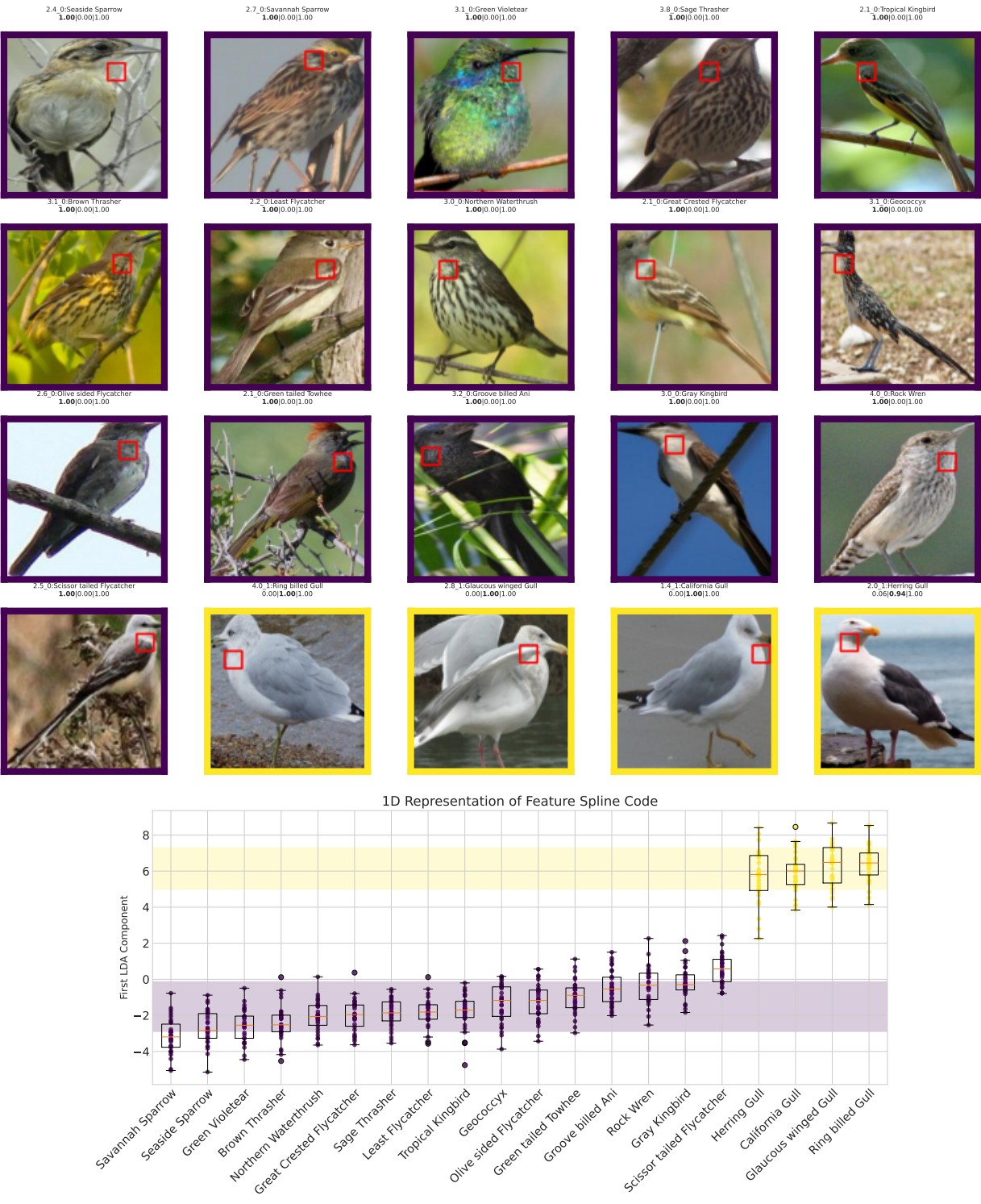

Figure 25: Exemplary Split Feature of QPM trained on CUB-2011. The top figure shows exemplary images and the bottom shows one-dimensional representation of internal activations, as described in appendix A.

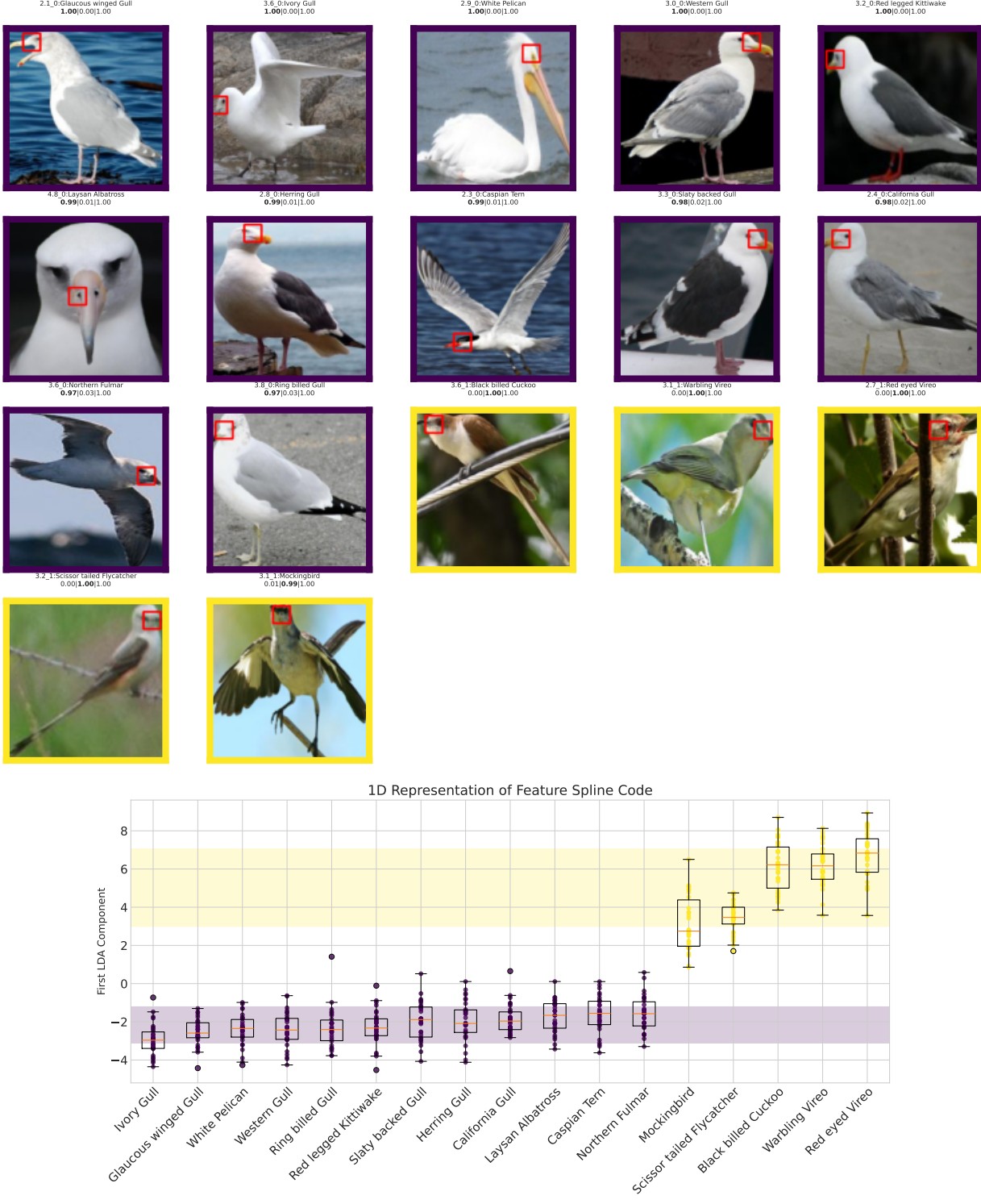

Figure 26: Exemplary Split Feature of QPM trained on CUB-2011. The top figure shows exemplary images and the bottom shows one-dimensional representation of internal activations, as described in appendix A.

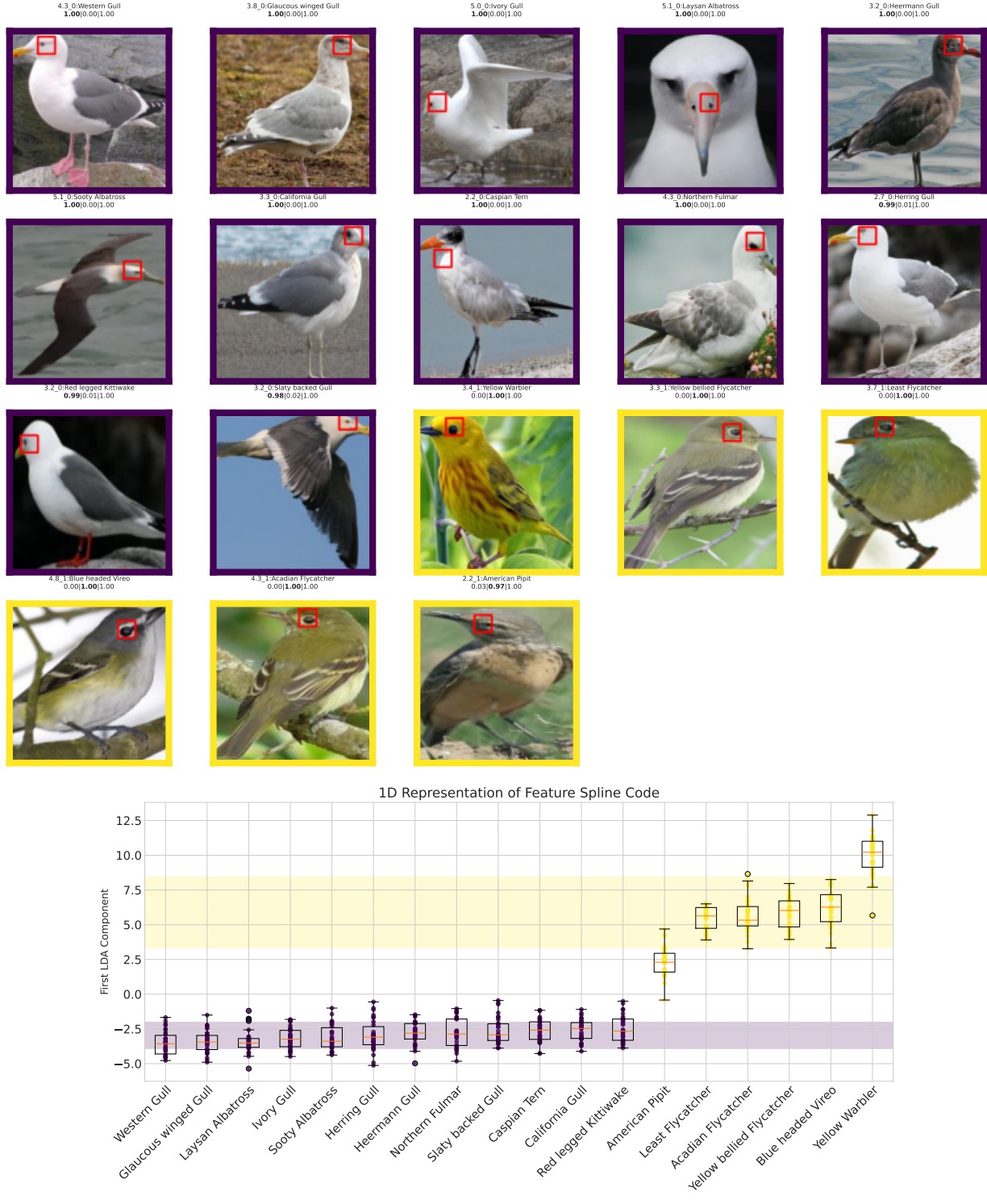

Figure 27: Exemplary Split Feature of QPM trained on CUB-2011. The top figure shows exemplary images and the bottom shows one-dimensional representation of internal activations, as described in appendix A.

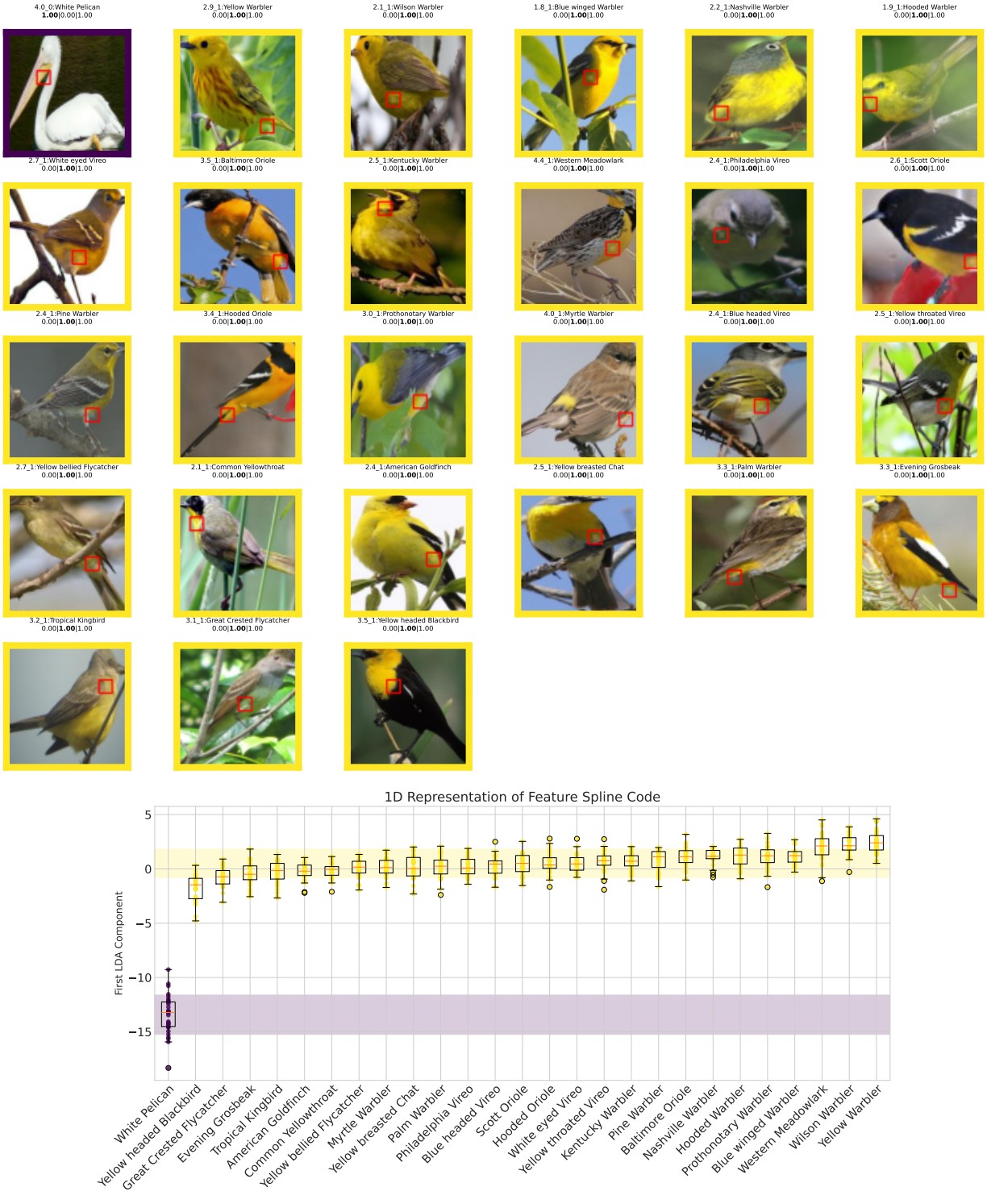

Figure 28: Exemplary Split Feature of QPM trained on CUB-2011. The top figure shows exemplary images and the bottom shows one-dimensional representation of internal activations, as described in appendix A.

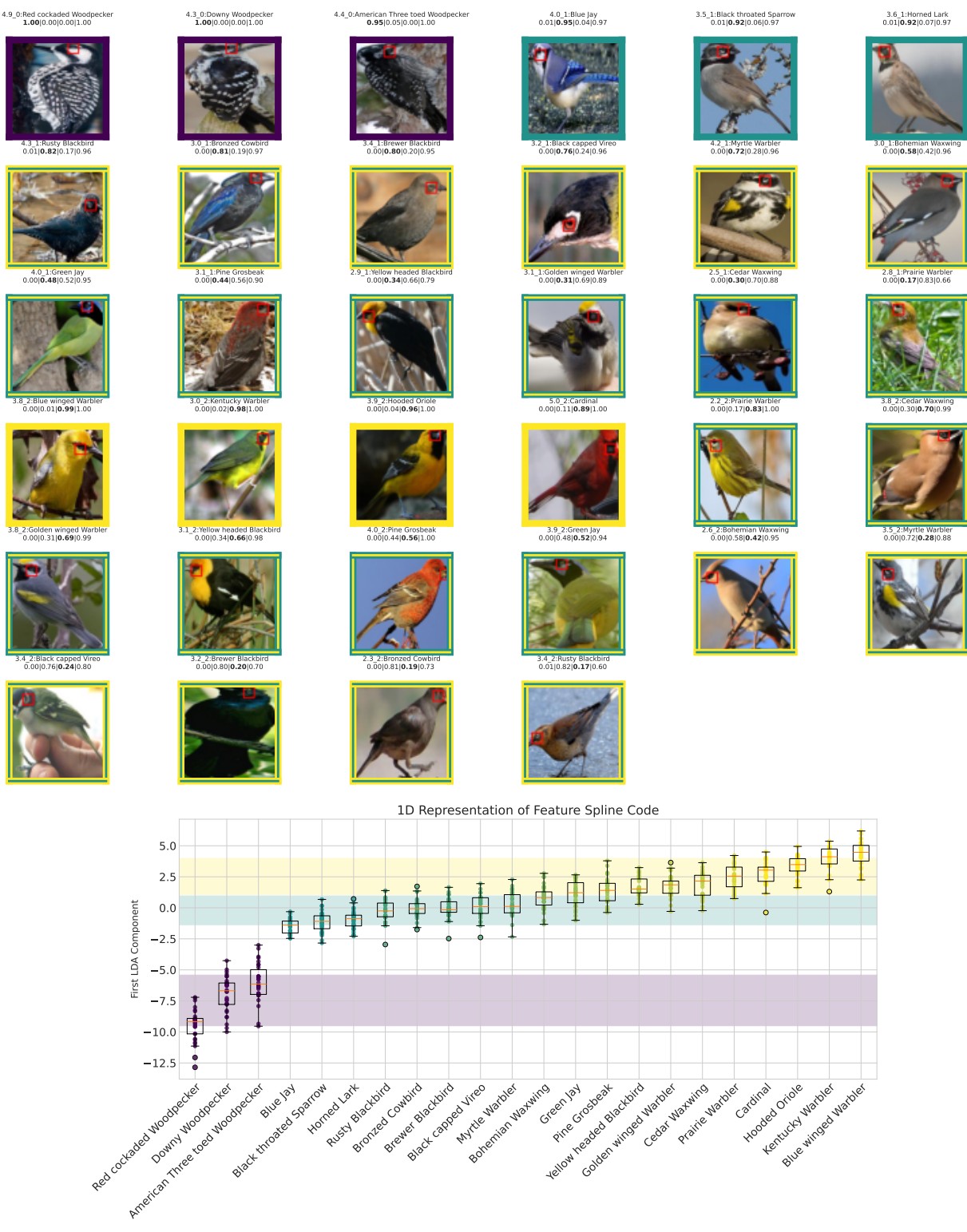

Figure 29: Exemplary Split Feature of QPM trained on CUB-2011. The top figure shows exemplary images and the bottom shows one-dimensional representation of internal activations, as described in appendix A.

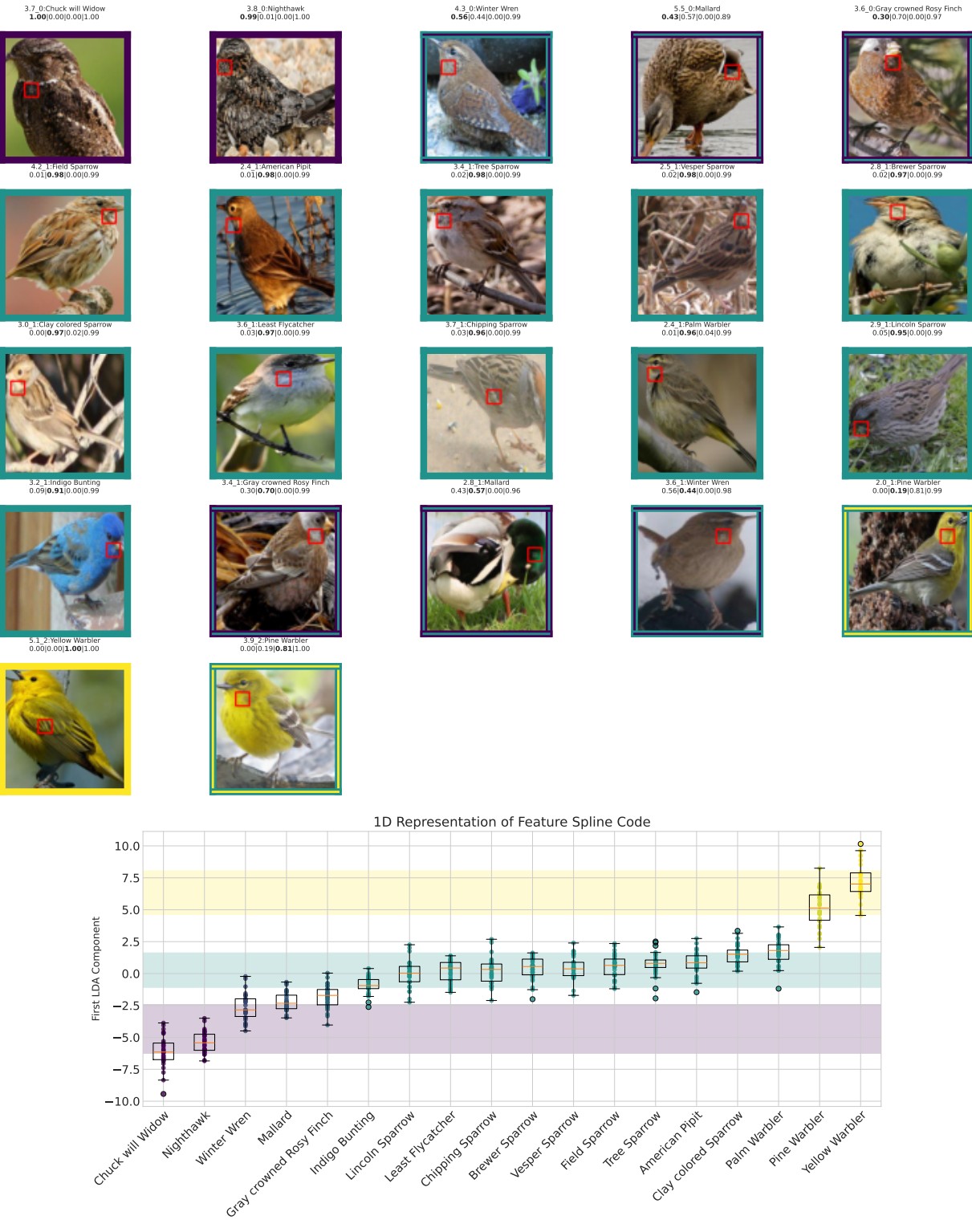

Figure 30: Exemplary Split Feature of QPM trained on CUB-2011. The top figure shows exemplary images and the bottom shows one-dimensional representation of internal activations, as described in appendix A.

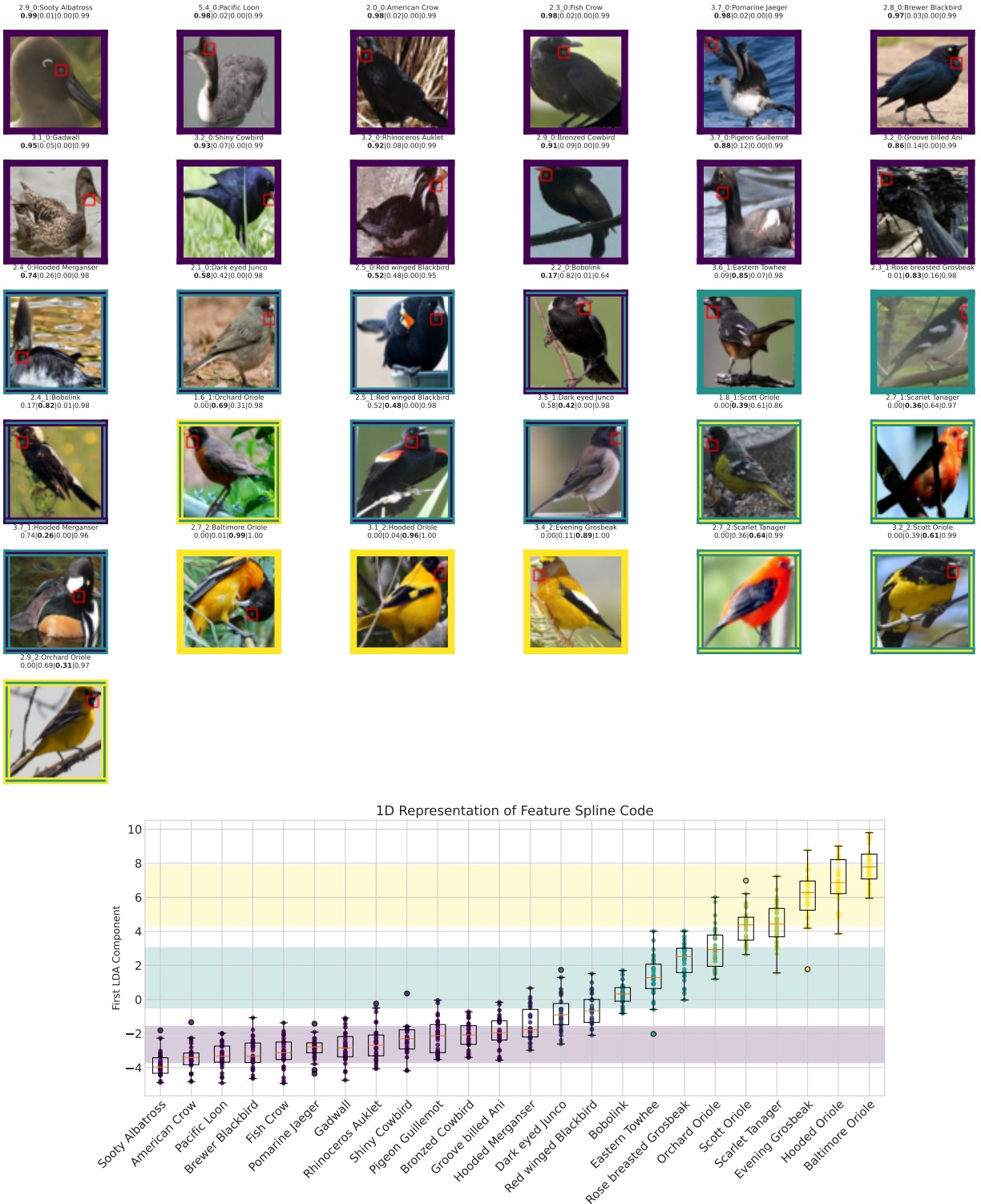

Figure 31: Exemplary Split Feature of QPM trained on CUB-2011. The top figure shows exemplary images and the bottom shows one-dimensional representation of internal activations, as described in appendix A.

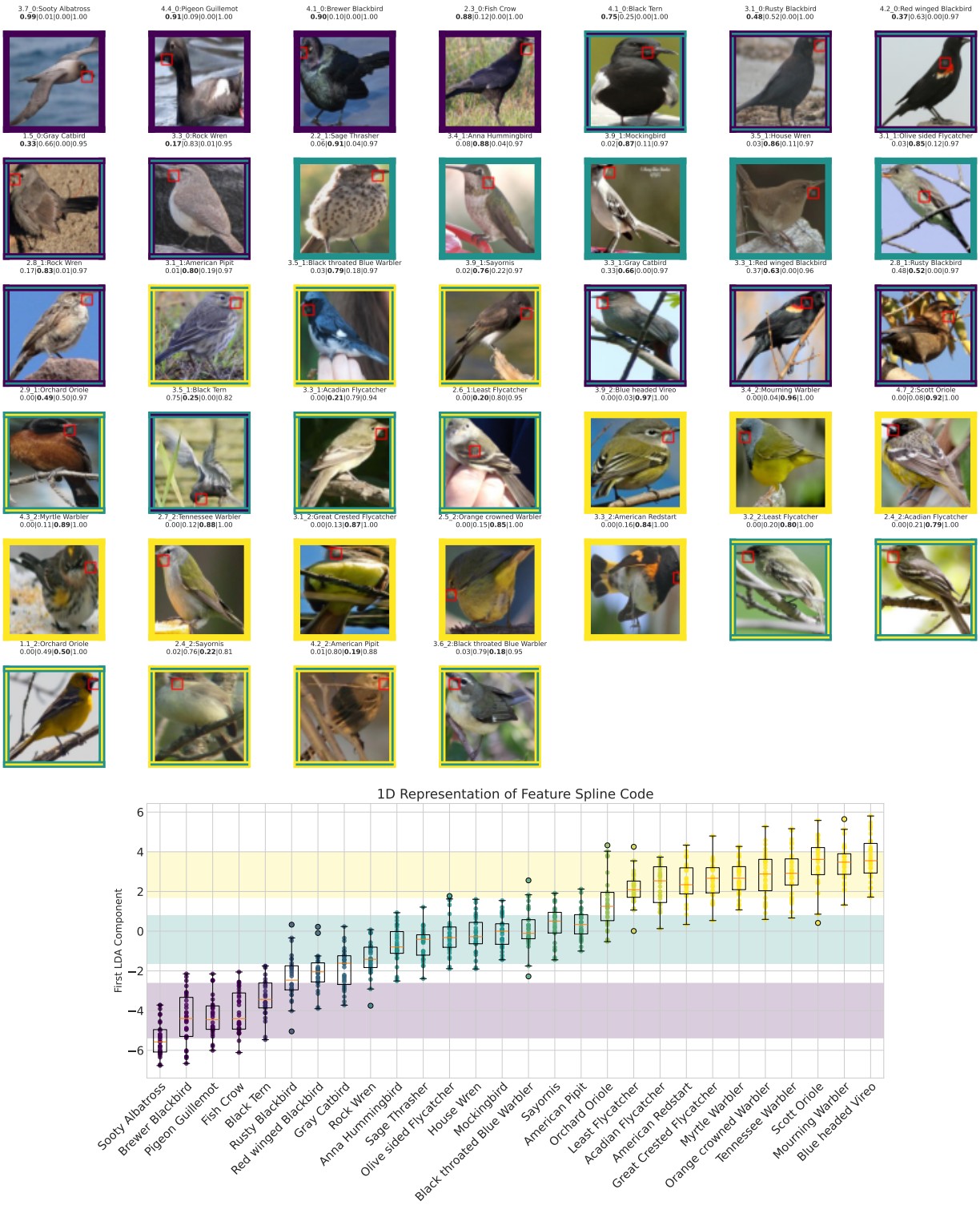

Figure 32: Exemplary Split Feature of QPM trained on CUB-2011. The top figure shows exemplary images and the bottom shows one-dimensional representation of internal activations, as described in appendix A.

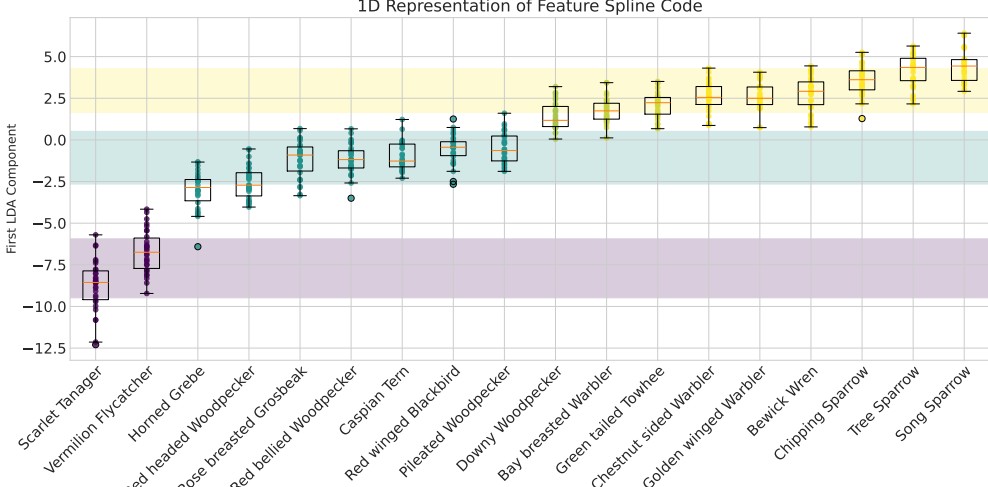

Figure 33: Exemplary Split Feature of QPM trained on CUB-2011. The top figure shows exemplary images and the bottom shows one-dimensional representation of internal activations, as described in appendix A.

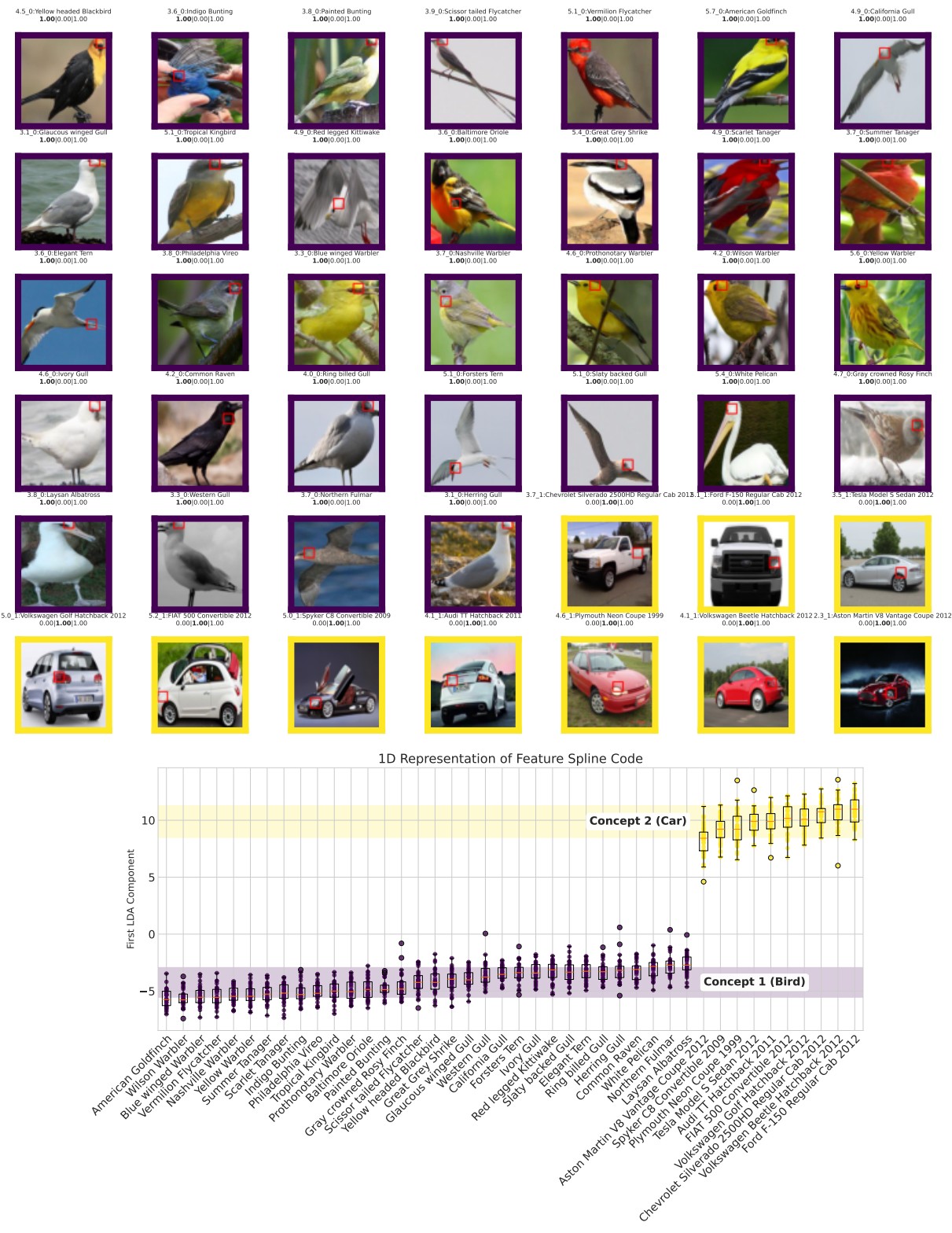

Figure 34: Exemplary Split Feature of QPM trained on CUBCars. The top figure shows exemplary images and the bottom shows one-dimensional representation of internal activations, as described in appendix A.

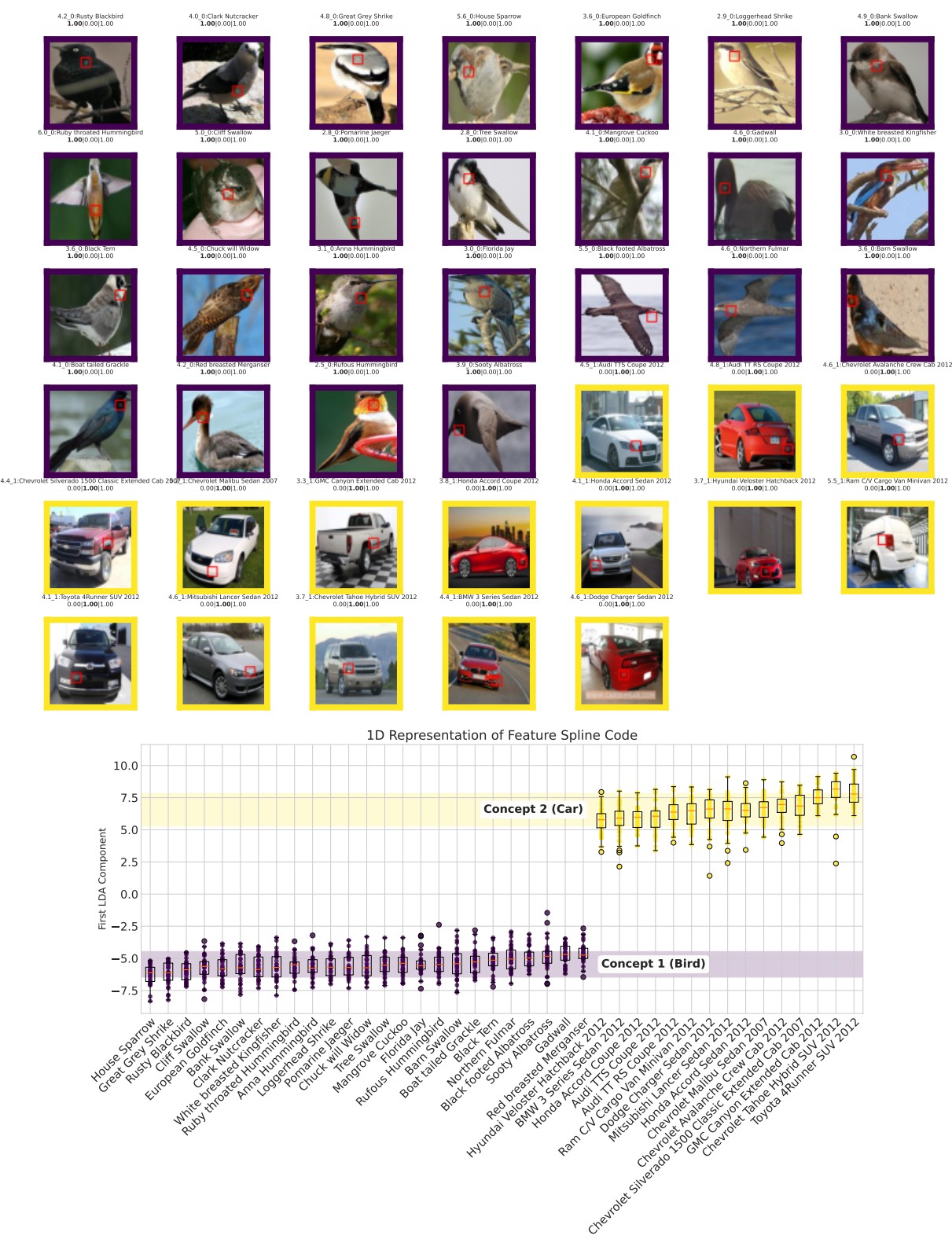

Figure 35: Exemplary Split Feature of QPM trained on CUBCars. The top figure shows exemplary images and the bottom shows one-dimensional representation of internal activations, as described in appendix A.

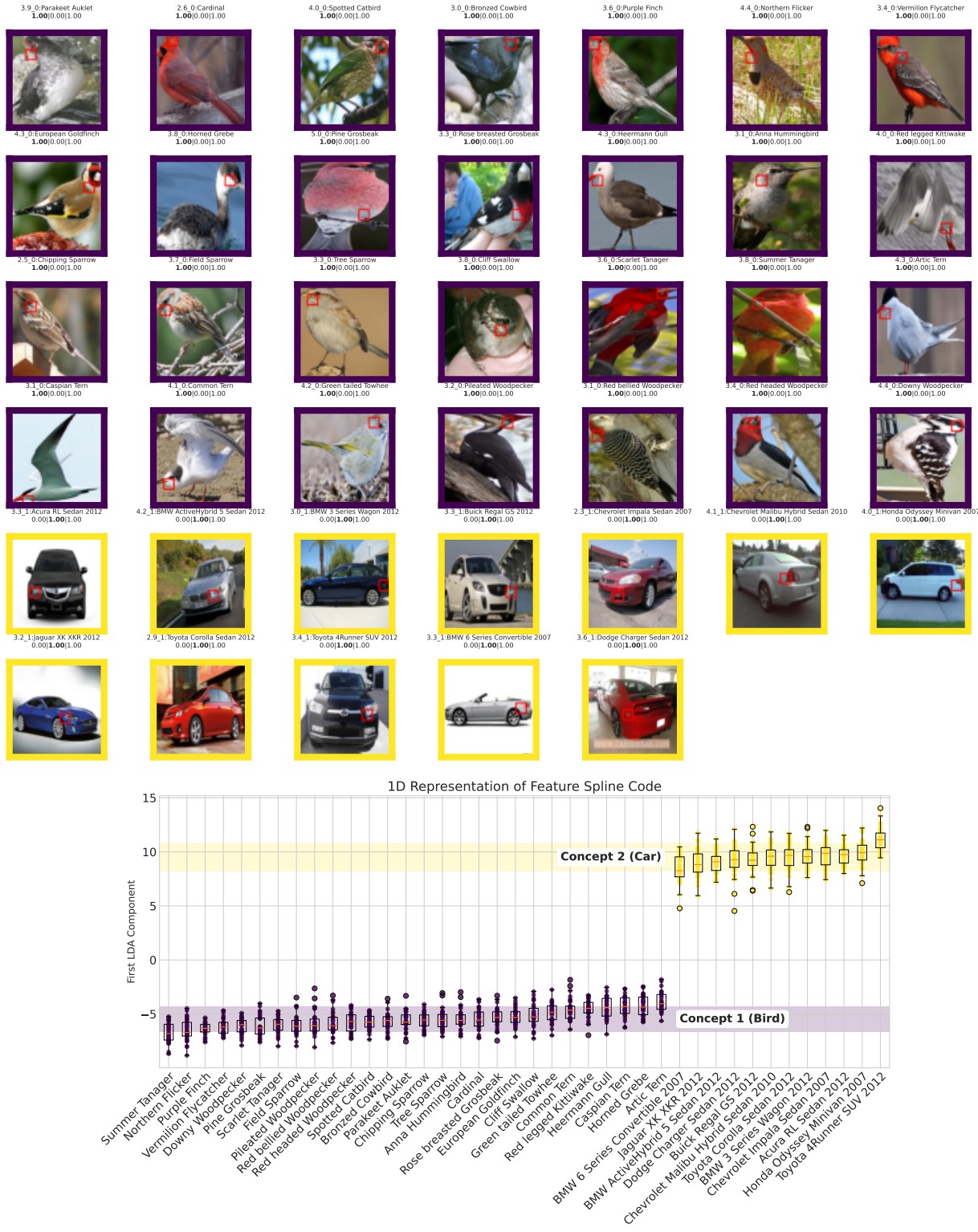

Figure 36: Exemplary Split Feature of QPM trained on CUBCars. The top figure shows exemplary images and the bottom shows one-dimensional representation of internal activations, as described in appendix A.

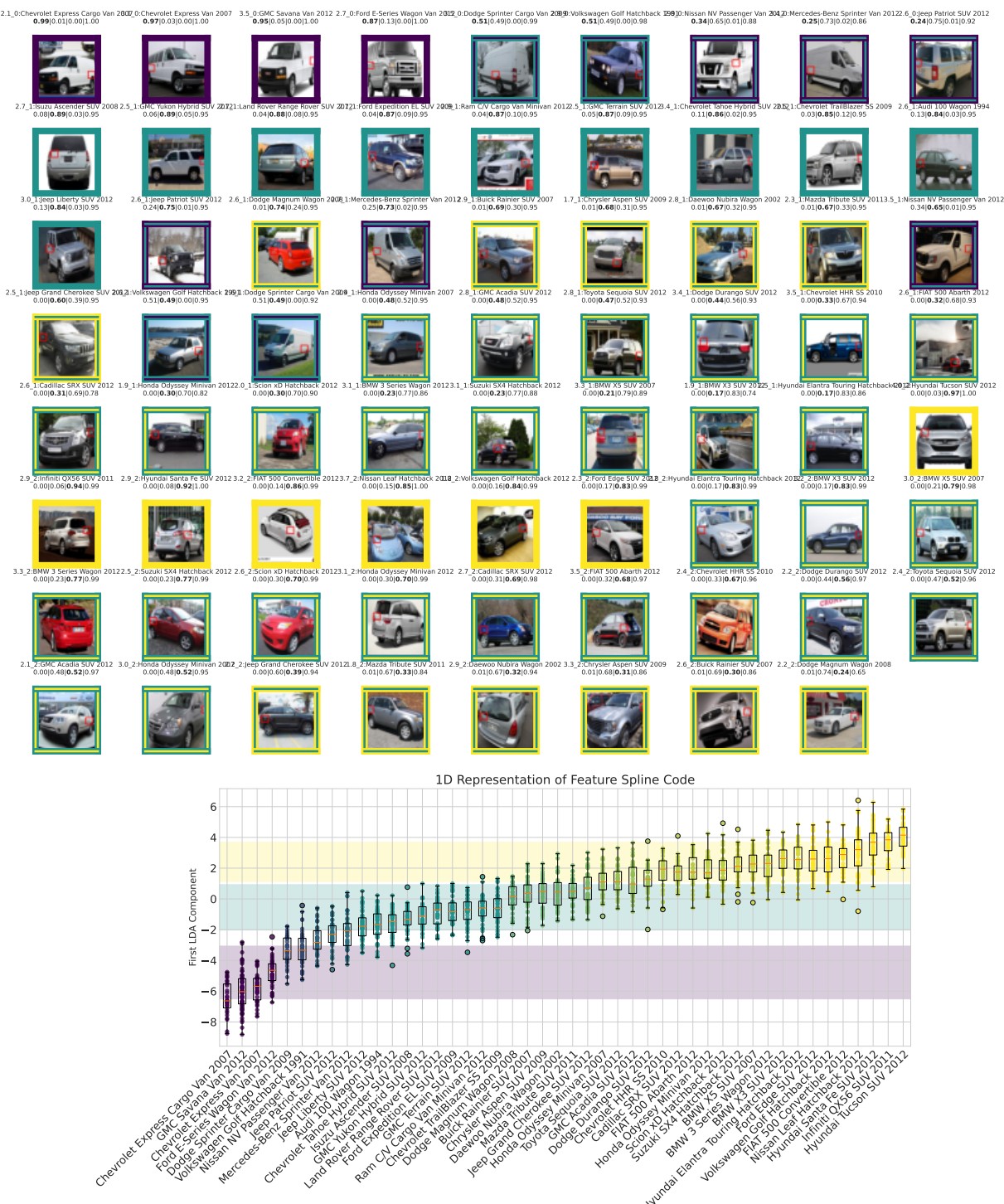

Figure 37: Exemplary Split Feature of QPM trained on CUBCars. The top figure shows exemplary images and the bottom shows one-dimensional representation of internal activations, as described in appendix A.

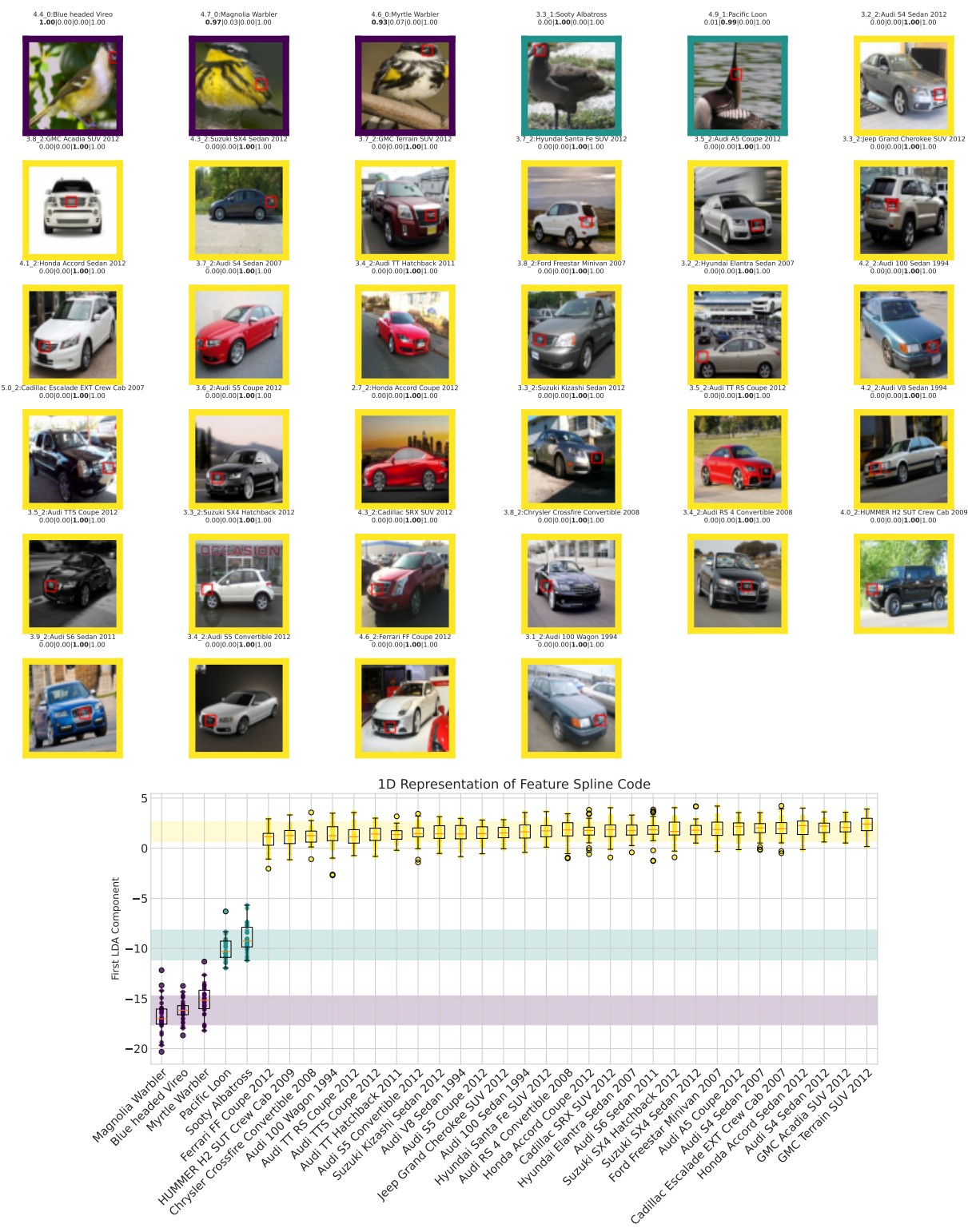

Figure 38: Exemplary Split Feature of QPM trained on CUBCars. The top figure shows exemplary images and the bottom shows one-dimensional representation of internal activations, as described in appendix A.

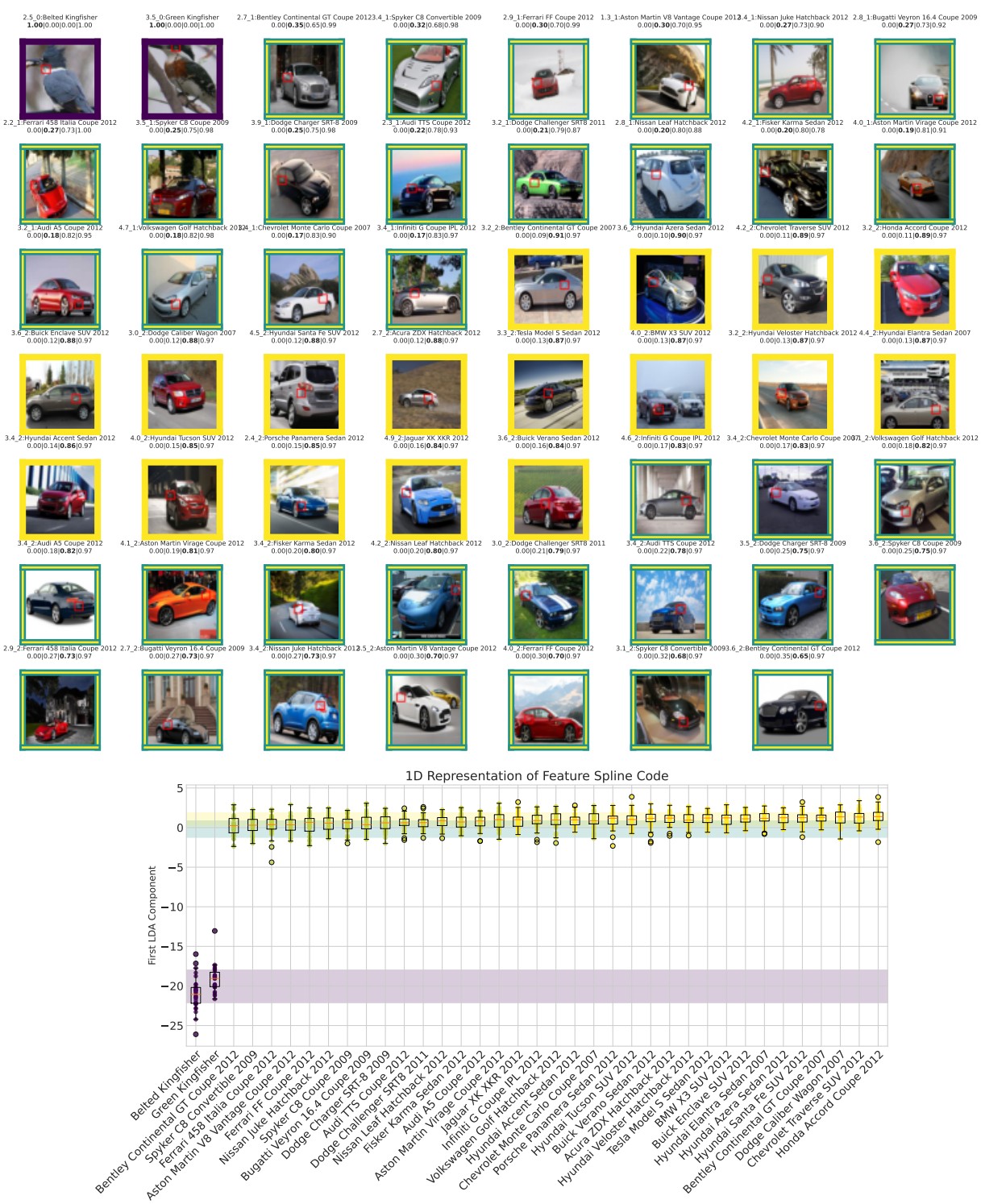

Figure 39: Exemplary Split Feature of QPM trained on CUBCars. The top figure shows exemplary images and the bottom shows one-dimensional representation of internal activations, as described in appendix A.

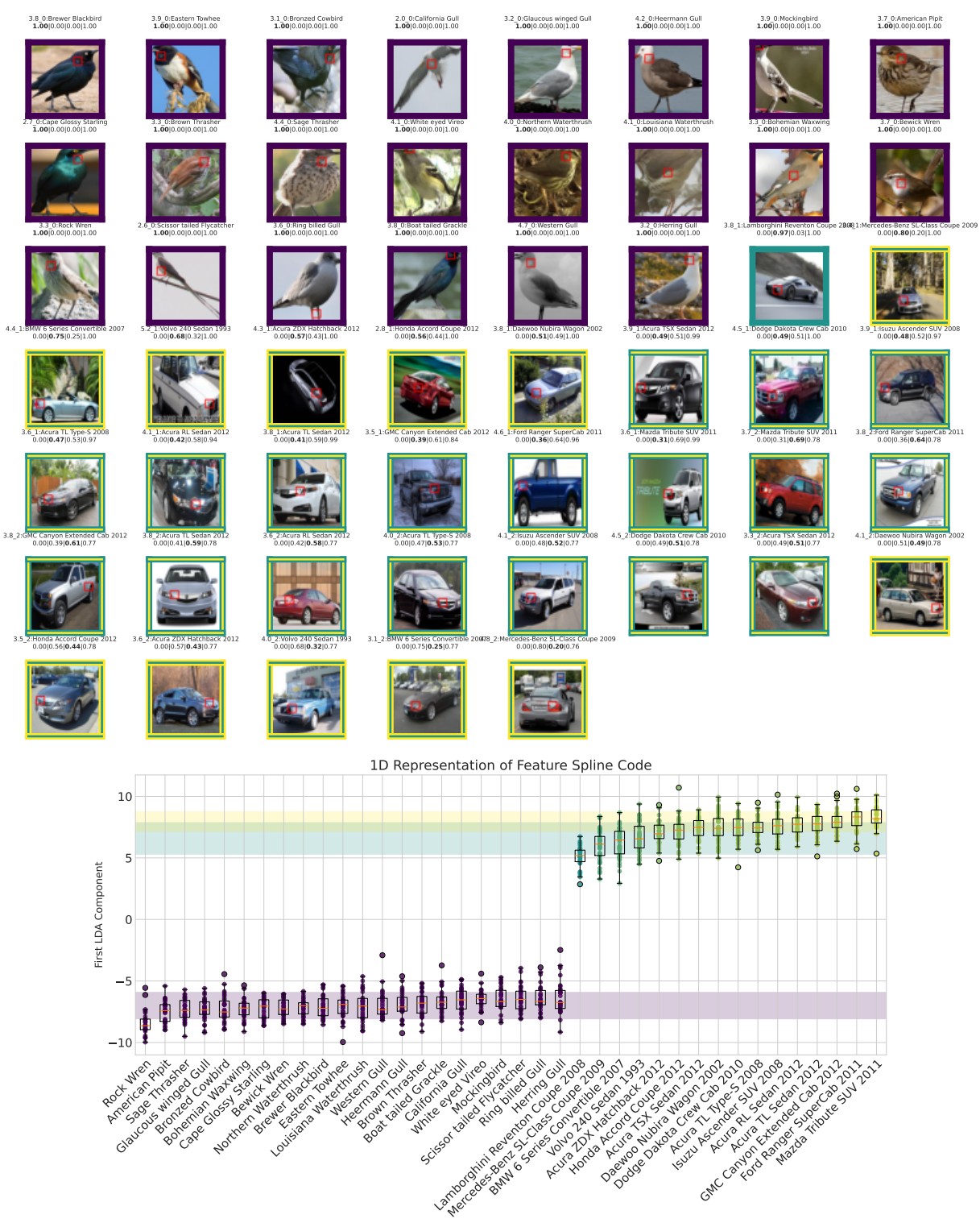

Figure 40: Exemplary Split Feature of QPM trained on CUBCars. The top figure shows exemplary images and the bottom shows one-dimensional representation of internal activations, as described in appendix A.

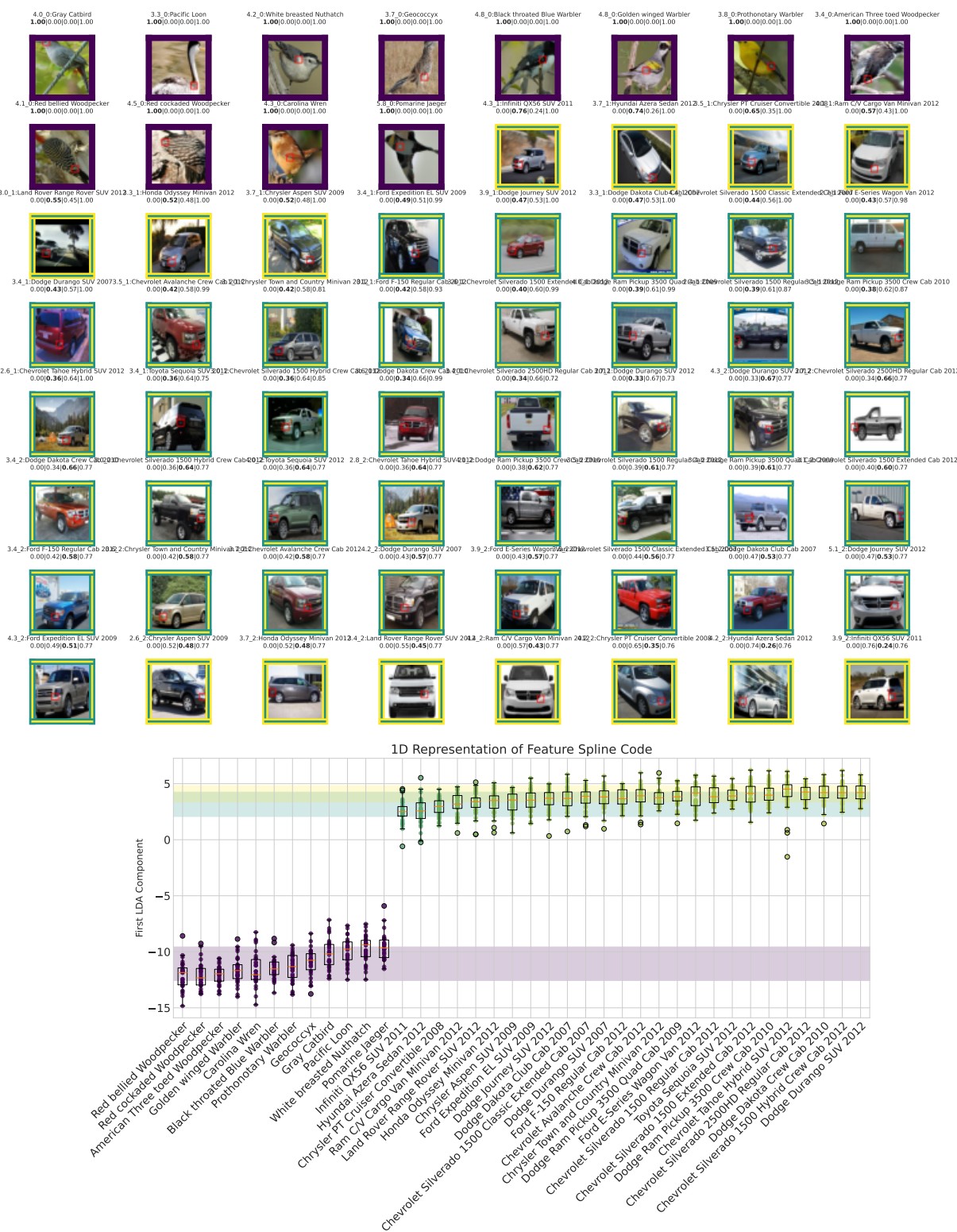

Figure 41: Exemplary Split Feature of model trained on CUBCars, where all car classes have been assigned to all car concepts. The top figure shows exemplary images and the bottom shows one-dimensional representation of internal activations, as described in appendix A.

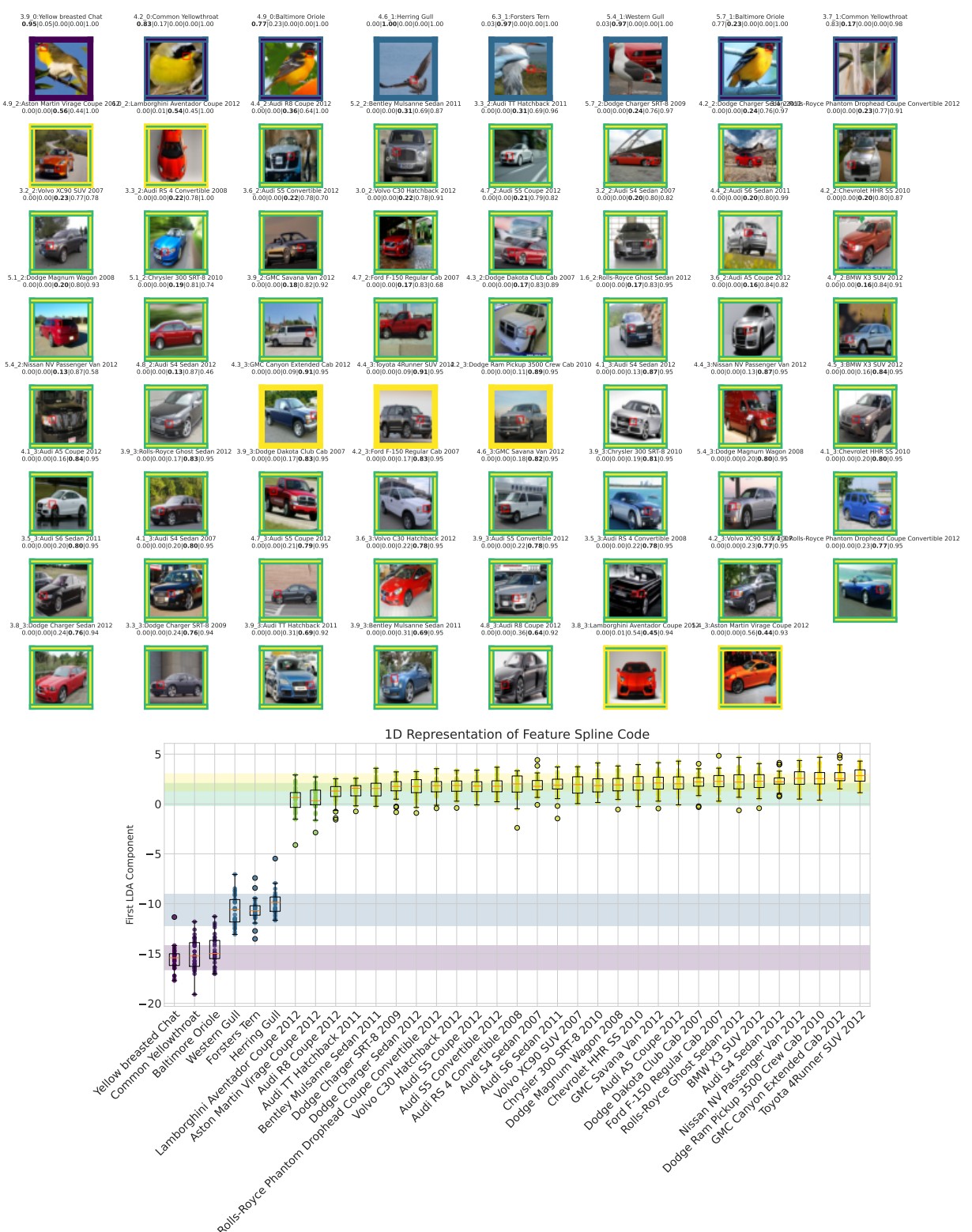

Figure 42: Exemplary Split Feature of QPM trained on CUBCars. The top figure shows exemplary images and the bottom shows one-dimensional representation of internal activations, as described in appendix A.

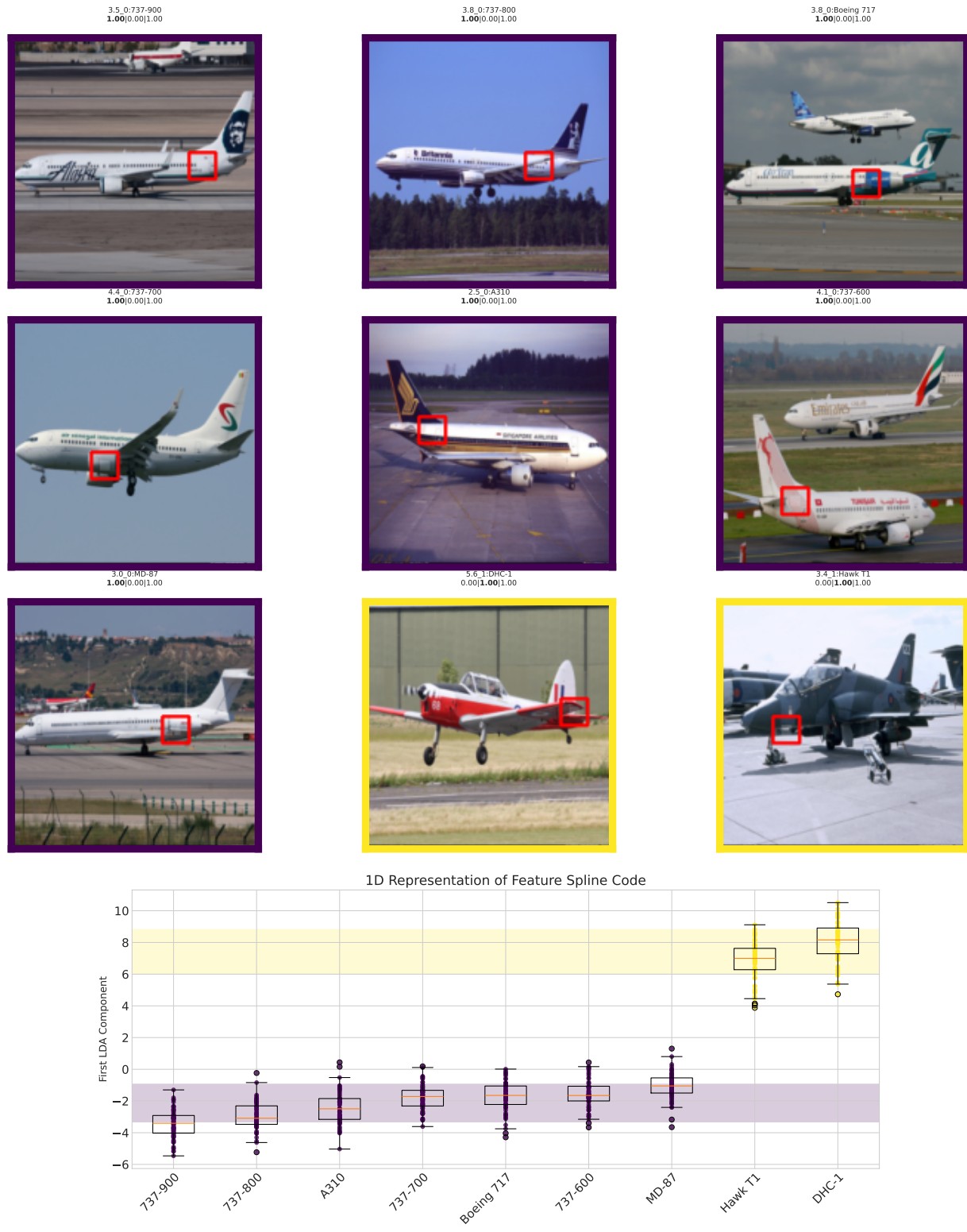

Figure 43: Exemplary Split Feature of QPM trained on FGVC-Aircraft. The top figure shows exemplary images and the bottom shows one-dimensional representation of internal activations, as described in appendix A.

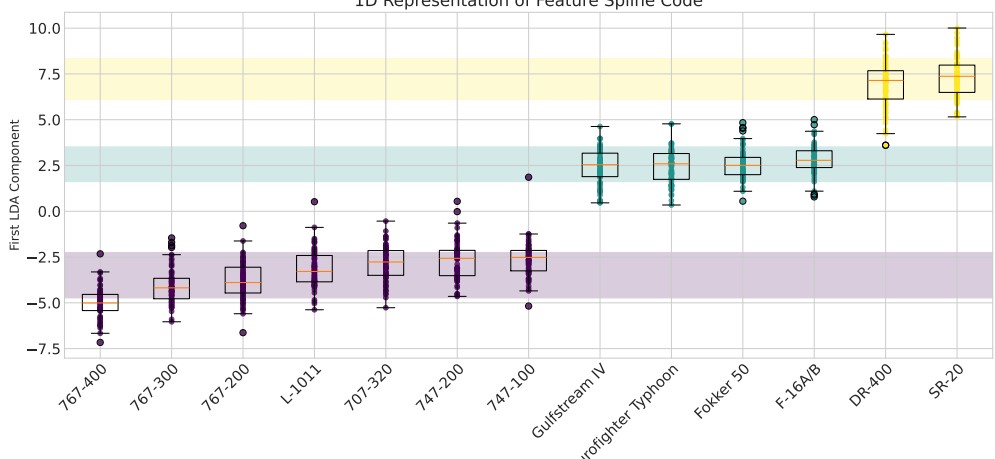

Figure 44: Exemplary Split Feature of QPM trained on FGVC-Aircraft. The top figure shows exemplary images and the bottom shows one-dimensional representation of internal activations, as described in appendix A.

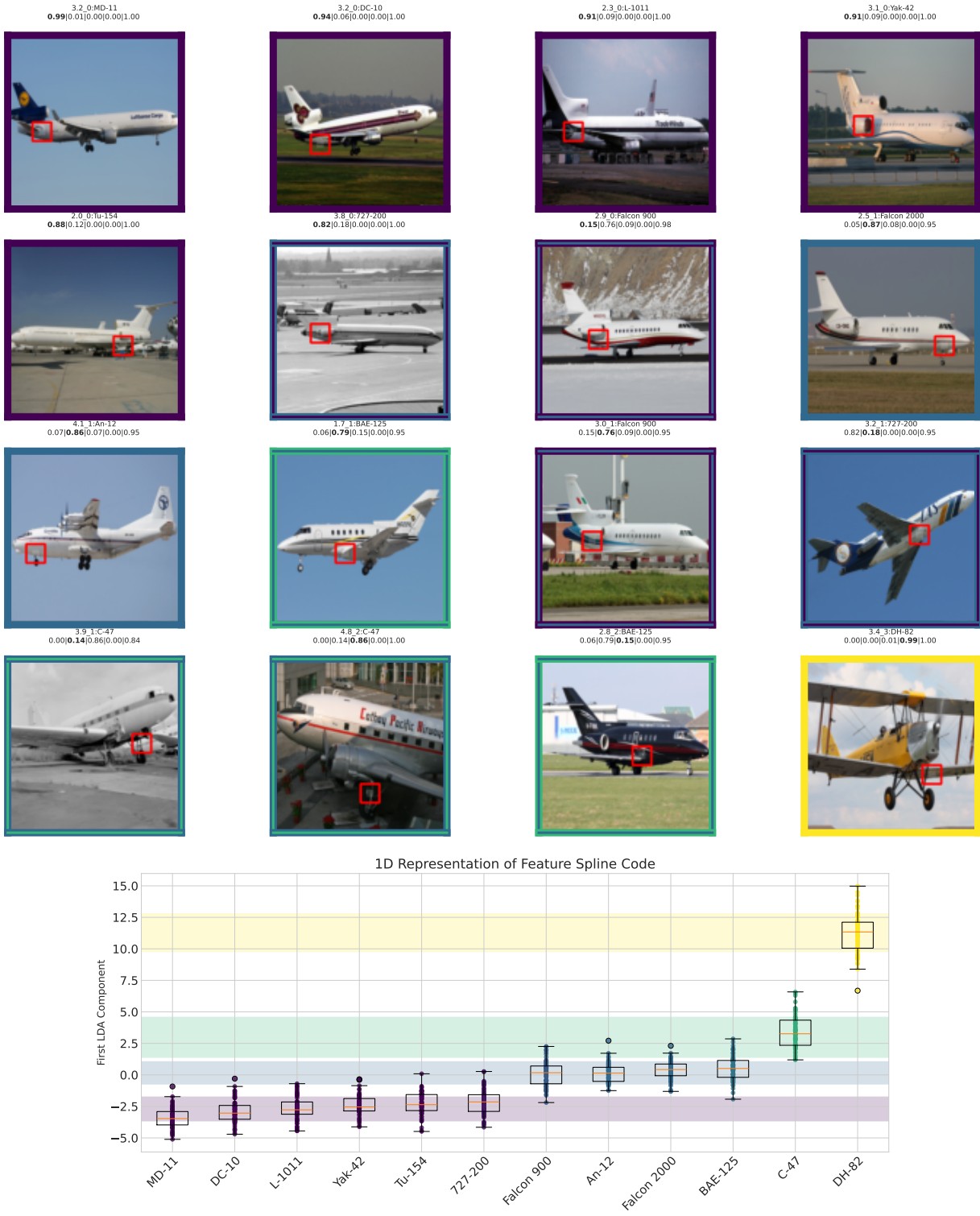

Figure 45: Exemplary Split Feature of QPM trained on FGVC-Aircraft. The top figure shows exemplary images and the bottom shows one-dimensional representation of internal activations, as described in appendix A.

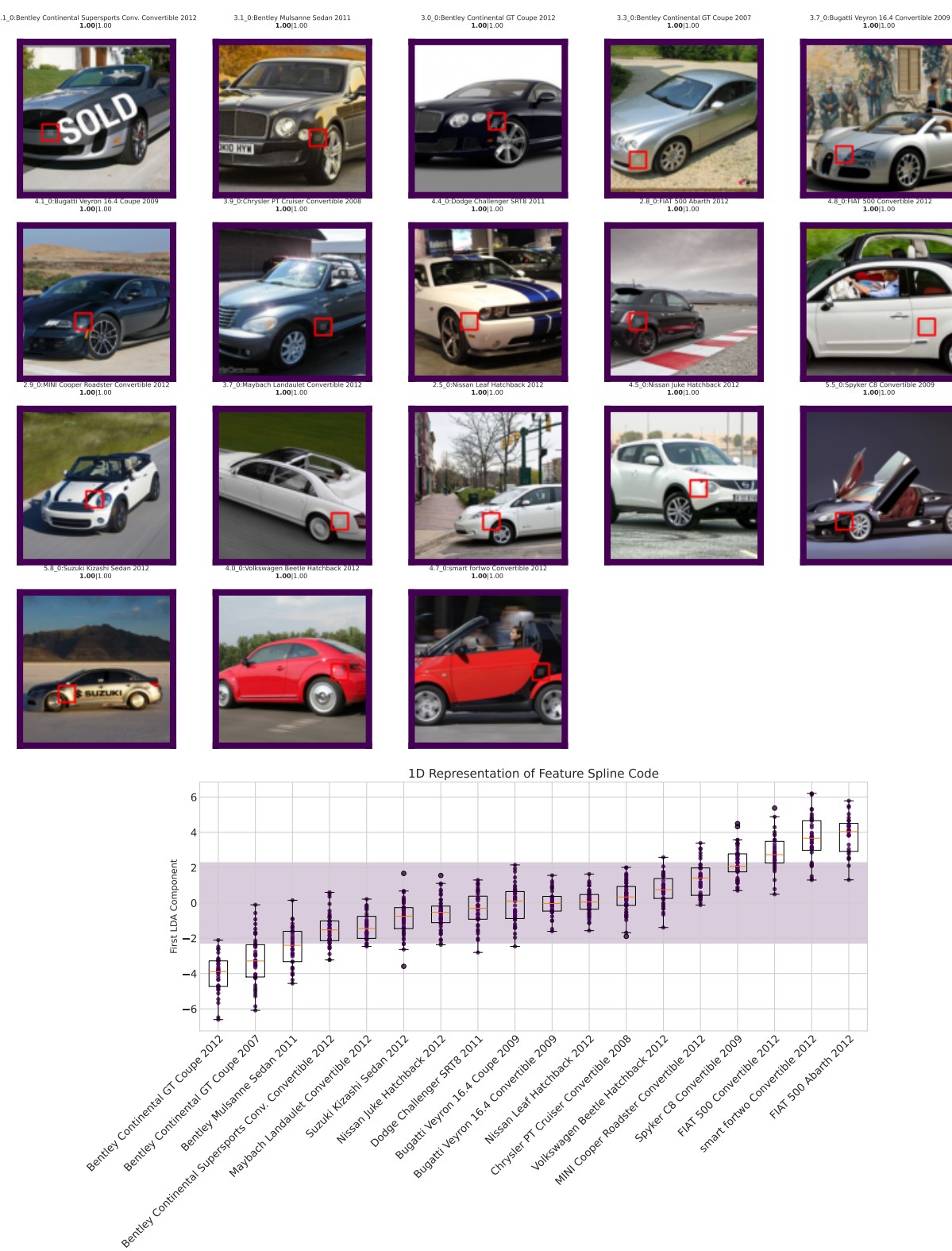

Figure 46: Exemplary Feature of QPM trained on Stanford Carsthat MonoQPM maintained as monosemantic concept. The top figure shows exemplary images and the bottom shows one-dimensional representation of internal activations, as described in appendix A.

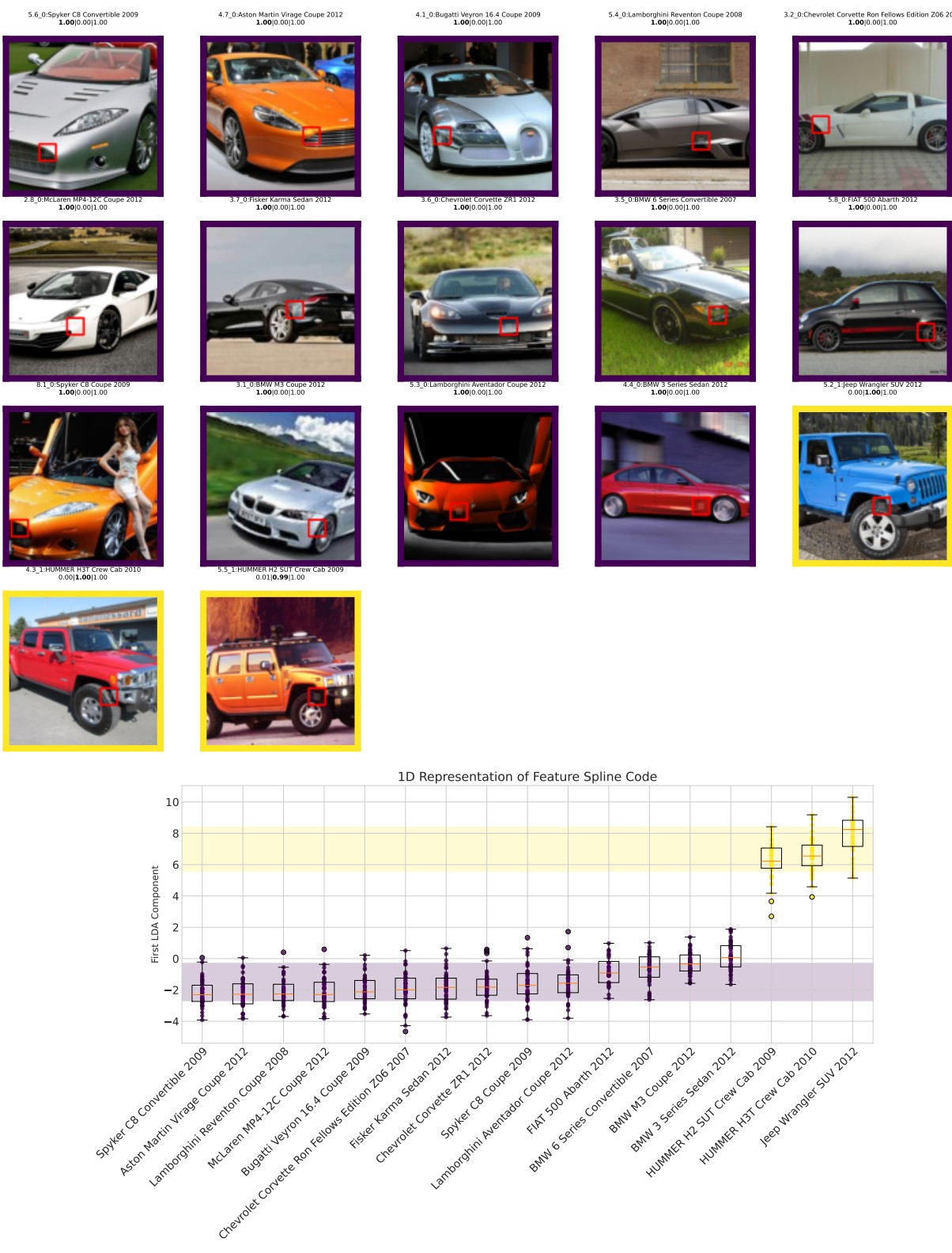

Figure 47: Exemplary Split Feature of QPM trained on Stanford Cars. The top figure shows exemplary images and the bottom shows one-dimensional representation of internal activations, as described in appendix A.

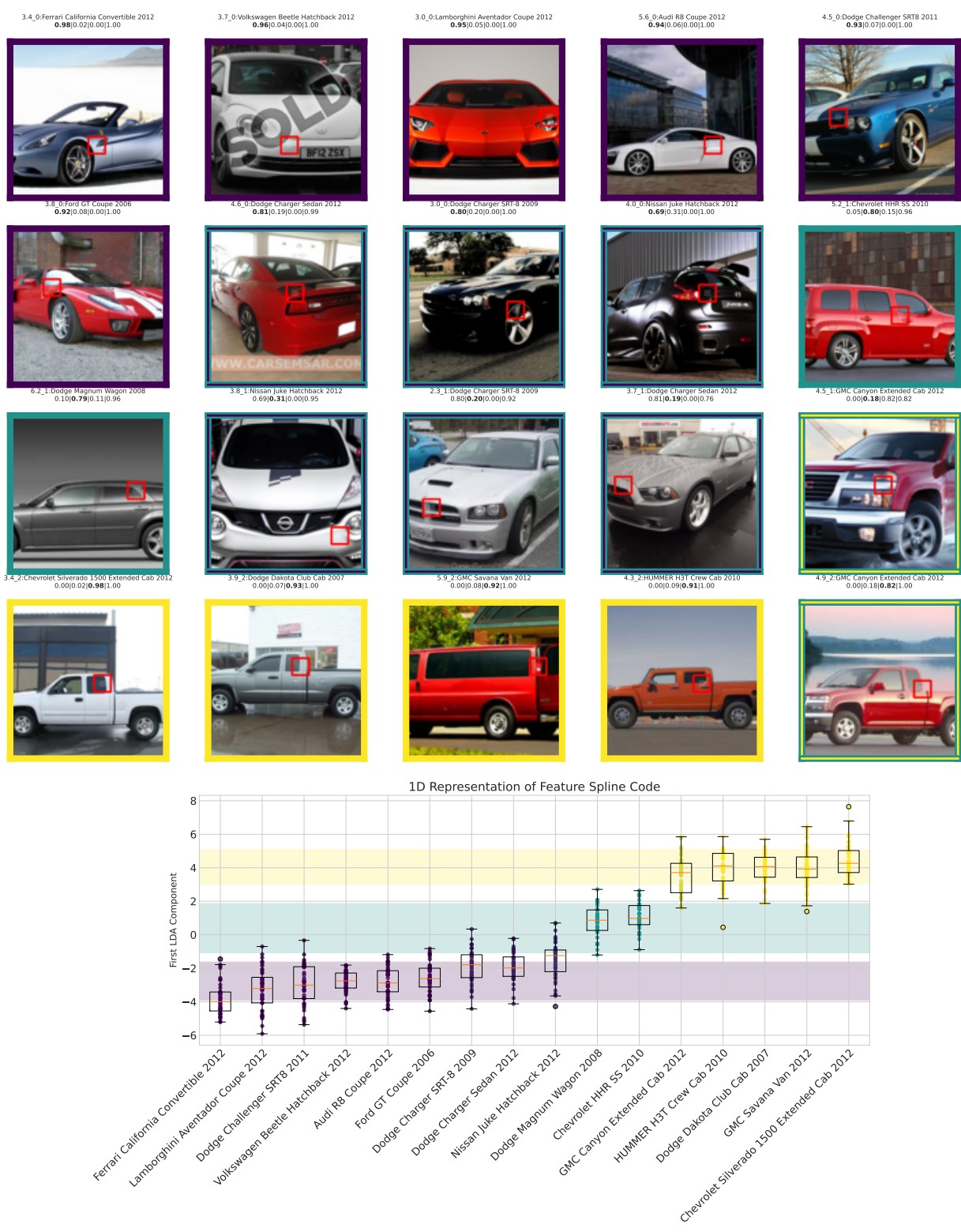

Figure 48: Exemplary Split Feature of QPM trained on Stanford Cars. The top figure shows exemplary images and the bottom shows one-dimensional representation of internal activations, as described in appendix A.

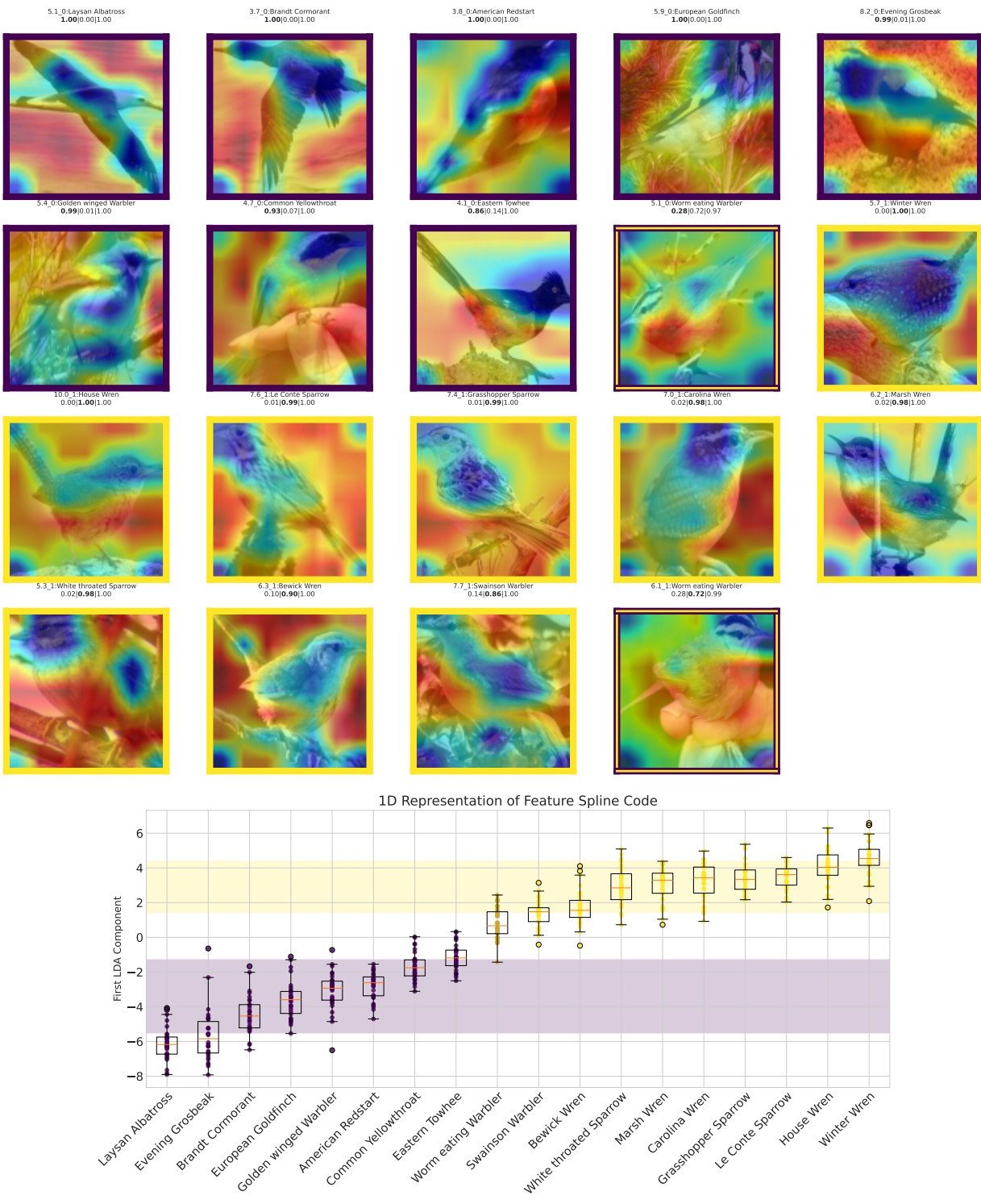

Figure 49: Visualization of heatmaps of one Feature of a Swin-tiny QPM on CUB-2011 with MonoQPM split The top figure shows exemplary images and the bottom shows one-dimensional representation of internal activations, as described in appendix A.

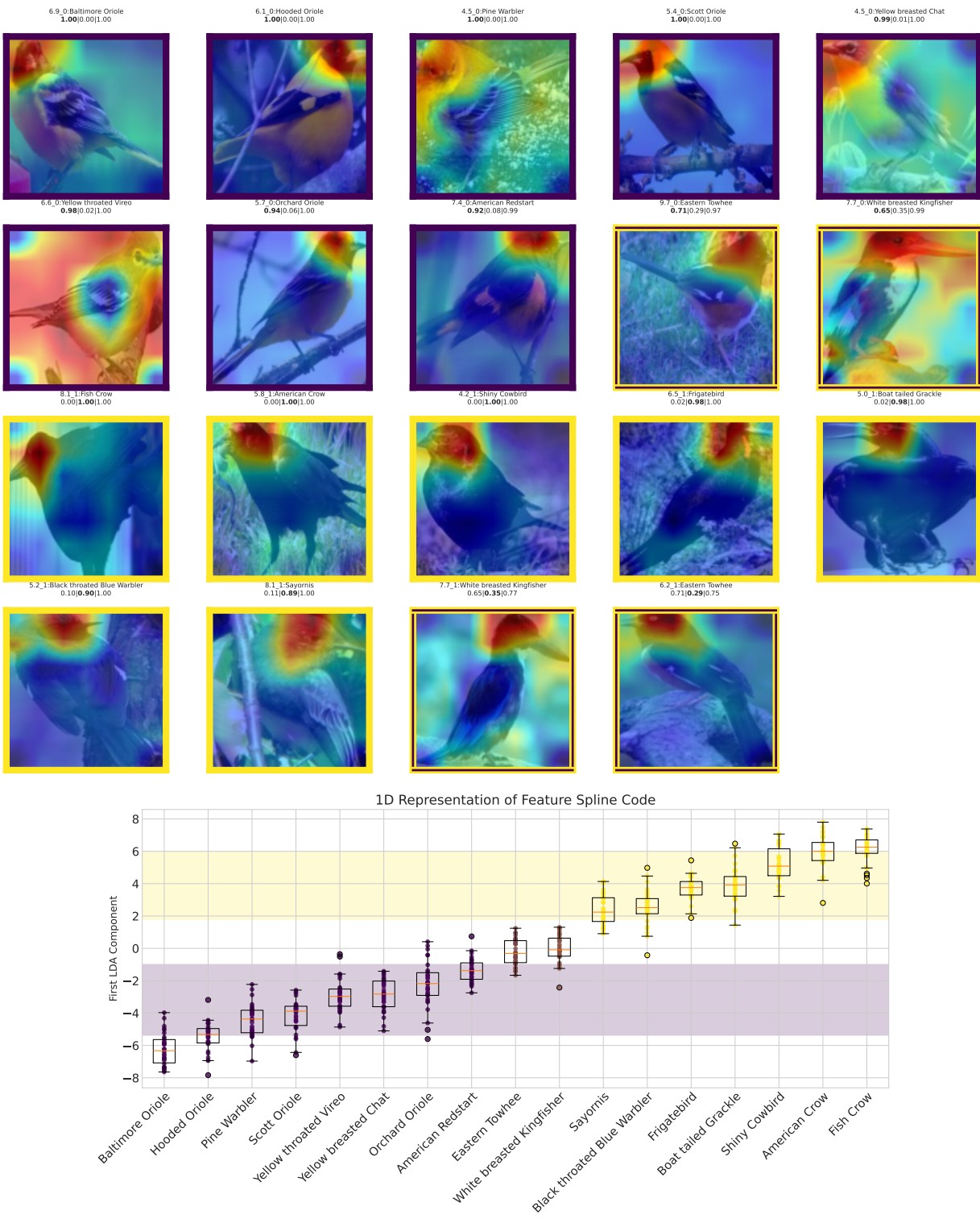

Figure 50: Visualization of mostly well localizing heatmaps of one Feature of a Swin-tiny QPM on CUB-2011 with MonoQPM split The top figure shows exemplary images and the bottom shows one-dimensional representation of internal activations, as described in appendix A.

Table 11: Accuracy comparison across datasets for Resnet50. We report the delta in Top-*k* accuracy (%) and Negative Log-Likelihood (NLL) compared to the polysemantic QPM. The best value is highlighted bold, all values underlined that beat the baseline QPM: MonoQPM's improved CP efficiency does not entail reduced Top-*k* performance.

| Dataset | Method | Top-1 | Top-2 | Top-5 | Top-10 | Top-20 | NLL ↓ |
|---|---|---|---|---|---|---|---|
| **CUB-2011** | 1-Split Baseline | **0.0** | -0.08 | -0.18 | -0.07 | -0.18 | 0.03 |
| | 2-Split Baseline | **0.0** | -0.26 | -0.43 | -0.22 | -0.26 | 0.08 |
| | MonoQPM (Ours) | **0.0** | **0.00** | -0.04 | **0.16** | **0.00** | **0.00** |
| **StanfordCars** | 1-Split Baseline | **0.0** | 0.04 | -0.02 | -0.03 | -0.05 | 0.01 |
| | 2-Split Baseline | **0.0** | 0.07 | -0.03 | -0.05 | -0.04 | 0.02 |
| | MonoQPM (Ours) | **0.0** | **0.08** | **0.08** | **0.03** | **0.05** | **-0.0** |
| **CUBCars** | 1-Split Baseline | **0.0** | **0.05** | **0.08** | 0.04 | 0.08 | **-0.01** |
| | 2-Split Baseline | **0.0** | -0.03 | -0.03 | -0.05 | 0.00 | 0.01 |
| | MonoQPM (Ours) | **0.0** | 0.02 | 0.07 | **0.06** | **0.10** | **-0.01** |
| **ImageNet** | 1-Split Baseline | **0.0** | -0.05 | -0.17 | -0.25 | -0.29 | 0.04 |
| | 2-Split Baseline | **0.0** | -0.17 | -0.43 | -0.59 | -0.57 | 0.11 |
| | MonoQPM (Ours) | **0.0** | **0.02** | **0.0** | -0.01 | -0.07 | 0.01 |

Table 12: Accuracy comparison across datasets for Inception-v3. We report the delta in Top-*k* accuracy (%) and Negative Log-Likelihood (NLL) compared to the polysemantic QPM. The best value among interpretable models is highlighted bold, all values underlined that beat the baseline QPM.

| Dataset | Method | Top-1 | Top-2 | Top-5 | Top-10 | Top-20 | NLL ↓ |
|---|---|---|---|---|---|---|---|
| **CUB-2011** | 1-Split Baseline | **0.00** | 0.34 | 0.63 | 0.36 | 0.20 | 0.02 |
| | 2-Split Baseline | **0.00** | **0.47** | **0.83** | **0.62** | **0.37** | 0.06 |
| | **Proposed Method** | **0.00** | 0.26 | 0.61 | 0.45 | 0.29 | 0.03 |
| **StanfordCars** | 1-Split Baseline | **0.00** | 0.23 | 0.28 | 0.19 | 0.02 | 0.01 |
| | 2-Split Baseline | **0.00** | **0.32** | **0.37** | **0.26** | **0.09** | 0.02 |
| | **Proposed Method** | **0.00** | 0.22 | 0.24 | 0.20 | 0.08 | 0.02 |
| **CUBCars** | 1-Split Baseline | **0.00** | 0.30 | 0.33 | 0.37 | 0.42 | 0.01 |
| | 2-Split Baseline | **0.00** | **0.40** | **0.44** | **0.56** | **0.54** | 0.03 |
| | **Proposed Method** | **0.00** | 0.13 | 0.19 | 0.17 | 0.29 | 0.01 |

Table 13: Accuracy comparison across datasets for Resnet34. We report the delta in Top-*k* accuracy (%) and Negative Log-Likelihood (NLL) compared to the polysemantic QPM. The best value among interpretable models is highlighted bold, all values underlined that beat the baseline QPM.

| Dataset | Method | Top-1 | Top-2 | Top-5 | Top-10 | Top-20 | NLL ↓ |
|---|---|---|---|---|---|---|---|
| **CUB-2011** | 1-Split Baseline | **0.00** | -0.03 | **0.10** | -0.09 | -0.22 | 0.05 |
| | 2-Split Baseline | **0.00** | -0.04 | -0.03 | -0.1 | -0.15 | 0.12 |
| | **Proposed Method** | **0.00** | **0.03** | **0.10** | **0.08** | **0.06** | 0.01 |
| **StanfordCars** | 1-Split Baseline | **0.00** | 0.00 | 0.00 | **0.00** | 0.00 | **0.00** |
| | 2-Split Baseline | **0.00** | 0.00 | 0.00 | **0.00** | 0.00 | **0.00** |
| | **Proposed Method** | **0.00** | **0.11** | **0.20** | -0.0 | **0.02** | **-0.0** |
| **CUBCars** | 1-Split Baseline | **0.00** | 0.11 | 0.14 | 0.10 | 0.06 | **-0.01** |
| | 2-Split Baseline | **0.00** | **0.15** | **0.22** | **0.16** | **0.11** | 0.01 |
| | **Proposed Method** | **0.00** | 0.05 | 0.08 | -0.02 | -0.03 | -0.0 |

