# OpenReview forum: "MonoQPM: Splitting Features into Concepts for Inherent Interpretability and Predictive Performance"
_TMLR — Under review for TMLR_

### Review · Reviewer_9RiH · 2026-06-11

**Summary Of Contributions:**

This paper analyzes the problem of polysemanticity in QPM models. The authors argue that although QPM models avoid superposition in their final feature representations, they are still susceptible to polysemanticity, which diminishes interpretability and causes interference by allowing a single feature to activate for multiple distinct concepts. To address this, the authors introduce Monosemantic QPM (MonoQPM), a post-hoc modification of a pretrained QPM that splits polysemantic QPM features into more specific concept-level components. MonoQPM works by examining the internal activation patterns of samples where a given feature is relevant, clustering those activations, and using the resulting clusters to split the original feature into more specific concepts. The original feature-class assignment matrix is then replaced with a new class-concept assignment matrix that reflects these discovered concepts. To evaluate MonoQPM, the authors use conformal prediction as a utility-focused measure of polysemantic interference, relying on metrics such as APS fuzziness, THR fuzziness, ACE, and NLL. They compare MonoQPM against the original QPM and hierarchical clustering baselines. The results show that MonoQPM generally reduces conformal prediction fuzziness and maintains calibration and NLL, suggesting that it can reduce polysemantic interference in QPM features.

This paper tackles an interesting problem in the context of QPM models by studying whether their final features, despite avoiding superposition, can still suffer from polysemanticity. This is a useful direction because it highlights that even models designed to be inherently interpretable may still contain features that are difficult to understand or that activate for multiple unrelated concepts. The proposed MonoQPM method is post-hoc and does not require retraining the full base model. Instead, it takes a pretrained QPM and attempts to split its learned features into more specific concept-level components, which can reduce polysemantic interference. The authors also introduced some metrics to help measure whether splitting features actually improves the model’s behavior, rather than only claiming that the resulting concepts appear more interpretable. However, the significance of the paper seems somewhat limited by its focus on QPMs. Since the method relies heavily on the sparse and white-box structure of QPM, it is unclear how directly this work would transfer to other interpretable models or to more standard dense neural networks. This limits the broader impact of the proposed approach. Another concern is that the practical use case of MonoQPM is not entirely clear. While the method improves conformal prediction efficiency and reduces some forms of interference, the paper could better explain where this would be useful in practice and why these improvements matter beyond the specific QPM setting. A further weakness is that although interpretability is one of the main motivations of the paper, the evidence for improved interpretability is still somewhat indirect. Much of the support comes from conformal prediction efficiency, concept-class similarity, and DINO-based coherence. These are useful metrics, but they do not fully demonstrate that a human user can better understand the model’s predictions after applying MonoQPM. The paper would be stronger if it included more concrete interpretability examples.

**Audience:**

Yes

**Audience Explanation:**

Although the audience is likely somewhat limited, there is likely to be some individuals interested in the findings of the paper. The paper focuses on a niche problem involving polysemanticity in QPM models, so it may not be broadly interesting to all TMLR readers. Researchers working on interpretability, concept-based models, or conformal prediction would likely find the findings relevant, especially the observation that even inherently interpretable models can still exhibit polysemantic features.

**Broader Impact Concerns:**

I do not see any major broader impact concerns with this work. The paper is primarily an analysis of polysemanticity in QPM models and introduces a post-hoc method for reducing polysemantic interference. Since the work is focused on interpretability and model analysis rather than deployment in a sensitive domain, the direct ethical risks appear limited.

**Claims And Evidence:**

Yes

**Claims Explanation:**

The claims made in the submission are mostly supported by the presented evidence. The authors provide quantitative results showing that MonoQPM reduces APS and THR fuzziness across several datasets and architectures, which supports the claim that splitting QPM features can reduce polysemantic interference. The use of conformal prediction as a utility-focused evaluation is also reasonably well motivated, since false residual activations on irrelevant classes should lead to larger prediction sets. However, some claims could be supported more convincingly. In particular, the claim that MonoQPM improves interpretability is supported mostly indirectly through conformal prediction efficiency, concept-class similarity, and DINO-based coherence. These metrics suggest that the resulting concepts are more coherent, but they do not fully demonstrate that human users can better understand the model’s predictions. The paper would be stronger if it included more concrete qualitative examples showing how specific QPM features are split, what visual concepts the resulting components correspond to, and how these splits improve the explanation of individual predictions.

**Requested Changes:**

I would request a more thorough analysis of the interpretability benefits of MonoQPM. While the paper argues that MonoQPM improves global interpretability by splitting polysemantic QPM features into more specific concepts, much of the evidence is based on indirect metrics. The paper would be stronger with qualitative examples showing how specific QPM features are split, which classes are assigned to each concept, what visual patterns those concepts correspond to, and how this improves explanations of individual predictions. I would also like to see a clearer discussion of scalability and runtime. MonoQPM requires activation collection, dimensionality reduction, GMM fitting, component selection, and construction of a new assignment matrix. Since the implementation appears to rely on multiple tools, such as torch-based models and scikit-learn clustering, reporting preprocessing time, memory cost, and runtime on larger datasets such as ImageNet would help clarify its practicality. Finally, the authors should expand the discussion of broader significance and future directions. Since the method is closely tied to QPM’s sparse white-box structure, it is unclear how well the findings transfer to other interpretable models or to dense models where polysemanticity and superposition are both present. A discussion of whether similar ideas could extend to models with superposition would help establish the broader relevance of the work.

---

> ### Author Response · Authors · 2026-06-23
> **Response to Reviewer 9RiH**
>
> We thank you for the thorough review and positive feedback. We are glad you consider this paper valuable and well evaluated. Additionally, we are thankful for the suggestions that have further improved the revised manuscript:
>
> We agree that qualitative examples are beneficial. That is why we have included a significant number of qualitative examples in the appendix, **Section E**, showcasing how individual features are split. Additionally, we now added a new **Figure 2** which highlights how MonoQPM improves the interpretability of QPM by identifying the specific concepts each polysemantic feature responds to for any class. That way, the class representation is more informative and one can reason about what concept was found in a concrete sample. We also added a discussion of it to the introduction.
>
> We added **Section D: Resource Requirements** that discusses the time and memory it takes to execute our proposed method. While the requirements do scale heavily with the number of samples, resulting in 12 hours and 300 GB of RAM with our current implementation for ImageNet, this one-time expenditure could be heavily optimized and is still relatively low compared to the training time.
>
> Finally, we added a discussion of broader significance to **Section F: Future Work**. While we agree that the full method is restricted to QPM, we see broad significance, as QPM itself can be broadly applied. Additionally, the proposed evaluation is likely able to evaluate one aspect of disentanglement in dense models. Finally, other interpretable models like SAEs could probably also benefit from the proposed disentanglement of individual neurons.
>
> We believe these changes improve the paper and are thankful for the suggestions.

---

### Review · Reviewer_JtQt · 2026-06-13

**Summary Of Contributions:**

This work focuses on polysemanticity in isolation within the interpretable QPM model free of feature superposition. It proposes the post-hoc MonoQPM method to decompose polysemantic features into monosemantic concepts by clustering internal activations, introduces a novel utility-driven paradigm for quantifying polysemanticity based on conformal prediction, and constructs the CUBCars benchmark dataset. Extensive experiments across multiple architectures and datasets verify that the method effectively reduces interference and improves conformal prediction efficiency with no degradation in classification accuracy.

**Audience:**

Yes

**Audience Explanation:**

Mechanistic interpretability is a central topic for trustworthy machine learning and a key focus of the TMLR audience. This work’s novel framework for studying polysemanticity in isolation from superposition, paired with its utility-focused measurement paradigm based on conformal prediction, will be of clear interest to researchers working on interpretable models, reliable classification, and the intersection of interpretability and conformal prediction.

**Broader Impact Concerns:**

This work uses standard, publicly available computer vision datasets, involves no sensitive data, biased applications, or critical deployment risks, and does not present ethical, fairness, or privacy concerns.  No unaddressed ethical implications require supplementation to a broader impact statement.

**Claims And Evidence:**

Yes

**Claims Explanation:**

1. The claim that polysemanticity can exist independently of superposition is solidly backed. The hierarchical clustering baseline shows a clear local minimum in fuzziness as concepts per feature increase (Figure 2), and this pattern holds across all tested architectures and datasets.
2. MonoQPM’s disentanglement effect is verified from two angles. On CUBCars, it achieves 100% accurate separation of bird and car concepts against ground truth. On all real-world datasets, it consistently cuts APS/THR fuzziness without hurting classification accuracy.
3. The utility-focused measurement paradigm is logically sound, and extensive ablation studies validate each design choice, making the results robust.

**Requested Changes:**

**Add ablation studies for key hyperparameters**:
1. The number of concepts per feature is selected only via BIC, with no comparison to other standard criteria (AIC, HQIC, or a simple elbow method).  It is worth showing whether the optimal concept count and the resulting fuzziness gains hold consistently across different selection rules.
2. The class-assignment threshold τ = 1/(2·n_c^d) is given as a fixed empirical value with no sweep or sensitivity analysis.

---

> ### Author Response · Authors · 2026-06-23
> **Response to Reviewer JtQt**
>
> We thank you for the review, excellent summary of our contributions, and very positive feedback. We are highly encouraged by the assessment that our claims regarding polysemanticity are solidly backed and that our measurement paradigm is logically sound.
>
> We also appreciate the constructive suggestions for additional ablation studies, which we agree strengthen the empirical foundation of MonoQPM. We have fully addressed both requested changes in the revised manuscript:
>
> * **Information Criteria Comparison:** We have added a new section to the Appendix (Section [B]: Impact of Information Criteria) evaluating BIC alongside AIC, HQIC, and the Elbow method. The results show that the fuzziness gains hold consistently across all criteria, with BIC being a good choice.
> * **Assignment Threshold Sensitivity:** We have added a new section to the Appendix (Section [C]: Impact of Assignment Threshold) detailing a sensitivity sweep over the threshold parameter $\tau$. The results demonstrate that MonoQPM robustly outperforms the baseline across the entire range of values and justifies our empirical default setting.
>
> Please refer to the updated PDF for the detailed figures and tables regarding these new experiments. We thank you again for helping us improve the manuscript.

---

### Review · Reviewer_gHPJ · 2026-06-18

**Summary Of Contributions:**

The Manuscript address the problem of interpretation in terms of the feature sets found in the training of the neural network. The Quantitative Predictive Monitoring  (QPM) is used to train the neural network with a significant result that the features that are linear combinations of lower features are suppressed. This is very good.

The method also bring new information of polysemantic features in more analyzable way. In most cases this is good and give a good  intuition about the functioning of the neural network. However, rooting out the polysemantic features an aiming to monosemantic features may not always be the best alternative to explain the behavior.

Another feature of the solution is that it may (I would really like to hear the opinions of the Authors on this) affect the generalization capabilities of the neural network. Think about the multilingual large language models. It has been found that anguages use features from the other languages to produce high quality output.  As these features are polysemantic, coming from different languages, I am afraid that the benefit of the other languages would be suppressed when forcing to a more monoseantic settings.

I will discuss aspects of this below.

**Additional Comments:**

A very interesting  and essential Manuscript and I would very much like to see how the topics mentioned will be addressed by the Authors.

**Audience:**

Yes

**Audience Explanation:**

The explainability of neural networks is paramount in Europe as it is required by the AI act.  So, a method that can provide bounds for the inference in medicine and robotics is very welcome. This is an alternative way to attribution studies, where the properties of the decision boundary is studied by linearized version of the neural network expanded close to the data point of inference.

Attribution also contains the role of the data used for the training  - if the data would contain features that arise from surroundings of the object of interest rather than from the object itself. This MonoQPM looks like an excellent algorithm to shed light on  the issue. In terms of batch corrections this is important as the origin of the X-ray image  may reflect the type of cancer as hospitals are specialized for certain kinds of cancer types.

A problem in the method here is that it is trying to force the explanation only on one batch of unique features specific to the class. This means that gender as polysemantic feature is removed, and we cannot check it for gender bias.

**Broader Impact Concerns:**

Yes,

The method looks like it would take out polysemantic features related to gender, ethnicity, age, .... that could affect a fair analysis in decisions that affect people.

**Claims And Evidence:**

Yes

**Claims Explanation:**

Generally a random initialization, for large enough layers create a set of templates that are enough to describe a vast set of features as linear combinations. More, the vanishing gradient in stochastic gradient based training usually keeps the lower layer templates unmoved fixing the initial.

Polysemanticity  may be a problem in a straightforward interpretability scenario aiming for understanding how inference was performed. It is good for many cases, where explicit "culprits" provide the an intuitive human explanation of the results.  However, the polysemanticity may be the key for generalization - different systems that relate themselves in similar ways using features that are shared.  The similarity can provide a flow of inference , where the logic is the same between the features but the representation is different.

Perhaps a simple example from robotics and equivariant neural networks.   Mathematical relations are expressed as a representation of a a symmetry group  (as an algebraic structure).  But, on other components of the robot there could be similar ore different representation of the symmetry group obeying the group multiplication table. A unified group representation for both concepts would bring more with less weights - of course, finding the irreducible presentations is  even a more effective way. Indeed, I would very much be interested how equivariant solutions in robotics works with the method on the Manuscript.

**Requested Changes:**

I am totally sure that doctors would prefer mono semantic features as polysemantic features like age, drinking habits, smoking are polysemantic features that doctors would like to know when making the diagnoses and planning the treatment.

This should be addressed in the paper, as know it was presented so that polysemantic features would be harmful for explainability. Actually they have paramount importance in, for example, estimating the role of exercising in preventing care plans, or more importantly looking and genetic oversensitivity for drugs in order to find out the proper treatment.

---

> ### Author Response · Authors · 2026-06-23
> **Response to Reviewer gHPJ**
>
> We thank you for your feedback and support of our methodology.
> Additionally, we thank you for your suggestions, which have helped
> improve our paper.
>
> In response, we added Figure 2 to the paper, clarifying the
> application of MonoQPM to a frozen QPM (Quadratic Programming
> Enhanced Model) and the meaning of polysemanticity. In the context
> of mechanistic interpretability, a neuron is "polysemantic" if it
> activates for multiple distinct and unrelated concepts in the data
> (Elhage et al., 2022). Therefore, attributes like age or gender are
> not polysemantic features themselves. They are distinct, meaningful
> concepts that might be undesirably entangled with other concepts
> inside a single polysemantic neuron.
>
> Additionally, a monosemantic neuron is not necessarily bound to a
> specific language, but could capture the concept "Old" in all
> languages, as observed in LLMs (Wendler et al., 2024) . Instead of
> disentangling multiple translations of "Old", MonoQPM aims to
> disentangle neurons that would respond to both the presence of age
> as well as other unrelated concepts (e.g., hospital-specific X-ray
> machinery) into individual neurons for each concept.
>
> Thus, in the resulting extended feature vector, no concept is
> removed or suppressed. On the contrary, the individual concepts that
> the frozen QPM picked up on are disentangled, which makes the
> desired fairness and bias checks more feasible. This is also why
> MonoQPM generally maintains, at minimum, the performance of the
> frozen QPM.
>
> We hope that the explanation surrounding Figure 2 clarifies these
> points.
>
> Finally, thank you for agreeing that multiple applications, such as
> robotics or the medical domain, are an exciting direction to apply
> this work to in the future.
>
> ---
>
> [1] Wendler, Chris, et al. "Do llamas work in english? on the latent
> language of multilingual transformers." Proceedings of the 62nd
> Annual Meeting of the Association for Computational Linguistics
> (Volume 1: Long Papers). 2024.

---

> > ### Comment · Reviewer_gHPJ · 2026-07-18
> > **On being old....**
> >
> > The Authors response clarify the method and the meaning of features. Thank you for the good work. I still have a situation where inherent concepts like Old or Age retains their meaning across the board. However,  the Age of people may be inferred from other components in the data and then it becomes a feature that could lead to polysemanticity and still remain essential - but could be discarded as an unnecessary feature as it can be created from others.
> >
> > Let me simplify even more, 2-2 contains three tokens but it is essentially 0 with one token
> >  even if the concept 0 has meanings that 2 has not. Orwellian aspects of the language are hard to handle as in Newspeak where the subtleties of the expressions are easily lost.
> >
> > But, I will not consider this against the publication of the Manuscript.

---

### Author Response · Authors · 2026-06-23
**Revised Manuscript Uploaded**

We have uploaded a revised PDF to incorporate the requested changes and improve the manuscript based on your valuable feedback.

**List of Changes:**

- **New Figure 2:** Added a qualitative example demonstrating how MonoQPM improves the interpretability of a frozen QPM by decomposing its polysemantic features into monosemantic ones. We have also integrated it into the introduction.
- **Appendix Section B:** Added a section analyzing the impact of the different Information Criteria used.
- **Appendix Section C:** Added a section demonstrating the impact of the Assignment Threshold $\tau$.
- **Appendix Section D:** Added a section detailing the resource requirements (time and memory) needed to apply MonoQPM to a frozen QPM.
- **Appendix Section F:** Expanded the section to discuss the broader significance and potential future applications of MonoQPM.

We thank you for your constructive reviews that helped improve this work.